# BBS8-dependent ciliary Hedgehog signaling governs cell fate in the white adipose tissue

Katharina Sieckmann [ID][1,12], Nora Winnerling[1,12], Dalila Juliana Silva Ribeiro[1], Seniz Yüksel [ID][1], Ronja Kardinal [ID][1], Lisa Maria Steinheuer[2], Fabian Frechen [ID][1], Luis Henrique Corrêa[3,4], Geza Schermann[5,6], Christina Klausen[1], Nelli Blank-Stein[7], Jonas Schulte-Schrepping[8], Collins Osei-Sarpong [ID][8], Matthias Becker [ID][8], Lorenzo Bonaguro [ID][8], Marc Beyer [ID][8,9], Helen Louise May-Simera[10], Jelena Zurkovic [ID][11], Christoph Thiele [ID][11], Kevin Thurley [ID][2], Lydia Sorokin [ID][3,4], Carmen Ruiz de Almodovar [ID][5,6], Elvira Mass [ID][7] & Dagmar Wachten [ID][1✉]

## Abstract

The primary cilium plays a crucial role in regulating whole-body energy metabolism, as reflected in Bardet-Biedl syndrome (BBS), where ciliary dysfunction leads to obesity due to hyperphagia and white adipose tissue (WAT) remodeling. Regulation of the fate and differentiation of adipocyte precursor cells (APCs) is essential for maintaining WAT homeostasis during obesity. Using $Bbs8^{-/-}$ mice that recapitulate the BBS patient phenotype, we demonstrate that primary cilia dysfunction reduces the stem-cell-like P1 APC sub-population by inducing a phenotypic switch to a fibrogenic progenitor state. This switch is characterized by extracellular matrix (ECM) remodeling and upregulation of the fibrosis marker CD9, even before the onset of obesity. Single-cell RNA sequencing reveals a direct transition of P1 APCs into fibrogenic progenitors, bypassing the committed P2 progenitor state. Ectopic ciliary Hedgehog signaling upon loss of BBS8 appears as a central driver of the molecular changes in $Bbs8^{-/-}$ APCs, altering their differentiation into adipocytes and promoting their lipid uptake. These findings unravel a novel role for primary cilia in governing APC fate by determining the balance between adipogenesis and fibrogenesis, and suggest potential therapeutic targets for obesity.

**Keywords** Cilia; BBS; Adipose Tissue; Cell Fate
**Subject Categories** Cell Adhesion, Polarity & Cytoskeleton; Molecular Biology of Disease; Stem Cells & Regenerative Medicine

## Introduction

Obesity is a global challenge, affecting hundreds of millions of people worldwide and leading to metabolic disorders like type 2 diabetes, non-alcoholic fatty liver disease, and cardiovascular disease (Blüher, 2019; Boutari and Mantzoros, 2022). The white adipose tissue (WAT) drives obesity by tissue expansion upon excess energy intake, reaching up to 50% of body mass in severely obese humans (Das et al, 2003). Understanding WAT homeostasis and remodeling during obesity is paramount to mitigating the onset of these diseases. Various phenotypic changes have been observed in WAT upon fat accumulation: adipocyte function is disturbed, pro-inflammatory cytokines are secreted, leading to immune cell recruitment, activation, and, in turn, inflammation, and the extracellular matrix (ECM) remodels, leading to tissue fibrosis (Hasegawa, 2022). Remarkably, WAT persists as one of the few tissues harboring a significant reservoir of adult stem cells even in adulthood. These are mesenchymal adipocyte progenitor cells (APCs) that commit to the adipocyte lineage to form adipocytes, a process called adipogenesis (Rosen and MacDougald, 2006). In response to caloric imbalance, WAT dynamically adapts by generating new adipocytes through adipogenesis (hyperplasia) or by storing excess lipids in existing adipocytes (hypertrophy). Balancing these two processes is key to maintaining tissue homeostasis (Chouchani and Kajimura, 2019; Vegiopoulos et al, 2017; White, 2023).

Various signaling pathways have been shown to regulate the intricate process of adipogenesis and, thereby, WAT homeostasis (Ye et al, 2023). A pivotal player in this context is the primary cilium, a microtubule-based membrane protrusion that extends from the plasma membrane of most vertebrate cells. Primary cilia operate as a cellular antenna by sensing extracellular stimuli and locally transducing this information into a cellular response (Gopalakrishnan et al, 2023). Primary cilia originate from a

[1]Institute of Innate Immunity, Biophysical Imaging, Medical Faculty, University of Bonn, Bonn, Germany. [2]Institute of Experimental Oncology, Biomathematics Division, University Hospital Bonn, Bonn, Germany. [3]Institute of Physiological Chemistry and Pathobiochemistry, University of Muenster, 48149 Muenster, Germany. [4]Cells-in-Motion Interfaculty Centre (CIMIC), University of Muenster, 48149 Muenster, Germany. [5]Institute for Neurovascular Cell Biology, Medical Faculty, University of Bonn, Bonn, Germany. [6]Schlegel Chair for Neurovascular Cell Biology, University of Bonn, Bonn, Germany. [7]Life and Medical Sciences (LIMES) Institute, Developmental Biology of the Immune System, University of Bonn, Bonn, Germany. [8]Platform for Single Cell Genomics and Epigenomics (Precise), German Center for Neurodegenerative Diseases and University of Bonn, 53127 Bonn, Germany. [9]Immunogenomics & Neurodegeneration, German Center for Neurodegenerative Diseases (DZNE), 53127 Bonn, Germany. [10]Institute of Molecular Physiology, Johannes Gutenberg-University, Mainz, Germany. [11]Life and Medical Sciences Institute, Biochemistry and Cell Biology of Lipids, University of Bonn, Bonn, Germany. [12]These authors contributed equally: Katharina Sieckmann, Nora Winnerling. ✉E-mail: dwachten@uni-bonn.de

modified mother centriole, the basal body that nucleates the microtubule-based ciliary axoneme. Protein transport in and out of the cilium is conveyed via the intraflagellar transport (IFT) machinery (Mill et al, 2023). Furthermore, the BBSome protein complex, formed by BBS proteins, functions as an adapter for retrograde transport out of the cilium (Nachury, 2018), and the transition zone (TZ) controls lateral diffusion of membrane proteins at the ciliary base (Park and Leroux, 2022). Together, the IFT, the BBSome, and the TZ determine the protein composition of the primary cilium. Although the molecular composition of the cilium has only been determined for some cell types, it is already evident that a unique ciliary protein composition delineates the distinct response and function of a given cell type (Wachten and Mick, 2021).

Primary cilia are present on mesenchymal stem cells (MSCs) and cells committed to the adipocyte lineage but are absent from mature adipocytes (Hilgendorf et al, 2019; Yanardag and Pugacheva, 2021). Genetic ablation of primary cilia in APCs of adult mice completely inhibits adipogenesis (Hilgendorf et al, 2019). Furthermore, signaling pathways that engage the primary cilium have been shown to exert opposing effects on adipogenesis: for example, canonical Hedgehog (Hh) signaling, which is transduced via primary cilia, inhibits adipogenesis, whereas omega-3 fatty acids, which are sensed via the free fatty acid receptor 4 (FFAR4) in the cilium, promote adipogenesis (Hilgendorf, 2021). Thus, the presence of primary cilia is key for adipocyte development and WAT homeostasis, and ciliary signaling pathways play a dual role in controlling adipogenesis.

Primary cilia dysfunction leads to severe disorders commonly termed ciliopathies, some of which are associated with obesity. This includes the Bardet-Biedl syndrome (BBS), caused by mutations in one of the *BBS* genes, encoding for BBS proteins that are part of the BBSome or its associated protein complexes (Forsythe and Beales, 2013).

BBS mouse models recapitulate the human phenotype and develop obesity over time. At the cellular level, this seems to be due to dysfunction of primary cilia in hypothalamic neurons, leading to hyperphagia, accompanied by decreased locomotion and hyperleptinemia (Brewer et al, 2022; DeMars et al, 2023; Engle et al, 2021; Rahmouni et al, 2008). However, the development of obesity also seems to be driven by primary cilia dysfunction outside the central nervous system, as normalizing food intake to suppress hyperphagia also increases adipose tissue mass (Rahmouni et al, 2008). However, the molecular mechanisms, in particular the changes in ciliary signaling underlying WAT remodeling in BBS, are not known.

Recent breakthroughs in single-cell technologies have revealed that APCs are a highly heterogeneous cell population. Based on gene expression analysis, lineage tracing, and surface marker expression (Altun et al, 2022; Scamfer et al, 2022), APCs include a stem cell-like, multipotent P1 subpopulation, a committed adipogenic P2 subpopulation, and different regulatory subpopulations, i.e., P3, also termed Aregs (Burl et al, 2018; Hepler et al, 2018; Merrick et al, 2019; Nahmgoong et al, 2022; Nguyen et al, 2021; Sarvari et al, 2021; Schwalie et al, 2018) (Fig. 1A). How the fate of these different subpopulations is determined at the molecular level is not well understood. Thus, it is important to shed light on APC complexity and diversity in terms of fate and function of the different subpopulations. Although the primary cilium and its associated signaling pathways have emerged as key regulators of adipogenesis, the cell-type-specific role of primary cilia signaling in the different APC subpopulations, as well as the contribution to determining APC fate and function and, thereby, WAT homeostasis are completely unknown.

Here, we investigated the role of primary cilia in controlling the fate and function of APC subpopulations in WAT under physiological conditions and during primary cilia dysfunction, i.e., in a BBS mouse model before the onset of obesity. Our results demonstrate that BBS8-dependent signaling determines the fate of APCs, in particular of the stem-cell-like P1 subpopulation. Loss of the BBSome component BBS8 reduces the stem-cell pool by inducing a phenotypic switch of P1 cells into a fibrogenic progenitor subpopulation, characterized by ectopic ciliary Hh signaling and loss of Hh responsiveness even before the onset of obesity in the lean state. In turn, this affects the adipogenic potential of APCs and their lipid uptake. Altogether, our data demonstrate a key function for primary cilia signaling in determining APC fate and function in WAT, and provide mechanistic insights into phenotypic fate changes underlying cellular remodeling in tissue *niches* in ciliopathies, in particular BBS.

## Results

### Primary cilia control adipocyte precursor subpopulations

We first analyzed whether primary cilia are present on all APCs. Based on results by others, we established a flow cytometry antibody marker panel to identify and sort the different subpopulations from the stromal vascular fraction (SVF) of subcutaneous inguinal WAT (iWAT): After lineage depletion (Lin: TER119+, CD45+, and CD31+, markers for erythrocytes, immune-, and endothelial cells, respectively) and selection of CD29+ (Integrin β1) and SCA1+ (stem cell antigen) APCs (Cho et al, 2019; Ferrero et al, 2020; Schwalie et al, 2018), we used the following markers to identify the APC subpopulations: CD55 (complement decay-accelerating factor) and CD26 (DPP4) for P1, CD54 (ICAM-1) for P2, and CD142 (tissue factor) for P3 (Fig. 1A–C; Appendix Fig. S1A) (Merrick et al, 2019; Schwalie et al, 2018). P1 was the most abundant APC subpopulation, followed by P2 and P3 (Fig. 1D). APCs of all three subpopulations displayed primary cilia, with P1 showing the highest proportion of ciliated cells, followed by P2 and P3 (Fig. 1E,F), whereas the cilia length was not different between the subpopulations (Appendix Fig. S1B). These results demonstrate that the proportion of ciliated cells is inversely related to differentiation towards committed adipocytes.

Next, we tested whether the different subpopulations maintain their in vivo characteristics when cultured in vitro for 48 h. To determine the differentiation state of the different APCs, we analyzed FABP4 expression, an adipocyte commitment marker. Since it has been shown that regulatory P3 cells are resistant to undergoing adipogenesis, we focused on the P1 and P2 subpopulations (Schwalie et al, 2018). Only a few P1 but the majority of P2 cells were positive for FABP4 (Fig. 1G,H), indicating that P1 cells are not yet committed and more stem-cell like, whereas P2 cells are already committed to the adipogenic lineage,

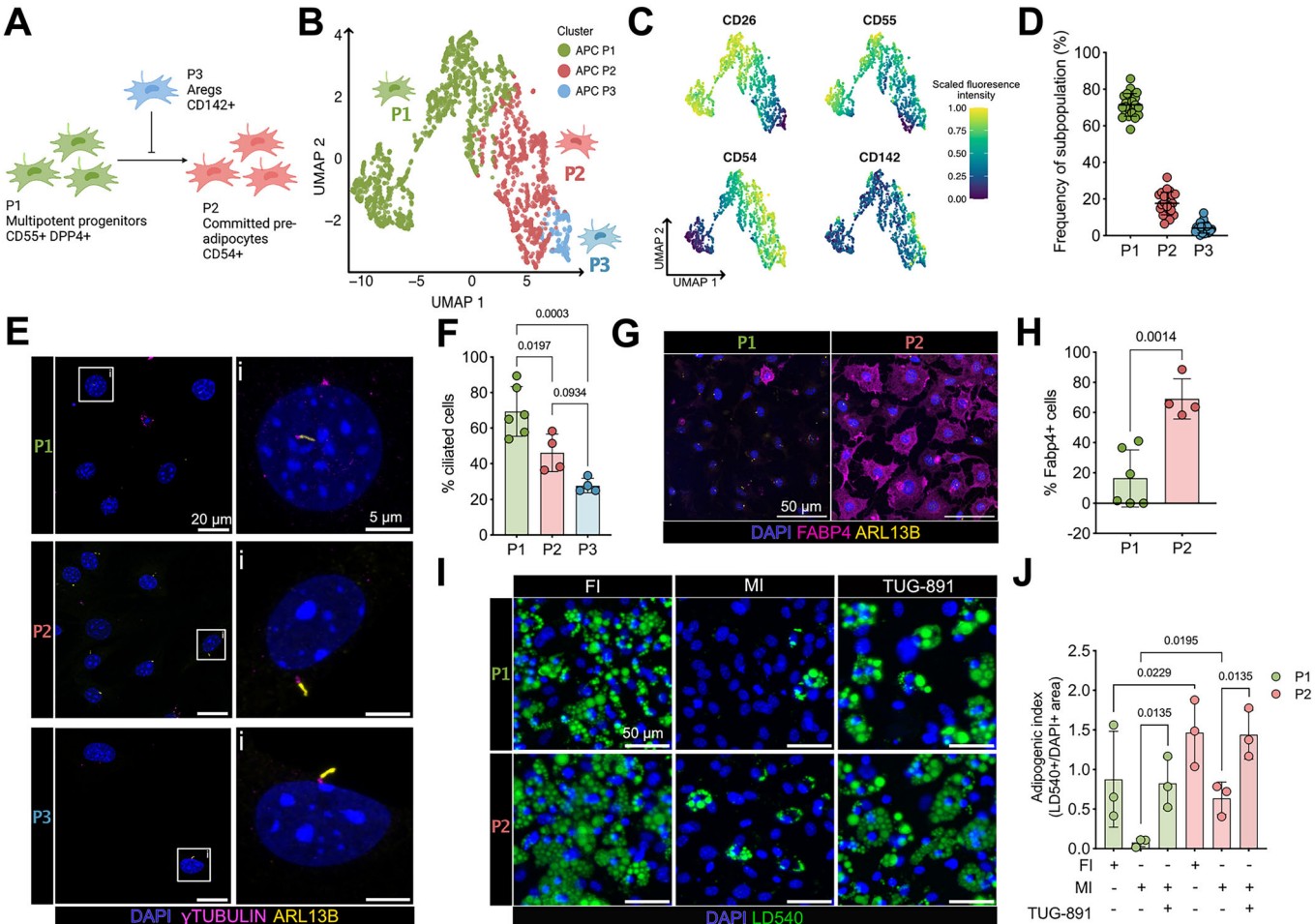

**Figure 1. Primary cilia regulate the adipogenic potential of adipocyte precursor subpopulations (APCs).**

(A) Schematics of the three adipocyte precursor subpopulations (APC) and their respective function and developmental hierarchy. (B) Dimension reduction analysis from flow cytometry data based on the expression of the surface markers CD26, CD55, CD54, and CD142 revealed three distinct APC subpopulations (P1, P2, and P3) in the uniform manifold approximation and projection (UMAP) two-dimensional map. (C) Fluorescence intensities of the different markers in the UMAP, highlighting the contribution of each marker to the respective clusters. (D) Frequency distribution of P1–P3 from the total APC pool (SCA1+ cells). Each data point represents one animal. (E) Fluorescence confocal images of FACS-sorted P1, P2, and P3 cells, labeled against ARL13B (yellow, cilia), γ-tubulin (magenta, basal body), and with DAPI (blue, DNA). (F) Quantification of the ciliation rate from the sorted APCs subpopulations. Each data point represents one image from $n = 3$ wild-type mice in total (>4 cilia per $n$), $p$ values were determined using one-way ANOVA. (G) Fluorescence confocal images of FACS-sorted P1 and P2 cells, labeled against FABP4 (magenta, commitment adipogenic lineage), ARL13B, and with DAPI. (H) Quantification of FABP4-positive cells. Each data point represents one image from $n = 3$ wild-type mice in total, $p$ values were determined using an unpaired Student's t-test. (I) Fluorescence images of in vitro differentiated P1 and P2 APCs. Cells were differentiated for 8 days under full induction (FI) or minimal induction (MI) with or without the addition of TUG-891 (100 μM). Cells were stained with LD540 (green, lipid droplets) and DAPI. (J) Adipogenic potential of P1 and P2 APCs was quantified by calculating the ratio of the lipid droplet area and the area of nuclei. Each data point represents one image from $n = 3$ wild-type mice in total, $p$ values were determined using two-way ANOVA. APCs were isolated from wild-type mice at 7–9 weeks. All data were represented as mean ± SD. Scale bars are indicated. Source data are available online for this figure

similar to what has been described in vivo (Merrick et al, 2019; Schwalie et al, 2018).

To investigate cilia-specific roles during adipogenesis, a minimal induction medium that allows to observe downstream effects of ciliary signaling is a prerequisite (Hilgendorf et al, 2019). Under minimal induction conditions alone, some P2 but none of the P1 cells differentiated into adipocytes (Fig. 1I,J), in line with P2 cells already being committed to the adipocyte lineage. To investigate the influence of ciliary signaling on adipogenesis, we stimulated the free fatty acid receptor 4 (FFAR4), which has been shown to induce ciliary signaling and promote adipogenesis (Hilgendorf et al, 2019). In the presence of TUG-891, a FFAR4

agonist, both P1 and P2 cells strongly differentiated into adipocytes, even to the same extent as upon addition of a full induction cocktail (Fig. 1J). These results indicate that both P1 and P2 cells respond to a cilia-specific, adipogenic stimulus.

## Loss of BBS8 results in obesity

To investigate whether primary cilia dysfunction, in particular loss of a BBS protein, affects fate and function of APCs, we used a mouse model that lacks BBS8 ($Bbs8^{-/-}$), a component of the BBSome complex (Tadenev et al, 2011). $Bbs8^{-/-}$ mice are lean until 10 weeks and then start to develop obesity (Fig. 2A). The gain in

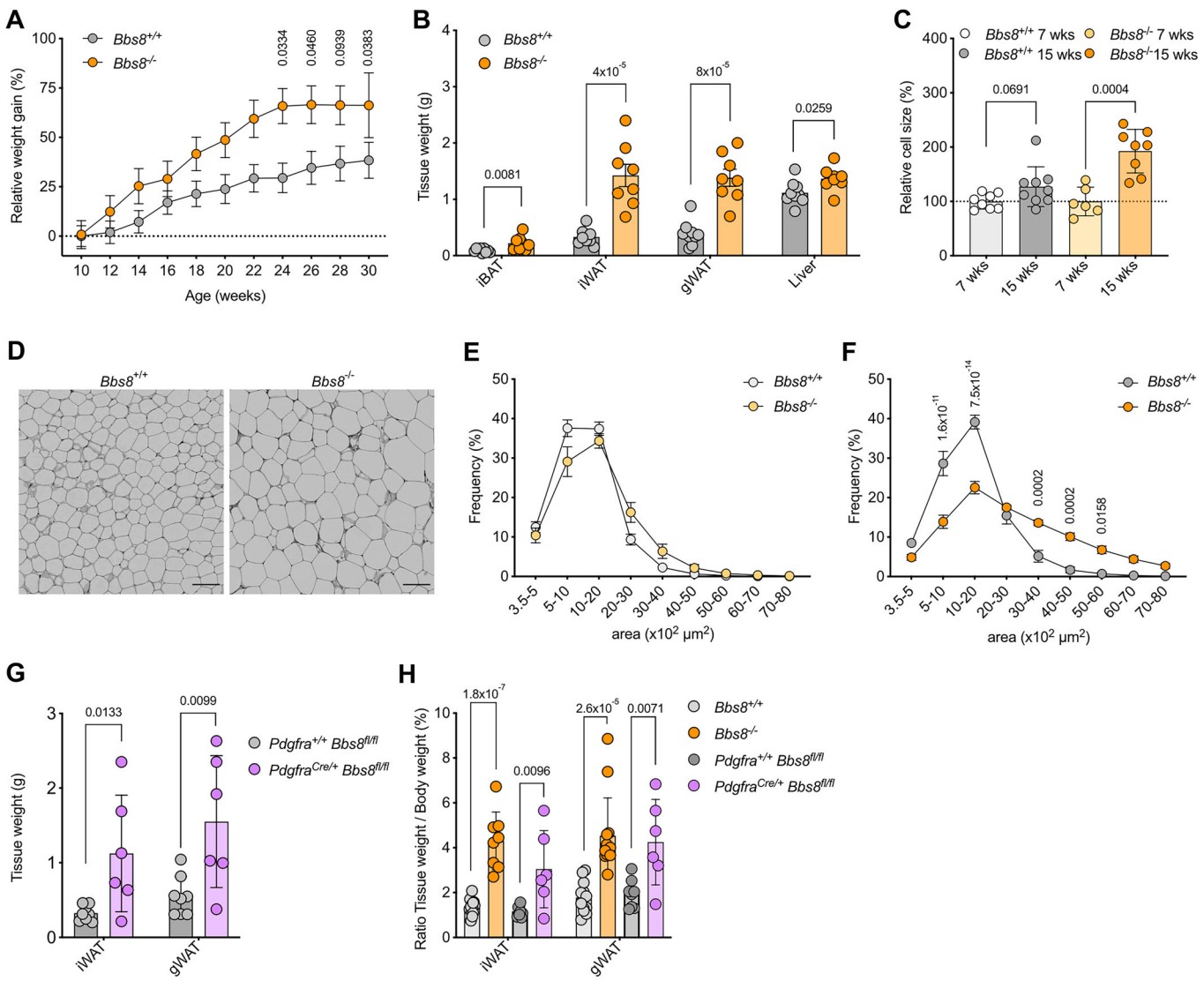

**Figure 2. Loss of BBS8 in APCs results in obesity.**

(A) Relative body weight gain of chow diet-fed $Bbs8^{+/+}$ and $Bbs8^{-/-}$ mice. Weights were normalized to the mean body weight of their respective genotypes at the lean time point (10 weeks). Data were shown mean ± SEM, p values were determined using a two-way ANOVA with repeated measurements (mixed models). Post hoc p value correction for multiple testing was performed using Bonferroni adjustment ($n = 14$). (B) Tissue weights of adipose tissues and liver at 15–18 weeks (obese state). Data were shown as mean ± SD, p values have been determined using an unpaired Student's t-test determined, each data point represents one animal. (C) Relative adipocyte cell size at the lean (7 weeks) and obese (15–18 weeks) state. Cell size was normalized to the lean time point of the respective genotype. Data were shown as mean ± SD, p values have been determined using an unpaired Student's t-test, and each data point represents one animal. (D) Representative images of hematoxylin-eosin (HE)-stained gonadal WAT (gWAT) at 15–18 weeks. Scale bar = 100 μm. (E, F), Adipocyte size distribution quantified using images as shown in (D) at 7 weeks (lean state) (E) and 15–18 weeks (obese state) (F). Data were shown as mean ± SEM, p values have been determined using an unpaired multiple Mann–Whitney tests with Bonferroni correction ($n = 6$–9 animals per genotype with >100 cells each). (G) Adipose tissue weights of chow diet fed $Pdgfra^{+/+}$, $Bbs8^{flox/flox}$ and $Pdgfra^{Cre/+}$, $Bbs8^{flox/flox}$ mice at 17–25 weeks (obese state). Data were shown as mean ± SD, p values have been determined using an unpaired Student's t-test determined, each data point represents one animal. (H) Ratio of adipose tissue vs. body weight for each respective animal from the different mouse lines. Data were shown as mean ± SD, p values have been determined using an unpaired Student's t-test determined, each data point represents one animal. Source data are available online for this figure

body weight is mirrored by an increase in all fat depot weights (Fig. 2B). We analyzed adipose tissue plasticity in further detail using our AdipoQ analysis pipeline (Sieckmann et al, 2022), which determines adipocyte size in a tissue section as well as the cell distribution within the tissue. We could show that the adipocyte cell size remained unchanged at 7 weeks, when the mice are lean, but it is increased in $Bbs8^{-/-}$ compared to $Bbs8^{+/+}$ mice at 15–17 weeks of age, when mice are considered obese (Fig. 2C–F).

Thus, obese $Bbs8^{-/-}$ mice display adipose tissue hypertrophy (Fig. 2F). To investigate the contribution of the mesenchymal lineage and APCs to WAT remodeling, and as APCs are platelet-derived growth factor receptor (Pdgfr) α positive, we generated conditional knockout mice, lacking $Bbs8$ in Pdgfrα-expressing cells ($Pdgfra^{+/cre}$, $Bbs8^{flox/flox}$). Similar to global $Bbs8^{-/-}$ mice, these mice also developed obesity compared to control littermates (Appendix Fig. S2A), and their fat depot weights were increased at 17–21 weeks

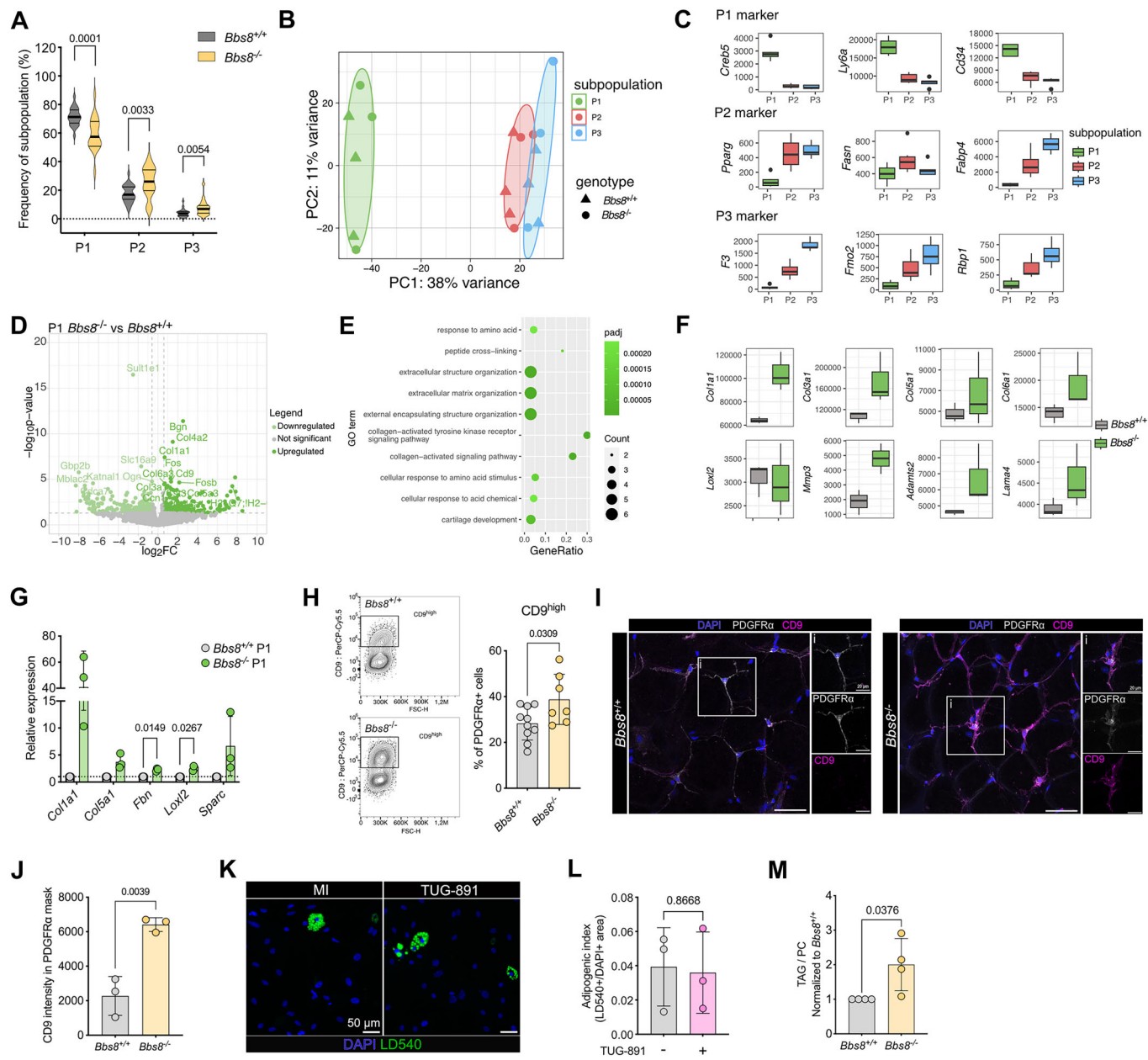

of age (Fig. 2G). When comparing the ratio of adipose tissue to body weight for the global and conditional knockout mice, we observed that the increase was similar for both (Fig. 2H), demonstrating that loss of BBS8 in Pdgfrα-expressing cells largely contributes to WAT remodeling. In summary, loss of BBS8 leads to obesity due to adipocyte hypertrophy and WAT remodeling, which is driven by primary cilia dysfunction in Pdgfrα-expressing cells.

## Loss of BBS8 induces a fibrogenic switch and ECM remodeling

As we aimed to investigate the role of primary cilia in controlling APC fate and function independent of secondary effects due to obesity development, we focused all our subsequent analyses on the lean time point (5–9 weeks) for all mouse models. To investigate the role of BBS8

in the P1–P3 APC subpopulations, we first compared their relative frequencies in the SVF from iWAT of lean $Bbs8^{+/+}$ and $Bbs8^{-/-}$ mice using the flow cytometry strategy described above. In comparison to $Bbs8^{+/+}$ mice, the P1 subpopulation in $Bbs8^{-/-}$ mice was significantly reduced, whereas the P2 and P3 subpopulations were slightly increased (Fig. 3A), indicating that lean $Bbs8^{-/-}$ mice harbor fewer stem cell-like APCs, which is recapitulated in the $Pdgfra^{+/cre}$, $Bbs8^{flox/flox}$ mice (Appendix Fig. S2B).

To investigate whether this change in frequency distribution resulted in cellular changes, we sorted P1–P3 from $Bbs8^{+/+}$ and $Bbs8^{-/-}$ mice by flow cytometry and performed bulk RNA-sequencing. According to the principal component analysis, overall gene expression was more similar in the P2 and P3 subpopulations and distinct from P1 (Fig. 3B), which is also reflected in the marker expression: P1 displayed high $Creb5$ (cAMP-responsive element

**Figure 3.  Loss of Bbs8 induces a fibrogenic switch and ECM remodeling.**

(A) Frequency distribution of P1–P3 from the total APC pool, isolated from $Bbs8^{+/+}$ ($n = 23$) and $Bbs8^{-/-}$ mice ($n = 20$) at 6–9 weeks (lean state). $P$ values were determined using an unpaired Student's $t$-test. (B) Principal component analysis (PCA) of the bulk RNA-seq data of FACS-sorted P1, P2, and P3 APCs isolated from $Bbs8^{+/+}$ and $Bbs8^{-/-}$ mice at 7–8 weeks (lean state) ($n = 3$) based on the 500 most variable genes. (C) Tukey box plots of the mean expression level of published P1, P2, and P3 markers. Data were shown as mean ± SD ($n = 3$). (D) Volcano plots depicting the DEGs for P1 APCs from $Bbs8^{-/-}$ and $Bbs8^{+/+}$ mice, isolated at 7–8 weeks (lean state) ($n = 3$). (E) Overrepresentation analysis (ORA) of DEGs, highlighting the top ten biological processes from gene ontology analysis in $Bbs8^{-/-}$ P1 APCs compared to $Bbs8^{+/+}$ P1 APCs. (F) Tukey box plots of the mean expression level of selected fibrosis marker genes in $Bbs8^{+/+}$ and $Bbs8^{-/-}$ P1 APCs. Data were shown as mean ± SD ($n = 3$). (G) Relative mRNA expression of fibrosis marker of sorted P1 APCs isolated from $Bbs8^{+/+}$ and $Bbs8^{-/-}$ mice at 7 weeks (lean state) assessed by qRT-PCR. mRNA expression was normalized to $Bbs8^{+/+}$. (H) Gating strategy demonstrated on concatenated $Bbs8^{+/+}$ and $Bbs8^{-/-}$ files (left). Quantification of CD9$^{high}$ cell surface expression on Pdgfr$\alpha^+$ cells from $Bbs8^{+/+}$ and $Bbs8^{-/-}$ mice at 5–8 weeks (lean state) (right). Data were shown as mean ± SD, $p$ values have been determined using an unpaired Student's $t$-test ($n = 7$–9). (I) Whole-mount staining of gWAT from $Bbs8^{+/+}$ and $Bbs8^{-/-}$ mice at 5 weeks (lean state), labeled with DAPI (blue), anti-Pdgfr$\alpha$ antibody (white) to label APCs, and anti-CD9 antibody (magenta). Scale bars are indicated. (J) Quantification of CD9 mean intensity in a Pdgfr$\alpha$ mask depicted in (I). Each data point represents a technical replicate from $n = 1$ mouse, $p$ value was determined using an unpaired Student's $t$-test. (K) Fluorescence images of FACS-sorted, in vitro differentiated gWAT Pdgfr$\alpha^+$/CD9$^{hi}$ $Bbs8^{-/-}$ cells. Cells were differentiated for 8 days using the minimal induction (MI) with and without TUG-891 (100 μM). Cells were stained with LD540 (lipid droplets) and DAPI. Scale bars is indicated. (L) Quantification of the adipogenic potential of Pdgfr$\alpha^+$/CD9$^{high}$ $Bbs8^{-/-}$ cells, isolated from mice at 7 weeks (lean state). Each data point represents the mean of all images from $n = 3$ mice in total. All data were represented as mean ± SD, $p$ values were determined using an unpaired Student's $t$-test. (M) Uptake of triacyl glycerides (TAG) with respect to phosphatidylcholine (PC) after 4 days of in vitro differentiation of P1 cells from $Bbs8^{+/+}$ and $Bbs8^{-/-}$ mice, isolated at 7 weeks (lean state), with full induction cocktail. Data were normalized to $Bbs8^{+/+}$. Each data point represents one mouse from $n = 4$ mice in total. All data were represented as mean ± SD, $p$ values were determined using an unpaired Student's $t$-test. Source data are available online for this figure

binding-protein 5), *Ly6a* (lymphocyte antigen 5 family member A), and *Cd34* expression, P2 high *Pparg* (peroxisome proliferator-activated receptor gamma), *Fasn* (fatty acid synthase), and *Fabp4* (fatty acid binding-protein 4) expression, whereas *F3* (coagulation factor III), *Fmo2* (flavin-containing dimethylaniline monoxygenase 2), and *Rbp1* (retinol-binding-protein 1) were enriched in P3 (Fig. 3C; Appendix Fig. S3A). When comparing the differentially expressed genes (DEGs, Dataset EV1) in P1–P3 between lean $Bbs8^{+/+}$ and $Bbs8^{-/-}$ mice, the highest number of DEGs was observed in the P1 subpopulation, whereas the P3 subpopulations showed the least changes (Appendix Fig. S3B). This is further depicted in the volcano plots, showing the DEGs for each subpopulation (Fig. 3D; Appendix Fig. S3C–F). Gene ontology enrichment analysis of the DEGs in P1 revealed that predominantly genes associated with extracellular matrix (ECM) remodeling were upregulated in $Bbs8^{-/-}$ mice (Fig. 3E). This included genes encoding for collagens (*Col1a1*, *Col3a1*, *Col5a1*, and *Col6a1*) as well for ECM-regulating enzymes like *Loxl2* (lysyl oxidase-like 2), *Mmp3* (metalloproteinase-3), *Adamts2* (ADAM metallopeptidase with thrombospondin type 1 motif 2), and laminin *Lama4* (Fig. 3F). We verified the increased expression of the ECM remodeling-associated genes by quantitative PCR in $Bbs8^{-/-}$ mice (Fig. 3G). ECM remodeling has been implicated in controlling preadipocyte maturation and regulating stem-cell fate (Jääskelainen et al, 2023; Watt and Huck, 2013). During early commitment and differentiation, the fibrillar collagens (types I and III) are predominantly expressed, whereas a switch to basement membrane-associated collagens (types IV, VI, XV, and XVIII) is associated with late differentiation (Jääskelainen et al, 2023). However, ECM remodeling is also a hallmark of tissue fibrosis, and excess ECM and fibrosis have been observed in WAT depots during obesity (Sun et al, 2023). The collagens that are upregulated during obesity in human and mouse adipose tissue (Jääskelainen et al, 2023) also include the collagens upregulated in lean $Bbs8^{-/-}$ mice. Altogether, loss of BBS8 reduced the stem-cell-like P1 APC subpopulation and increased the expression of ECM components in the lean state before the onset of obesity.

Our data suggests that loss of BBS8 drives a fibrogenic, phenotypic shift in Pdgfrα-expressing progenitor cells in lean WAT. So far, a fibrogenic shift in this cell population has only been observed during obesity, associated with a shift from a CD9$^{low}$

(tetraspanin-29) toward a CD9$^{high}$ pro-fibrotic phenotype (Marcelin et al, 2017). We, therefore, analyzed CD9 expression in APCs from lean $Bbs8^{+/+}$ and $Bbs8^{-/-}$ mice by gating the CD9$^{high}$ *vs.* CD9$^{low}$ population in Pdgfrα-expressing cells using flow cytometry (Fig. 3H; Appendix Fig. S3H). Strikingly, the frequency of the CD9$^{high}$ subpopulation in Pdgfrα-expressing cells in the SVF was increased in lean $Bbs8^{-/-}$ compared to $Bbs8^{+/+}$ mice (Fig. 3H; Appendix Fig. S3H), demonstrating that loss of BBS8 induces a fibrogenic, phenotypic switch in P1 cells already in the lean state. We confirmed the increase in CD9 expression also on the protein level using whole-mount labeling of adipose tissue from lean $Bbs8^{+/+}$ and $Bbs8^{-/-}$ mice (5 wks) (Fig. 3I,J) and lean $Pdgfra^{+/+}$ $Bbs8^{floxlflox}$ and $Pdgfra^{Cre/+}$ $Bbs8^{floxlflox}$ mice (7 wks) (Appendix Fig. S3I,J). This was further supported by histological analysis on adipose tissue sections, also showing remodeling of the ECM in WAT from lean $Bbs8^{-/-}$ mice (Appendix Fig. S3K).

In summary, our data demonstrate that loss of the BBSome component BBS8 dysregulates the fate of the APC P1 population in the lean state. In fact, loss of BBS8 shifts P1 towards a fibrogenic phenotype, with enhanced fibrillar collagen synthesis and, thereby, WAT remodeling in the lean state.

## Phenotypic switch of P1 cells impairs adipogenesis and promotes lipid uptake

To investigate whether the changes in $Bbs8^{-/-}$ APCs affect their function and impair adipocyte differentiation, we analyzed adipogenesis of APCs in vitro. It has been previously shown that the fibrogenic switch to CD9$^{high}$ cells during obesity diminishes their adipogenic potential (Marcelin et al, 2017). To investigate whether a similar phenotype is also observed in the CD9$^{high}$ population from lean $Bbs8^{-/-}$ mice, we performed in vitro differentiation in the presence of TUG-891, which promotes cilia-dependent adipogenesis (Fig. 1J) (Hilgendorf et al, 2019). Our results demonstrate that the CD9$^{high}$ population in lean $Bbs8^{-/-}$ mice does not respond to an adipogenic stimulus (Fig. 3K,L), as has been shown for CD9$^{high}$ cells from obese mice (Marcelin et al, 2017). We also analyzed whether the cell-intrinsic properties of lipid uptake are altered. To this end, we isolated P1 APCs from lean $Bbs8^{+/+}$ and $Bbs8^{-/-}$ mice, cultured and differentiated them in vitro, and fed the cells with alkyne-labeled

fatty acids. Subsequent mass spectrometric analysis allows tracing of fatty acid incorporation into glycerolipids. Our results revealed that the synthesis of triacyl glycerides (TAG) is increased in lean *Bbs8*⁻/⁻ compared to lean *Bbs8*⁺/⁺ adipocytes (Fig. 3M). Altogether, these results show that the phenotypic switch in APCs from lean *Bbs8*⁻/⁻ mice diminishes their adipogenic potential and promotes fatty acid uptake and, in turn, adipocyte hypertrophy.

## Loss of BBS8 drives P1 cells towards a fibrogenic fate

To analyze the fibrogenic switch in more detail, we performed single-cell RNA-sequencing (scRNA-seq) of the iWAT SVF from lean *Bbs8*⁺/⁺ and *Bbs8*⁻/⁻ mice. Unsupervised clustering of the gene expression profiles identified the following known main cell types: APC P1–P3, fibroblastic progenitor cells (FPC), pericytes, endothelial cells as well as different immune cells (Fig. 4A; Appendix Fig. S4A), based on described markers for the different cell types and unbiasedly identified signature genes (Appendix Fig. S4B,C; Dataset EV2). Since we observed a shift towards a fibrogenic phenotype in lean *Bbs8*⁻/⁻, we focused on the APC subpopulations and the FPCs for the following analyses. The APC cell populations were characterized based on the expression of *Dpp4* (dipeptidyl peptidase 4) and *Pi16* (peptidase inhibitor 16) (for P1), *Col15a1* (collagen 15a1) (for P2), *Mmp3* (matrix metalloprotease 3), and *Bgn* (biglycan) (for P3) (Fig. 4B,D), as shown before (Merrick et al, 2019; Schwalie et al, 2018). Down-sampling to compare the abundance of cell populations between lean *Bbs8*⁺/⁺ and lean *Bbs8*⁻/⁻ mice revealed that the FPC abundance was increased in lean *Bbs8*⁻/⁻ compared to lean Bbs8⁺/⁺ mice (Fig. 4C,E; Appendix Fig. S4D,E). According to the Reactome analysis, the FPCs were characterized by a fibrotic signature, including an ECM remodeling signature (Appendix Fig. S4F; Datasets EV3 and EV4), underlining the fibrotic phenotype observed in the tissue under lean conditions.

To spatially analyze the fibrogenic switch in the adipose tissue on the protein level, we performed whole-mount labeling of adipose tissue from lean *Bbs8*⁺/⁺ and *Bbs8*⁻/⁻ mice (5 wks), as well as from lean *Pdgfra*⁺/⁺ *Bbs8*^flox/flox and *Pdgfra*^Cre/+ *Bbs8*^flox/flox mice (7 wks). We labeled against fibronectin, and our results demonstrate that the expression is dramatically increased in lean *Bbs8*⁻/⁻ and *Pdgfra*^Cre/+ *Bbs8*^flox/flox mice compared to *Bbs8*⁺/⁺ and *Pdgfra*⁺/⁺ *Bbs8*^flox/flox mice, respectively (Fig. 4F,G; Appendix Fig. S4G,H).

To investigate the fate change at the single-cell level, we performed computational trajectory inferences on the P1–P3 subpopulations and the FPCs (Fig. 4H) using Monocle3. Pseudo-temporal analysis predicted, in line with previous data (Merrick et al, 2019; Schwalie et al, 2018), that P1 progenitor cells develop into P2 or P3 (Fig. 4H), and FPCs predominantly develop from P2 (Fig. 4H). However, in lean *Bbs8*⁻/⁻ mice, P1 cells preferably develop into FPC and then into P2 cells (Fig. 4H). Thus, these data further underline that loss of BBS8 results in a change in cell fate, with trajectory analysis supporting that P1 cells directly entering a fibroblastic precursor cell state with compromised ability to develop into P2 or P3.

## A subpopulation of P2 cells preferably interacts with the WAT vasculature, and their interactions are enriched in *Bbs8*⁻/⁻ mice

A change in cell fate not only alters cell-autonomous functions but also determines how cells communicate in the tissue niche. APCs

interact with various cell types in the different WAT depots (Hildreth et al, 2021; Massier et al, 2023; Vijay et al, 2020). It has been described that the stem-cell-like P1 APCs reside in the reticular interstitium/stroma between adipocytes (Merrick et al, 2019), while differentiation of committed precursor cells occurs along the vessels within the tissue in the perivascular *niche* (Gupta et al, 2012; Tang et al, 2008). As loss of BBS8 affects the differentiation trajectory of APCs in lean mice, we wondered whether these changes also altered cell–cell interactions in the tissue. To this end, we applied CellChat (Jin et al, 2021), which identifies potential cell-to-cell molecular interactions, based on a database of ligand-receptor interactions, in scRNA-seq data. CellChat analysis showed a complex interaction map between the different APC populations, pericytes, endothelial cells, and FPCs when looking at the combined data from lean *Bbs8*⁺/⁺ and lean *Bbs8*⁻/⁻ mice (Fig. 4I). The three APC populations were the most active in sending signals to the other cells, mainly in their predicted interactions towards endothelial cells (Fig. 4I). Thus, we further characterized these cellular interactions with endothelial cells by sub-clustering (Dataset EV5) and focused on the APC P2 subpopulation, which has been proposed to be close to the vasculature. We identified a subcluster in the P2 subpopulation, named APC P2_1 (Dataset EV5), which showed the strongest potential interaction with the endothelial cells (Fig. 4J). The APC P2_1 subcluster was predominantly characterized by an enrichment for laminin and collagen ligands (Fig. 4K; Appendix Fig. S4I). Strikingly, the stronger interaction was due to an increase in the collagen and laminin interactions, which were significantly increased in lean *Bbs8*⁻/⁻ compared to lean *Bbs8*⁺/⁺ mice (Fig. 4L; Appendix Fig. S4I).

Altogether, our data indicate that loss of BBS8 leads to stronger interactions of the APCs, in particular P2_1, with the vasculature due to an increase in the expression of laminin and collagen ECM components. This suggests that primary cilia dysfunction in APCs upon loss of BBS8 not only causes a fibrogenic switch in the stem-cell-like P1 APCs but also alters the cellular interactions of committed APCs (P2_1) in the perivascular *niche* by ECM remodeling, which could further impact the obesity phenotype.

## Loss of BBS8 results in ectopic Hedgehog signaling and diminishes the Hh-dependent signaling response

To investigate the molecular mechanisms underlying the cilia-dependent change in cell fate, we analyzed primary cilia signaling in APCs isolated from lean *Bbs8*⁺/⁺ and *Bbs8*⁻/⁻ mice in more detail. When analyzing the single-cell transcriptomics data, we observed that the Hedgehog (Hh) signature was increased in *Bbs8*⁻/⁻ compared to *Bbs8*⁺/⁺ APCs (Fig. 5A). To test whether a low-grade stimulation of Hh signaling is sufficient to induce a fibrogenic gene expression pattern, we stimulated 3T3-L1 cells, a murine preadipocyte model, with 100 nM SAG for 24 h. The stimulation with SAG increased the ciliary SMO localization and induced Hh target gene expression (Appendix Fig. S5A–C). SAG treatment also induced GLI1 protein expression and inhibited processing of GLI3 into its repressor form (GLI3R) (Appendix Fig. S5D,E). Furthermore, low-grade stimulation of Hh signaling was sufficient to induce a fibrogenic gene expression program, inducing the expression of *Col1a1, Sparc,* and *Tnc* (Fig. 5B).

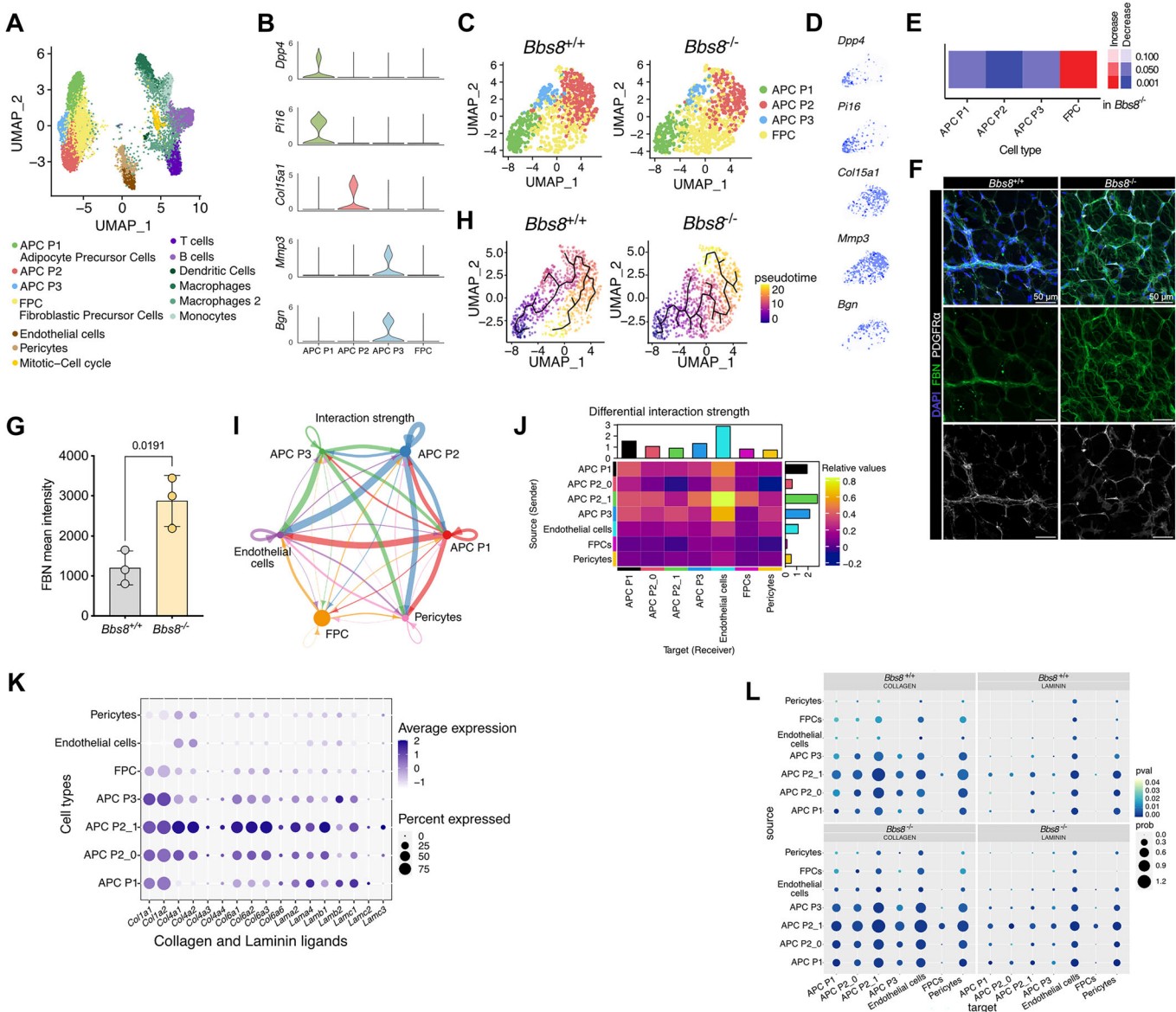

**Figure 4. Loss of BBS8 drives P1 cells into fibrogenic cells.**

(A) UMAP plot analysis from scRNA-seq data on iWAT SVF from lean *Bbs8*^+/+^ and *Bbs8*^−/−^ mice (7–8 weeks) shows several distinct cell clusters. (B) Expression levels of published P1, P2, and P3 markers in APCs and fibrogenic precursor cells (FPC) (*n* = 1). (C) UMAP plot of the P1-P3 APCs and FPC cluster in *Bbs8*^+/+^ and *Bbs8*^−/−^ mice. (D) P1, P2, and P3 marker expression overlaid over the subsetted UMAP displayed in (C). (E) Differential abundance analysis of downsampled clusters in *Bbs8*^−/−^ compared to *Bbs8*^+/+^ mice, the color-code shows FDR-corrected p-values. (F) Whole-mount staining of gWAT of *Bbs8*^+/+^ and *Bbs8*^−/−^ mice at 7 weeks (lean state), labeled with DAPI (blue), anti-Pdgfrα antibody (white) to label APCs, and anti-fibronectin antibody (FBN, green). Scale bar = 50 μm. (G) Quantification of the mean intensity of FBN from (F). Each data point represents a technical replicate from *n* = 1 mouse, *p* value was determined using an unpaired Student's *t*-test. (H) In silico pseudo-time analysis of the P1-P3 APCs, and FPCs along differentiation trajectories using Monocle3. (I) Circos plots depicting the interaction strength between endothelial cells, pericytes, APC P1-P3, and FPCs as determined via the CellChat pipeline. (J) Heat map depicting the differential interaction strength of the different APC subpopulations, endothelial cells, pericytes, and FPC. The interaction strength difference is color-coded. (K) Dot plot showing the expression of collagen and laminin genes in the different APC subpopulations, in endothelial cells, pericytes, and FPC. The average gene expression is color-coded, the percentage of expression is expressed by the size of the dot. (L) Dot plot of the collagen and laminin pathways in *Bbs8*^+/+^ and *Bbs8*^−/−^ cells at 7 weeks (lean state). The *p* value is color-coded, the interaction probability score is expressed by the size of the dot. Source data are available online for this figure

To test whether Hh signaling is affected in *Bbs8*^−/−^ APCs, we analyzed the ciliary localization of SMO and the expression of Hh target genes. When stimulating Hh signaling using SAG, the ciliary SMO localization was significantly increased in *Bbs8*^+/+^ and *Bbs8*^−/−^ APCs and P1 cells (Fig. 5C,E; Appendix Fig. S5F,G), indicating that

SMO relocalizes to the cilium in *Bbs8*^+/+^ and *Bbs8*^−/−^ APCs and P1 cells upon stimulation. However, under basal conditions, the ciliary localization of SMO was already increased in *Bbs8*^−/−^ compared to *Bbs8*^+/+^ APCs and P1 cells (Fig. 5C,E; Appendix Fig. S5F,G). As BBS proteins have been shown to regulate the retrograde

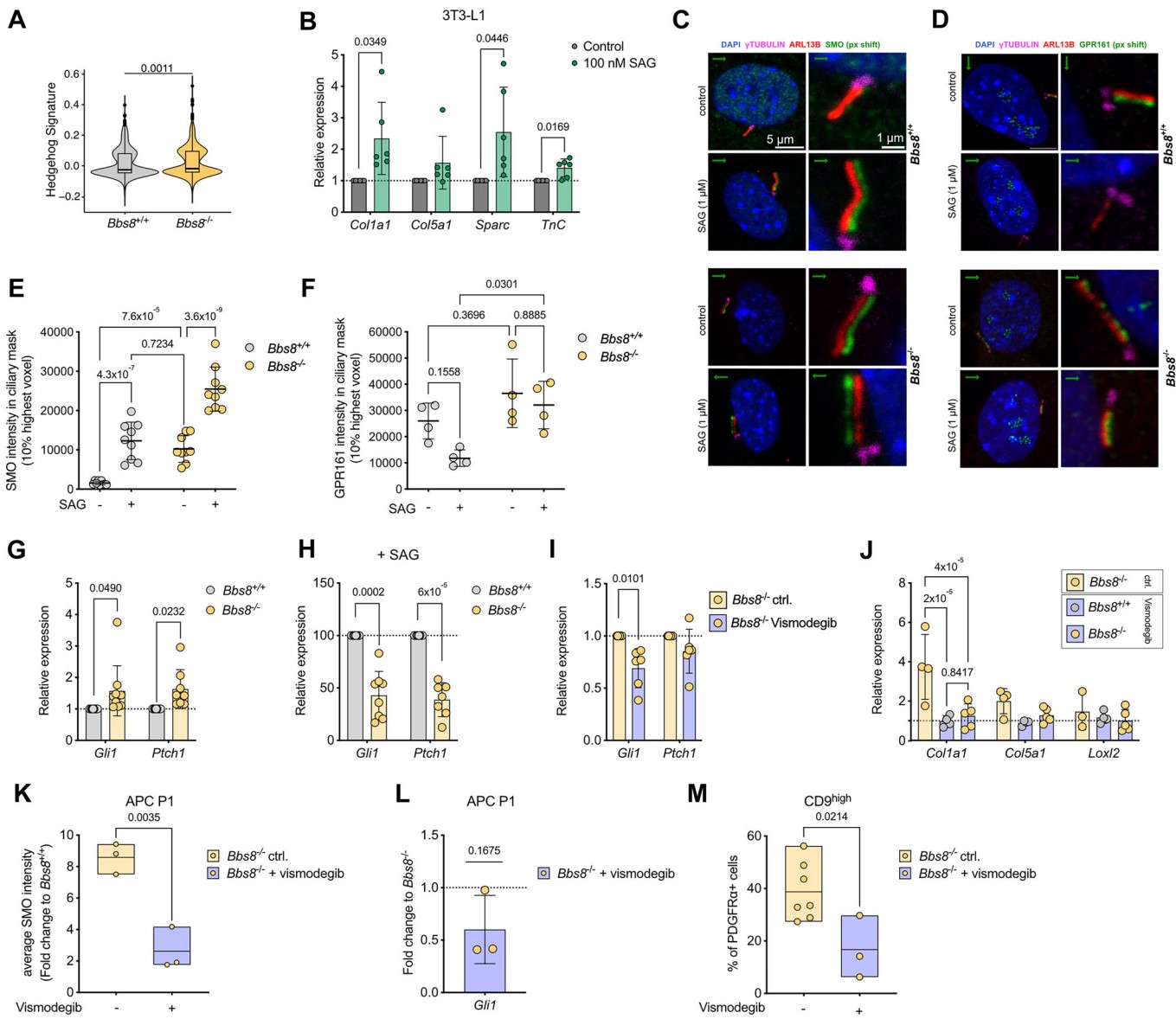

transport of GPCRs, we also analyzed the localization of the GPR161, an orphan GPCRs, which represses Hh signaling under basal conditions and is therefore removed from the cilium upon Hh stimulation (Mukhopadhyay et al, 2013). GPR161 was localized in the primary cilium of $Bbs8^{+/+}$ APCs, and upon stimulation with SAG, the ciliary localization decreased (Fig. 5D,F). However, in $Bbs8^{-/-}$ APCs, GPR161 remained in the cilium (Fig. 5D,F), demonstrating that the loss of BBS8 affects retrograde transport of GPCRs, i.e., GPR161.

To investigate whether higher basal ciliary SMO levels and higher ciliary GPR161 levels affect downstream Hh signaling, we analyzed the expression of Hh target genes. Under basal conditions, the mRNA expression of $Gli1$ and $Ptch1$ was increased in $Bbs8^{-/-}$ compared to $Bbs8^{+/+}$ APCs and P1 cells (Fig. 5G; Appendix Fig. S5H), demonstrating that the increase in ciliary SMO is promoting ectopic Hh signaling under basal conditions. Upon stimulation with SAG, the response was, however, blunted in $Bbs8^{-/-}$ compared to $Bbs8^{+/+}$ APCs and P1 cells (Fig. 5H; Appendix Fig. S5I),

demonstrating that although the ciliary SMO localization can be further increased in $Bbs8^{-/-}$ APCs upon SAG stimulation, this is not sufficient to promote Hh signaling as GPR161 remains in the cilium and dampens the response. We confirmed these results first in MEFs from $Bbs8^{+/+}$ and $Bbs8^{-/-}$ mice, which also showed an increase in the ciliary localization of SMO in $Bbs8^{-/-}$ compared to $Bbs8^{+/+}$ MEFs under basal conditions (Appendix Fig. S5J) and responded with an increase in the ciliary SMO localization in both genotypes after stimulation with SAG (Appendix Fig. S5J). Downstream, the expression of $Ptch1$ was higher in $Bbs8^{-/-}$ compared to $Bbs8^{+/+}$ MEFs under basal conditions (Appendix Fig. S5K) and the response to SAG was dampened (Appendix Fig. S5L). We aimed to rescue the effects on Hh signaling in $Bbs8^{-/-}$ MEFs by overexpressing BBS8 tagged with an HA-tag. BBS8-HA localized to the primary cilium (Appendix Fig. S5M,N). In turn, the ciliary SMO localization in $Bbs8^{-/-}$ MEFs under basal conditions was significantly reduced (Appendix Fig. S5O), and the expression of $Ptch1$ decreased (Appendix Fig. S5P).

**Figure 5. Loss of BBS8 results in ectopic Hedgehog (Hh) signaling and diminishes the Hh-dependent signaling response, promoting fibrogenic gene expression.**

(A) Violin plot with Tukey boxplot depicting the average expression of the hedgehog signature for the GSEA gene set HALLMARK_HEDGEHOG_SIGNALING calculated as the module score for all $Bbs8^{+/+}$ and downsampled $Bbs8^{-/-}$ APCs, isolated at 7–8 weeks (lean state), derived from scRNA-seq data. Violin plots represent the expression distribution of all cells. $p$ value was determined by the Wilcoxon test. (B) Relative mRNA expression of fibrogenic genes in 3T3-L1 cells treated with SAG (100 nM) for 24 h. Expression was normalized to the expression values of the vehicle control ($H_2O$). Each data point represents a biological replicate ($n = 4$). (C, D) Fluorescence confocal images of lineage-depleted iWAT SVF from $Bbs8^{+/+}$ and $Bbs8^{-/-}$ mice, isolated at 6–8 weeks (lean state), labeled against (C) Smoothened (green, SMO) or (D) GPR161 (green), and ARL13B (red, cilia), γ-Tubulin (magenta, basal body), and with DAPI (blue). Cells were treated with $H_2O$ (control) or 1 µM SAG for 24 h. Scale bars are indicated. In all images, the green channel (SMO or GPR161, respectively) was shifted by 5 px (green arrow indicates direction of shift) for better visualizing SMO or GPR161 accumulation in cilia. (E) Quantification of the ciliary SMO localization in (C). The average intensity of the 10% highest SMO pixels in the ciliary (ARL13B) mask is depicted. Each data point represents a technical replicate from $n = 3$ mice (>30 cilia per $n$, 7 weeks (lean state)) in total. All data were shown as mean ± SD, $p$ values have been determined using a two-way ANOVA with repeated measurements. (F) Quantification of the ciliary GPR161 localization in (D). The average intensity of the 10% highest GPR161 pixels in the ciliary (ARL13B) mask is depicted. Each data point represents one animal ($n = 3$ mice in total, >8 per $n$, 5–7 weeks (lean state)). All data were shown as mean ± SD, $p$ values have been determined using a two-way ANOVA with repeated measurements. (G) Relative mRNA expression of $Gli1$ and $Ptch1$ in lineage-depleted SVF from $Bbs8^{+/+}$ and $Bbs8^{-/-}$ mice, isolated at 5–8 weeks (lean state), assessed by qRT-PCR. The expression was normalized to $Bbs8^{+/+}$ expression values. All data were shown as mean ± SD, $p$ values have been determined using a one-sample $t$-test. (H) Relative mRNA expression of $Gli1$ and $Ptch1$ in lineage-depleted SVF from $Bbs8^{+/+}$ and $Bbs8^{-/-}$ mice, isolated at 5–8 weeks (lean state), treated with SAG (1 µM) for 24 h. Expression was normalized to $Bbs8^{+/+}$ expression values. All data were shown as mean ± SD, $p$ values have been determined using a one-sample $t$-test. (I) Relative mRNA expression of $Gli1$ and $Ptch1$ in lineage-depleted SVF from $Bbs8^{-/-}$ mice, isolated at 7 weeks (lean state), treated with vismodegib (2 µM) for 48 h. Expression was normalized to the expression values of the vehicle control (DMSO). All data were shown as mean ± SD, $p$ values have been determined using a one-sample $t$-test. (J) Relative mRNA expression of fibrogenic genes in lineage-depleted SVF from $Bbs8^{+/+}$ and $Bbs8^{-/-}$ mice, isolated at 7 weeks (lean state). Expression was normalized to $Bbs8^{+/+}$ expression values. All data were shown as mean ± SD, $p$ values have been determined using a two-way ANOVA with repeated measurements (mixed models). (K) Quantification of the ciliary SMO localization in P1 cells from untreated $Bbs8^{-/-}$ animals or following in vivo vismodegib injections. The average SMO intensity of each cilium was normalized to control-treated $Bbs8^{+/+}$ intensity values. Each data point represents a biological replicate ($n = 3$; >4 cilia per $n$), data were shown as floating bars (min to max) with a line depicting the mean, $p$ value has been determined using an unpaired Student's $t$-test. (L) Relative mRNA expression of $Gli1$ in $Bbs8^{-/-}$ P1 cells isolated after in vivo vismodegib injections. Relative expression was calculated in relation to control-treated $Bbs8^{+/+}$ expression, and then the fold change to untreated $Bbs8^{-/-}$ P1 cells $Gli1$ expression values was calculated. $P$ value has been determined using a one-sample $t$-test. Each data point represents one animal ($n = 3$). (M) Quantification of CD9$^{high}$ cell surface expression on PDGFRα$^+$ cells from $Bbs8^{-/-}$ mice, untreated or following in vivo vismodegib injections. Data were shown as floating bars (min to max) with a line depicting the mean, $p$ values have been determined using an unpaired Student's $t$-test ($n = 3-7$). Source data are available online for this figure

Next, we tested whether the phenotype observed in $Bbs8^{-/-}$ APCs is unique to loss of BBS8 or whether it is also recapitulated in other models of primary cilia dysfunction. To this end, we first analyzed human iPSCs carrying a homozygous missense mutation (M390R) in the $BBS1$ gene, the most common disease-causing mutation for BBS (Davis et al, 2007), to determine whether the loss of other BBSome components also affects SMO localization. Indeed, we observed a similar phenotype: the ciliary SMO localization was significantly increased at the basal state in $BBS1^{M390R/M390R}$ hiPSCs compared to $BBS1^{+/+}$ cells (Appendix Fig. S6A,B), although to a lesser extent compared to $Bbs8^{-/-}$ APCs.

We also analyzed a model of constitutive Hh signaling based on the loss of $Ptch1$, which represses ciliary SMO localization under basal conditions. To this end, we generated $Pdgfra^{+/creERT2} Ptch1^{fl/fl}$ mice, which allows to delete $Ptch1$ in Pdgfrα-expressing cells using tamoxifen treatment. We isolated APCs from $Pdgfra^{+/creERT2} Ptch1^{fl/fl}$ mice and treated the cells with tamoxifen. Deletion of $Ptch1$ increased the ciliary SMO localization and, consequently, increased expression of the Hh target gene $Gli1$, whereas the response to Hh stimulation was blunted (Appendix Fig. S6C–E). In muscle tissue, FAPs (fibro/adipogenic progenitors) lacking $Ptch1$ display ectopic Hh activation, which pushes the cells into a fibrotic fate (Norris et al, 2023), resembling the phenotype that we have characterized in the adipose tissue.

To assess whether complete loss of the primary cilium induces a fate change in APCs, we generated mice that lack primary cilia in Pdgfrα-expressing cells ($Pdgfra^{+/cre} Ift20^{fl/fl}$). Since these mice die within a few weeks after birth, we isolated the gonadal white adipose tissue (gWAT), which fully differentiates postnatally, at P4, when it only contains Pdgfrα$^+$ APCs and not yet mature adipocytes. This allows to investigate the changes in cell fate in a synchronized manner in a whole tissue ex vivo (Han et al, 2011). At this

developmental time point, the Pdgfrα$^+$ cells are ciliated and Hh responsive (Appendix Fig. S6F–I). Loss of cilia in $Pdgfra^{+/cre} Ift20^{fl/fl}$ mice led to an upregulation of $Gli1$ and $Ptch1$ in the adipose tissue (Appendix Fig. S6J), which was accompanied by an increased fibrogenic gene expression pattern (Appendix Fig. S6K).

To test whether inhibiting SMO activity is sufficient to reduce the expression of Hh target genes in $Bbs8^{-/-}$ mice, we first treated $Bbs8^{-/-}$ APCs in vitro with vismodegib, a SMO inhibitor. Indeed, inhibiting SMO in $Bbs8^{-/-}$ APCs was sufficient to significantly reduce $Gli1$ mRNA expression (Fig. 5I). Thus, we also tested whether vismodegib treatment was sufficient to reduce the fibrogenic gene expression. The expression of $Col1a$ was reduced in $Bbs8^{-/-}$ after vismodegib treatment and, therefore, no longer different to $Bbs8^{+/+}$ APCs (Fig. 5J). Similar tendencies were observed for the fibrogenic marker genes $Col5a$ and $Loxl2$ (Fig. 5J). Next, we investigated whether in vivo treatment with vismodegib after i.p. injection in lean $Bbs8^{-/-}$ mice (5–7 wks) would also reduce ectopic Hh signaling and fibrogenic gene expression. Vismodegib treatment significantly reduced the ciliary SMO localization in $Bbs8^{-/-}$ P1 cells (Fig. 5K) and decreased $Gli1$ expression in APC P1 cells (Fig. 5L). To test whether suppression of Hh signaling would also revert the fibrogenic fate change in $Bbs8^{-/-}$ mice, we analyzed the fibrogenic CD9$^{high}$ population in the gWAT of $Bbs8^{-/-}$ mice after vismodegib treatment and observed a significant reduction in the CD9$^{high}$ population compared to untreated $Bbs8^{-/-}$ animals (Fig. 5M). These results support our hypothesis that vismodegib can counteract ectopic Hh signaling and may act as an anti-fibrotic agent, as also suggested by others (Philips et al, 2011; Pratap et al, 2012).

Thus, we demonstrate that loss of BBS8 affects the ciliary localization of Hh signaling components, i.e., it promotes ciliary SMO accumulation under basal conditions and retains GPR161 in

the cilium upon stimulation of the pathway. This evokes ectopic Hh signaling under basal conditions, while the response to stimulation of Hh signaling is dampened. This phenotype can be reversed in vitro and in vivo by vismodegib treatment, rescuing ectopic Hh signaling and the fibrogenic gene expression.

In summary, our results demonstrate that BBS8 regulates canonical Hh signaling in APCs of WAT to control APC cell fate.

## Discussion

Tightly orchestrated commitment and differentiation of progenitor cells in the adipose tissue is key to maintain tissue homeostasis and function. Our study revealed that BBSome (BBS8)-dependent primary cilia signaling maintains the identity of APCs, specifically of the stem cell-like P1 subpopulation. In turn, primary cilia dysfunction upon loss of *Bbs8* results in ectopic Hh signaling, inducing a fibrogenic cell fate change that remodels the WAT. This primary cilia-dependent remodeling occurs in the lean stage prior to the onset of obesity. Our study adds a new chapter to how APC fate and function in WAT is maintained and regulated, highlighting the key role of primary cilia as a central regulator of tissue homeostasis.

BBS proteins control ciliary trafficking of GPCRs (Nachury, 2018; Wingfield et al, 2018), and SMO is one of the ciliary GPCRs whose ciliary trafficking is affected by loss of BBS proteins (Hey et al, 2021; Seo et al, 2011; Zhang et al, 2013). However, it remained largely unknown whether and how loss of a BBS protein alters ciliary signaling and, in turn, cellular functions, in particular during cell fate determination in the adult tissue. It has been shown that BBS7, another BBSome component, also controls the ciliary localization of SMO (Zhang et al, 2013).

Previous reports have shown that ciliary Hh signaling determines the fate of mesenchymal progenitor cells in skeletal muscle, the so-called fibro/adipogenic progenitors (FAPs) (Kopinke et al, 2021; Kopinke et al, 2017). FAPs promote differentiation of muscle stem cells and, thereby, muscle regeneration upon acute muscle injury. However, during chronic diseases, FAPs differentiate into adipocytes and produce fibrotic scar tissue. Ciliary Hh signaling in FAPs is maintained at a low level, but is strongly induced upon muscle injury, evoking a pro-fibrotic and anti-adipogenic response (Kopinke et al, 2021; Kopinke et al, 2017). In accordance with this finding, ectopic Hh signaling in FAPs upon loss of PTCH1 has been shown to evoke a pro-fibrotic and anti-adipogenic response (Norris et al, 2023). Our results are in accordance with these findings and broaden the concept of how Hh signaling controls cell fate determination in the adult tissue. Our results indicate that ectopic activation of Hh signaling in *Bbs8*$^{-/-}$ WAT APCs induces a fibrogenic switch, reducing the pool of adipocyte progenitors and, in turn, the pool of cells that are key for tissue plasticity. Strikingly, these changes already occur in the lean state without obesity development. Thereby, we link BBS upon loss of BBS8 and cell fate regulation by Hh signaling. Based on the results of FAPs in skeletal muscle and our results from APCs in WAT, this could be a general mechanism how BBSome-dependent primary cilia signaling controls cell fate determination of mesenchymal progenitor cells. Whereas Hh signaling controls

FAP fate and function in a non-cell-autonomous manner (Kopinke et al, 2017), loss of a BBSome component evokes a cell-autonomous action on APC fate and function. Thus, molecularly, BBS seems to mimic tissue injury, resulting in ectopic Hh signaling in the cilium and, in turn, a switch in cell fate. Of note, BBS proteins have also been proposed to play non-ciliary functions. For example, BBS proteins have been identified in the nucleus (Ewerling et al, 2023), where they seem to regulate gene expression (Gascue et al, 2012; Horwitz and Birk, 2021; Scott et al, 2017). Whether non-ciliary functions also play a role here, needs to be investigated in future studies.

ECM remodeling in the adipose tissue is not per se pathological. In fact, ECM remodeling seems to be crucial for differentiation of a mesenchymal cell into an adipocyte, which undergoes dramatic morphological and cell size changes when storing triglycerides in lipid droplets (Jääskelainen et al, 2023). ECM remodeling occurs in the lean state upon feeding and is accompanied by an induction of lipogenesis (Toyoda et al, 2022). However, upon loss of BBS8, in the lean state, ectopic Hh signaling induces the fibrogenic switch, whereby the progenitor pool is reduced. Furthermore, also the committed P2 APC subpopulation seems to be affected by loss of BBS8. Here, the ECM remodeling also plays a role, whereby the interaction of a specific P2 subcluster, P2_1, with the ECs is increased. How this remodels WAT plasticity, needs to be further investigated.

The change in differentiation trajectory and remodeling of ECM in WAT of *Bbs8*$^{-/-}$ mice in the lean state is particularly interesting when considering that pathological ECM remodeling and fibrosis is a key driver of metabolic syndrome during obesity (Sun et al, 2023). *Bbs8*$^{-/-}$ mice develop obesity due to hyperphagia and WAT remodeling. The tissue displays hypertrophic expansion, which is underlined by our in vitro lipidomics analysis. Whether ciliopathy-induced, syndromic obesity is metabolically distinct from non-syndromic obesity needs to be investigated in future studies. Mice that lack the BBS chaperonin-complex protein BBS12 also develop obesity but expand the adipose tissue by hyperplasia and retain normal glucose tolerance and insulin sensitivity (Horwitz and Birk, 2023; Marion et al, 2012). Whether this phenotype is due to primary cilia dysfunction in APCs or other cell types, is not known. Also, studies in BBS patients suggest that the metabolic outcome in BBS patients might be distinct from patients with non-syndromic obesity (Picon-Galindo et al, 2022).

Targeting Hh signaling is a major aim in cancer research as oncogenic Hh pathway activation occurs in many tumors. Vismodegib as a SMO inhibitor has been used to treat basal cell carcinoma (Meiss et al, 2018). Here, we show that a low-grade stimulation of Hh signaling with SAG induced a fibrogenic switch in APCs, and inhibiting SMO in *Bbs8*$^{-/-}$ APCs using vismodegib reduced downstream Hh signaling and, in turn, the fibrogenic gene expression pattern. Thereby, our results underline that targeting Hh signaling should not only be considered for cancer treatment but might also be suitable to remodel early stages of fibrosis during tissue remodeling in obesity. Thereby, our study advances the understanding of how cell fate and tissue plasticity in WAT are regulated by ciliary signaling and reveal that targeting ciliary signaling, in particular Hh signaling, might be a novel target to treat obesity-associated pathologies like tissue fibrosis.

# Methods

### Reagents and tools table

| Reagent/resource | Reference or source | Identifier or catalog number |
|---|---|---|
| **Experimental models** | | |
| *Bbs8*⁻/⁻ mice (*M. musculus*) | Tadenev et al, (2011) | N/A |
| *Bbs8*⁻/⁻ (KOMP) mice (*M. musculus*) | This study | Stock Nr. 050244-UCD (MMRRC) |
| *Pdgfra*-Cre mice (*M. musculus*) | Krueger et al, (2014) | JAX 013148 |
| *Ift20*fl/fl mice(*M. musculus*) | Jonassen et al (2008) | N/A |
| *Ptch1*fl/fl mice (*M. musculus*) | Gift from Heidi Hahn (University of Göttingen) | N/A |
| *Pdgfra*+/creERT2 mice (*M. musculus*) | Chung et al, (2018) | JAX #032770 |
| 3T3-L1 cell clone #27 (*M. musculus*) | Gift from Christoph Thiele (LIMES, Bonn) | N/A |
| hiPSC *BBS1*M390R/M390R (H. sapiens) | Cell Programming Core Facility, University Hospital Bonn, Germany | N/A |
| MEF (*M. musculus*) | This study | N/A |
| **Recombinant DNA** | | |
| hCMV-mTtc8--3 x HA-hPGK-Blasticidin | This study | N/A |
| **Antibodies** | | |
| Rabbit anti-SMO | Anderson Lab | N/A |
| Rabbit anti-GPR161 | Mukhopadhyay Lab | N/A |
| Mouse anti-Arl13b | Abcam | ab136648 |
| Rabbit anti-Arl13b | Proteintech | 17711-1-AP |
| Mouse anti-Tubulin, gamma | Sigma | T6793 |
| Goat anti-FABP4 | R&D Systems | AF1443 |
| Goat anti-GLI1 | R&D Systems | AF3455-SP |
| Goat anti-GLI3 | R&D Systems | AF3690-SP |
| Rat anti-CD9 | Thermo Fisher | 12-0091-81 |
| Rabbit anti-Fibronectin | Sigma | F3648 |
| Goat anti-PDGFRα | R&D Systems | AF1062-SP |
| Mouse anti-Tubulin, beta | Sigma | T4026 |
| Streptavidin BV785 | BioLegend | 405249 |
| Rat anti-CD31 Biotin | BioLegend | 102503 |
| Rat anti-CD31 BV785 | BioLegend | 102435 |
| Rat anti-CD45 Biotin | BioLegend | 103103 |
| Rat anti-CD45 AF700 | BioLegend | 147716 |
| Mouse anti-TER119 Biotin | BioLegend | 116203 |
| Rat anti-TER119 AF700 | BioLegend | 116220 |

| Reagent/resource | Reference or source | Identifier or catalog number |
|---|---|---|
| Rat anti-CD26 (DPP4) PE/Cy7 | BioLegend | 137809 |
| Rat anti-CD9 PerCP/Cyanine5.5 | BioLegend | 124818 |
| Rat anti-CD140a (Pdgfra) BV421 | BioLegend | 135923 |
| Armenian Hamster anti-CD55 APC | BioLegend | 122513 |
| Armenian Hamster anti-CD29 PerCP-efluor710 | eBioscience (Thermo) | 46-0291 |
| Rabbit anti-CD142 PE | Sino Biological | 50413-R001 |
| Rat anti-SCA1 BV510 | BioLegend | 108129 |
| Rat anti-CD54 APC-Fire 750 | BioLegend | 116125 |
| Rat anti-VAP1 | Abcam | ab81673 |
| Rat anti anti-CD54 Biotin | BioLegend | 116103 |
| Goat anti-mouse IgG1 (y1) Alexa Fluor 488 | Thermo Fisher | #A21121 |
| Donkey anti-rat Alexa Fluor 488 | Dianova | 712-545-153 |
| Donkey anti-rabbit Cy3 | Dianova | 711-165-152 |
| Donkey anti-rat Alexa Fluor 647 | Dianova | 712-605-153 |
| Donkey anti-goat Alexa Fluor 647 | Life Technologies | A21447 |
| Goat anti-mouse IgG2a (y2a) Alexa Fluor 647 | Thermo Fisher | #A21241 |
| IRDye 680RD Mouse IgG (H + L) | LICORBio | 926-68070 |
| IRDye 800RD Goat IgG (H + L) | LICORBio | 926-32214 |
| Rat IgG (H + L), pre-absorbed Alexa Fluor 588 | Abcam | ab150155 |
| Goat IgG (H + L) Alexa Fluor 594 | Abcam | ab150136 |
| Rabbit IgG (H + L) Alexa Fluor 488 | Abcam | ab150065 |
| **Oligonucleotides and other sequence-based reagents** | | |
| qPCR Primers | This study | Appendix Table S1 |
| SMART PCR Primer | AAGCAGTGGTATCAACGCAGAGT | N/A |
| **Chemicals, Enzymes and other reagents** | | |
| Collagenase II | Merck | C6885 |
| DNAse I | PanReac AppliChem | A3778 |
| Red blood cell lysis buffer | BioLegend | 420301 |
| Trypsin-EDTA (0.05%) | Thermo Fisher | 25200072 |
| EDTA | Thermo Fisher | 15575020 |
| DMEM/F-12, GlutaMAX™ | Gibco | 10565018 |

| Reagent/resource | Reference or source | Identifier or catalog number |
|---|---|---|
| DMEM/GlutaMAX™ | Gibco | 31966047 |
| StemMACS iPS Brew XF Medium | Miltenyi Biotec | 130-104-368 |
| RPMI1640 | Gibco | 11875093 |
| Penicillin Strep | Gibco | 15140122 |
| Recombinant human Vitronectin | Thermo Fisher | A14700 |
| Hoechst 33258 | Thermo Fisher | H3569 |
| Paraformaldehyde | Alfa Aesa | 12789044 |
| DAPI | Invitrogen | 10184322 |
| LD540 | Spandl et al, (2009) | N/A |
| Biotin | Sigma-Aldrich | B4510 |
| D-pantothenate | Sigma-Aldrich | P5155 |
| Insulin | Sigma-Aldrich | I9278 |
| Dexamethasone | Sigma-Aldrich | D1756 |
| 3-Isobutyl-1-methylxanthine (IBMX) | Sigma-Aldrich | I5879 |
| Rosiglitazone | Sigma-Aldrich | R2408 |
| TUG-891 | Tocris | #4601 |
| Smoothend Agonist (SAG) | Sigma-Aldrich | SML1314 |
| Vismodegib | LC Laboratories | V-4050 |
| ChemiBLOCKER | Merck | 2170 |
| Triton X-100 | Sigma-Aldrich | X-100 |
| FA 11:0;Y | TCI Deutschland GmbH | N/A |
| FA 18:2;Y | multistep synthesis done by Dr. Christoph Thiele | N/A |
| 13C6-FA 16:0;Y | multistep synthesis done by Dr. Christoph Thiele | N/A |
| Streptavidin MicroBeads | Miltenyi Biotec | 130-048-101 |
| mPIC | Sigma-Aldrich | P8340 |
| Protein Marker VI | AppliChem | A8889 |
| Intercept™ Blocking Buffer | LI-COR Biosciences | 927-70001 |
| Tween® 20 | Sigma-Aldrich | P9416 |
| TRI Reagent® | Sigma-Aldrich | T9424 |
| Mowiol® 4-88 solution | Merck | 475904 |
| Mayer's hemalum solution | Sigma-Aldrich | 1.09249 |
| Eosin Y | Carl Roth | 7089.1 |
| Resorcinol-Fuchsin | Carl Roth | **X877.1** |
| Picro-Sirius Red Solution | abcam | ab246832 |
| Barcoded mRNA capture beads | ChemGenes | MACOSKO-2011-10 |
| AMPure XP SPRI Reagent | Beckman Coulter | A63882 |

| Reagent/resource | Reference or source | Identifier or catalog number |
|---|---|---|
| **Software** | | |
| FlowJo software (Version 10) | https://www.flowjo.com/flowjo/overview | N/A |
| ImageJ | ImageJ | N/A |
| FIJI (Version 2.14) | ImageJ | N/A |
| Imaris software (Version 9.9) | Oxford Instruments | N/A |
| Zen Blue (Version 3.0 or 3.1) | Zeiss | N/A |
| nf-core RNA-seq (v1.4.2) | Ewels et al, (2020) | N/A |
| STAR | Dobin et al, (2013) | N/A |
| bcl2fastq2 (v2.20) | Illumina | N/A |
| Seurat analysis pipeline (v4.1.1) | https://satijalab.org/seurat/ | N/A |
| monocle3 package (Version 1.0.0) | https://cole-trapnell-lab.github.io/monocle3/ | N/A |
| CellChat (Versio 1.6.1) | Jin et al, 2021 | N/A |
| GraphPad Prism (Version 10) | https://www.graphpad.com | N/A |
| **Other** | | |
| 4D-Nucleofector X Unit | Lonza | AAF-1003X |
| Amaxa™ P4 Primary Cell 4D-Nucleofector™ X Kit | Lonza | V4XP-4024 |
| LIVE/DEAD™ Fixable Near-IR Dead Cell Stain Kit | Thermo Fisher | L34975 |
| Pierce™ BCA Protein Assay Kit | Thermo Fisher | 23227 |
| NuPAGE™ 4–12% Bis-Tris Gels | Invitrogen | NP0321PK2 |
| High-Capacity cDNA Reverse Transcription Kit | Thermo Fisher | 4368813 |
| FastStart Universal SYBR Green Mastermix | Roche® Life Science | 4913914001 |
| PureLink® RNA Mini Kit | Thermo Fisher | |
| miRNeasy Micro Kit | Qiagen | 217084 |
| iScript TM cDNA synthesis Kit | Bio-Rad | 1708840 |
| LD columns | Miltenyi Biotec | 130-042-901 |
| MS columns | Miltenyi Biotec | 130-042-201 |
| Tapestation 4200 system | Agilent | |
| Smart-seq2 | | |
| NovaSeq6000 system | Illumina | |
| Nextera XT kit | Illumina | |
| KAPA HiFi Hotstart Readymix PCR Kit | Kapa Biosystems | KK2602 |

| Reagent/resource | Reference or source | Identifier or catalog number |
|---|---|---|
| Nextera XT DNA Library Preparation Kit | Illumina | |
| Qubit™ 1X dsDNA High Sensitivity (HS) and Broad Range (BR) Assay Kits | Thermo Fisher | Q33230 |
| Attune NxT Flow Cytometer | Thermo Fisher | |
| Fluoview FV1000 confocal microscope | Olympus | |
| BD FACSAria III | BD Biosciences | |
| ID7000 Spectral Cell Analyzer 5 L | Sony Biotechnology | |
| Leica SP8 AOTF | Leica | |
| Leica Stellaris 8 | Leica | |
| Zeiss Laser Scanning Microscope LSM 900 | Zeiss | |
| Nanodrop | NanoDrop Products | |
| Odyssey Imaging System | LI-COR Biosciences | |
| PCR cycler | SensoQuest | |
| QuantStudio™ 6 Pro Real-Time PCR | Thermo Fisher | |
| Thermocycler (Labcycler) | SensoQuest | |
| Vi-CELL BLU | Beckmann Coulter | |
| Zeiss AXIO SCAN.Z1 | Zeiss | |
| Zeiss CELLDISCOVERER 7 | Zeiss | |

## Animal Studies

All animal experiments were performed in agreement with the German law of animal protection and local institutional animal care committees (Landesamt für Natur, Umwelt und Verbraucherschutz, LANUV). Mice were kept in individually ventilated cages in the mouse facility of University Hospital Bonn (Haus für Experimentelle Therapie [HET], Universitätsklinikum, Bonn). Mice were raised under a normal circadian light/dark cycle of each 12 h, and animals were given water and a complete diet (ssniff Spezialdiäten) ad libitum (LANUV Az 81-02.04.2019.A170). Mice were sacrificed using cervical dislocation. Generation and breeding of $Bbs8^{-/-}$ mice and genotyping was previously described (Tadenev et al, 2011) and approved (LANUV Az 81-02.04.2019.A1428). Tissues and samples were collected from age-matched control ($Bbs8^{+/+}$) and mutant ($Bbs8^{-/-}$) littermates. The conditional mouse lines were generated based on the lines C57BL/6N-Ttc8tm1a(-KOMP)Wtsi/MbpMmucd, stock number 050244-UCD (MMRRC); B6.129S7(129S4)-Ift20tm1.1Gjp/J, and Ptch1tm1Hahn/J (kindly provided by Heidi Hahn, University of Göttingen). To generate conditional $Bbs8$ knockout mice, the line was first crossed to Flp$^e$ mice (Rodriguez et al, 2000) to generate $Bbs8^{flox/flox}$ mice. To

generate conditional or inducible knockout lines, the respective lines were crossed to $Pdgfra^{+/cre}$ (Krueger et al, 2014) mice (JAX 013148) or to $Pdgfra^{+/creERT2}$ mice (JAX #032770) (Chung et al, 2018).

Vismodegib (SelleckChem Cat. No. S1082; Batch: S108207) was dissolved and stored in DMSO. For administration, it was further dissolved in corn oil, and mice were injected with 5 mg/kg i.p. every other day for a total duration of 20 days (AZ 81-02.04.2023.A233).

## Cell lines and cell culture

The 3T3-L1 cell clone #27 was kindly provided by Christoph Thiele, LIMES Institute, University of Bonn, Germany. Cells were maintained in DMEM, supplemented with 1% GlutaMAX-I (both: Life Technologies/Life Technologies) and 10% fetal calf serum (FCS, Biochrom) at 37 °C and 5% $CO_2$. Cells had been tested for mycoplasma twice a year and were free from mycoplasma.

## Adipose tissue progenitor cell isolation

Progenitor cell isolation from the stromal vascular fraction (SVF) was performed as previously described (Sieckmann et al, 2022). Briefly, inguinal WAT was surgically removed, minced, and digested with 2 mg/mL collagenase II (Merck) and 15 kU/mL DNAse I (PanReac AppliChem) in 0.5% bovine serum albumin (BSA; Sigma) in phosphate-buffered saline (PBS) at 37 °C with agitation. The digestion was quenched by adding AT buffer (0.5% BSA in PBS). The dissociated cells were passed through a 100-μm filter (Corning) and subjected to centrifugation at 500 × g for 10 min. The resulting supernatant containing mature adipocytes was aspirated, and the pellet, consisting of the stromal vascular fraction, was resuspended in red blood cell lysis buffer (BioLegend) for 2 min at RT. The reaction was stopped by adding AT buffer and centrifugation at 500 × g for 10 min. Isolated cells were then further subjected to antibody staining for MACS or FACS processing. Adipocyte progenitors were only isolated at the lean state (7 weeks) for all mouse models.

## Isolation and Immortalization of mouse embryonic fibroblasts

For the isolation of mouse embryonic fibroblasts (MEFs), timed matings were set up with one male and two females of the desired genotype. At day 13, the pregnant mouse was anesthetized using isoflurane (Piramal Healthcare), followed by a cervical dislocation. The lower abdomen was opened by an abdominal incision to extract the two uterine horns. Embryos were isolated, transferred into a 24-well plate filled with PBS, and the head and the red organs (heart and liver) were removed. The rest of the embryo was placed into a 12-well plate filled with 2 ml ice-cold 0.25% Trypsin/PBS (diluted from 2.5% Trypsin, Gibco). The embryos were chopped into small pieces and incubated overnight at 4 °C. Then, the trypsin solution was discarded, and the remaining Trypsin/tissue mixture was incubated for 30 min in a 37 °C water bath. Afterwards of medium (composition: DMEM/Glutamax, 10% FCS, 1% sodium pyruvate (100x), 1% Pen Strep), was added, and the cell suspension was pipetted vigorously up and down to break up the digested tissue into a single cell suspension. After 1 min to allow sedimentation of the remaining tissue, the cell suspension was

transferred into a new tube. This step was repeated, and afterwards, the cell suspension was filtered through a 100-µm-cell-strainer (Corning). Cells were plated and after 24 h, the medium was changed. Immortalization of MEFs was performed as described previously (Todaro and Green, 1963). Briefly, cells are split every three days and seeded with the same cell density. From passage three onwards, cells were seeded on at least two 10 cm culture dishes. After around 15 passages, cells started to regrow. Once MEFs were immortalized, frozen back-ups were made.

## Nucleofection of MEFs

For transient expression of Bbs8 (hCMV-mTtc8-3 x HA-hPGK-Blasticidin, generated and provided by Rainer Stahl, University of Bonn) in MEFs, 200,000 cells were spun down at $400 \times g$ for 4 min at RT. Cells were resuspended in 20 µl of nucleofection master mix (AmaxaTM P4 Primary Cell 4D-NucleofectorTM X Kit) (16.4 µl P4 Solution, 3.6 µl of Supplement 1, and 1 µg of plasmid DNA). Cell suspension was transferred into separate wells in the Nucleostrip and electroporated in the 4D-Nucleofector X Unit (Lonza) with the pulse code CZ 167. After electroporation, the cell suspension was incubated at RT for 10 min before the addition of 40 µl of prewarmed media. The cell suspension was seeded at a density of 20,000 cells per well of a black PhenoPlate 96-well plate (Revvity) or at 200,00 cells per well in a 24-well plate and incubated at 37 °C and 5% $CO_2$ for 48 h before fixation for immunocytochemistry or stored for RNA isolation.

## iPSC culture

The hiPSC $BBS1^{M390R/M390R}$ clone and the isogenic controls were generated by the Cell Programming Core Facility, University Hospital Bonn, Germany. Cells were cultured on recombinant human Vitronectin (rhVTN, Thermo Fisher) coated plates in StemMACS iPS Brew XF Medium (Miltenyi Biotec, 130-104-368) containing 1% Pen/ Strep (Gibco/ Life Technologies).

## Ex vivo gonadal adipose tissue explants

To investigate adipose tissue development, pre-gWAT of male mice on P4 was dissected and separated from the testis, epididymis, and vas deferens. Tissues were either fixed immediately with 4% PFA for 1 h to prepare for immunostaining. Alternatively, tissues were cultivated for 24 h in RPMI1640 medium (Gibco) containing 10% FCS and 1% Pen/ Strep (Gibco/ Life Technologies) for pharmacological stimulation and subsequently frozen for RNA extraction.

## Magnetic activated cell sorting (MACS)

MACS was performed according to the manufacturer's instructions (Miltenyi Biotec). For negative selection, suspended cells were incubated with biotinylated anti-CD45, anti-CD31, and anti-TER119 antibodies (Biolegend), followed by incubation with Streptavidin MicroBeads (Miltenyi Biotec). Samples were run through LD columns (Miltenyi Biotec) followed by three washes with MACS buffer (0.5% BSA, 2.5 mM EDTA in PBS). Unlabeled cells were collected and either cultured (referred to as SVF Lin- cells or further incubated with anti-CD54 (Biolegend) and anti-VAP1 (abcam) antibodies, followed by Streptavidin MicroBeads.

Unlabeled cells were collected using MS columns (Miltenyi Biotec). After three washes with MACS buffer, cells were eluted and used for downstream applications.

## Flow cytometry and fluorescence-assisted cell sorting (FACS)

Mouse antibodies for flow cytometry were purchased from Biolegend, Thermo Fisher Scientific, Miltenyi Biotec, and BD Biosciences (Reagents and Tools Table).

Isolated SVF cells were stained with primary antibodies for 30 min on ice, washed, and stained with the secondary antibody for 15 min on ice. Hoechst 33258 (Thermo Fisher) or LIVE/DEAD™ Fixable Near-IR Dead Cell Stain Kit (Thermo Fisher) were used to exclude dead cells.

Data were acquired on a Sony ID700 spectral cytometer (BD Biosciences) or the Attune NxT Flow Cytometer (Thermo Fisher Scientific) and analyzed with the FlowJo software (Tree Star). To generate the UMAP, the SCA1$^+$ gate was downsampled, and samples from all experiments were concatenated into one file to be analyzed by a custom-made R script kindly provided by the group of Elvira Mass. FACS was performed on a BD FACSAria III cell sorter.

## Immunocytochemistry

Cells were fixed with 4% paraformaldehyde (PFA, 16% wt/vol ag. Soln., methanol free, Alfa Aesa) for 10 min and subsequently washed with PBS before blocking with CT (0.5% Triton X-100 (Sigma-Aldrich), 5% ChemiBLOCKER (Merck Millipore) in 0.1 M NaP, pH 7.0) for 30 min at room temperature. Primary and secondary antibodies were diluted in CT and incubated for 60 min at room temperature. DAPI (4′,6-diamidino-2-phenylindole, dihydrochloride, 1:10,000, Invitrogen) was used as a DNA counterstain together with the secondary antibody. For staining of lipid droplets, cells were incubated with the lipophilic dye LD540 (1:10,000) (Spandl et al, 2009) for 15 min and washed again with PBS. All antibodies are shown in the Reagents and Tools Table.

## SVF culture and in vitro assays

Cells were maintained in maintenance medium containing DMEM/ F-12 (1:1), supplemented with 1% GlutaMAX-I, 1% penicillin-streptomycin (all Life Technologies/Life Technologies), 10% FCS (Biochrom), 33 mM biotin (Sigma), and 17 mM D-pantothenate (Sigma) at 37 °C with 5% $CO_2$ in a cell culture incubator.

For the adipogenesis assay, isolated APCs were seeded on CellCarrier Ultra 96- or 384- well plates. When cells reached confluency, adipogenesis was induced by switching to their respective induction cocktail: (a) Full Induction (FI) containing 5 µg/ml insulin, 1 µM Dexamethasone, 100 µM IBMX, and 1 µM rosiglitazone (Sigma); (b) Minimal Induction (MI) (Hilgendorf et al, 2019) containing 0.4 µg/ml insulin (Sigma), 0.1 µM Dexamethasone (Sigma), and 20 µM 3-isobutyl-1-methylxanthine (IBMX; Sigma); (c) Insulin only condition containing 0.4 µg/ml insulin. TUG-891 (Tocris, #4601) was added at a concentration of 100 µM.

The medium was exchanged to freshly prepared maintenance medium, containing 1 µg/ml insulin on days 3 and 5. Additionally, as a negative control, undifferentiated cells without induction medium were kept in maintenance medium.

To induce Hh signaling in APCs or MEF cells, APCs were starved for 12 or 24 h, respectively. Cells were then stimulated with 1 μM Smoothend Agonist (SAG, Sigma) or $H_2O$ (vehicle) for 24 h. Cells were either harvested for RNA isolation or fixed for immunocytochemistry.

To activate Hh signaling in 3T3-L1 cells, the cells were seeded and grown to full confluency over 48 h. Then, cells were stimulated with 100 nM SAG (Sigma) for 24 h and harvested for RNA isolation.

To investigate Hh signaling in hiPSCs, $BBS1^{M390R/M390R}$ and $BBS1^{+/+}$ cells were seeded on rhVTN-coated μ-slide eight well (Ibidi, 80826) and stimulated with 1 μM Smoothend Agonist for 24 h before fixation for immunocytochemistry.

To block Hh signaling in APCs, the cells were incubated with 2 μM vismodegib (LC Laboratories) or DMSO (vehicle) for 48 h. Cells were then harvested for RNA isolation.

## Pulse chase analysis of fatty acid incorporation

Pulse chase analysis using alkyne-labeled fatty acids followed the protocol described in (Wunderling et al, 2023). Isolated APCs were seeded on 48-well plates. When cells reached confluency, adipogenesis was induced by switching to a full induction cocktail for three days and then switching to 1 μg/ml insulin for one more day. Subsequently, cells were fed with growth medium containing 50 μM of each alkyne-fatty acid: FA 11:0;Y (TCI Deutschland GmbH), FA 18:2;Y (multistep synthesis done by Dr. Christoph Thiele) and 13C6-FA 16:0;Y (multistep synthesis done by Dr. Christoph Thiele), for 1 h. After 1 h media were removed, cells were washed once with medium and fresh medium was added for the indicated chase times. After the chase, the vmedia were removed, and cells were washed with PBS and processed for extraction and analysis.

## Lipid extraction and click reaction

All solvents were HPLC grade or LC-MS grade and were purchased from VWR International GmbH (Darmstadt, Germany) and Merck KGaA (Darmstadt, Germany).

To each well, 500 μl of extraction mix (490 μl $MeOH/CHCl_3$ 5/1, 10 μl internal standard mix containing alkyne-labeled standard lipids (Wunderling et al, 2023) and a non-alkyne internal standard for TG (TG 50:1[D4]) as indicated above were added and the entire plate was sonicated for 1 min in a bath sonicator. The extract, including the cell remnants, were collected into 1.5 ml original Eppendorf tubes and centrifuged at $20,000 \times g$ for 5 min to pellet protein. The supernatants were transferred into fresh tubes. After the addition of 400 μl $CHCl_3$ and 600 μl 1% AcOH in water, samples were shaken for 30 s and centrifuged for 5 min at $20,000 \times g$. The upper phase was removed, the lower phase transferred into a fresh tube and dried for 20 min at 45 °C in a speed-vak. $CHCl_3$ (8 μl) was added, and the tubes were briefly vortexed. To each tube, 40 μl Click mix were added (prepared by mixing 10 μl of 100 mM C175-7x in 50% MeOH (stored as aliquots at −80 °C) with 200 μl 5 mM Cu(I) $AcCN_4BF_4$ in AcCN and 800 μl ethanol), followed by sonication for 5 min and incubation at 40 °C for 16 h.

To the samples clicked with C175-7x, 100 μl $CHCl_3$ per sample was added, and multiplex samples were pooled. To the pool, 600 μl water was added, and the pools were briefly shaken and centrifuged

for 2 min at $20,000 \times g$. The upper phase was removed, and the lower phase dried in a speed-vak as above. Spray buffer (200–1000 μl) was added, the tubes sonicated for 5 min, and the dissolved lipids analyzed by MS.

## Mass spectrometry

The dissolved lipids were analyzed on a Thermo Q Exactive Plus spectrometer equipped with a standard heated electrospray ionization source and established data analysis procedure (Wunderling et al, 2023). Raw files were converted to. mzml files using MSconvert and analyzed using LipidXplorer for lipid species that incorporated the alkyne-fatty acid.

## Protein preparation and Western blot analysis

All steps were performed at 4 °C and in the presence of a mammalian protease inhibitor cocktail (mPIC, Sigma-Aldrich, #P8340-1 ML). Cells were washed with PBS and lysed in the plates using RIPA buffer (20 mM Tris-HCl pH 7.4, 150 mM NaCl, 1 mM EDTA, 1% Triton X-100, 10% glycerol, 0.1% SDS, and 0.5% deoxycholate). The protein concentration was determined with a Pierce™ BCA Protein Assay Kit (Thermo Fisher Scientific, #23227) according to the manufacturer's instructions.

Samples were suspended in SDS sample buffer (200 mM Tris/HCl, pH 6.8, 8% SDS, 4% ß-mercapto-ethanol, 50% Glycerin, and 0.04% bromphenol blue). Discontinuous gels were casted with a 10% separation gel (1.5 mL Tris-HCl (pH 8.8), 60 μL 10% (w/v) SDS, 40 μL (w/v) APS, 6 μL TEMED, 2 mL 30% (v/v) acrylamide/bis-acrylamide (37.5:1) solution, and 2.42 mL H2O), and a 5% stacking gel (0.5 mL Tris-HCl (pH 6.8), 20 μL 10% (w/v) SDS, 40 μL (w/v) APS, 6 μL TEMED, 340 μL 30% (v/v) acrylamide/bis-acrylamide(37.5:1) solution, and 1.12 mL H2O). Alternatively, NuPAGE™ 4–12% Bis-Tris Gels (Invitrogen) were used. Samples and Protein Marker VI (AppliChem) were loaded onto the gel and proteins were separated in an electric field of 90–120 V for varying time spans in SDS Running Buffer (25 mM Tris-base, 0.192 M Glycine, 0.1% SDS). SDS-PAGE separated proteins were transferred to a methanol-activated PVDF membrane by wet transfer. Wet transfer was performed in a blotting chamber with ice-cold transfer buffer (48 mM Tris-base, 39 mM glycine, 0.037% SDS and 20% methanol) for 2–3 h at 120 mA per membrane. Before and after blotting, the membrane was briefly activated in methanol. Membranes were blocked with Intercept™ Blocking Buffer (in PBS, LI-COR Biosciences, #927-70001) supplemented with Tween® 20 (Sigma, #P9416). Membranes were incubated overnight at 4 °C with 20 ml of the respective primary antibody solution (Reagents and Tools Table). Membranes were washed three times for 10 min with PBS-T and incubated for 1 h at room temperature with respective secondary antibodies (Reagents and Tools Table). Wash steps were repeated. Images were taken on the LI-COR Odyssey imaging system (for IRDye detection). Protein expression was quantified using ImageJ (version 2.14).

## Quantitative RT-PCR

To isolate mRNA, primary adipocytes and adipose tissues were lysed within a monophasic solution of phenol and guanidine isothiocyanate reagent (TRI Reagent®, Sigma-Aldrich #T9424)

following silica-based RNA spin column enrichment and eluted with RNAse/DNAse-free water. RNA concentrations were measured using a spectrophotometer (Nanodrop, Thermo Fisher), and RNA were used for reverse transcription (High-Capacity cDNA Reverse Transcription Kit, Thermo Fisher #4368813). Abundance of genes of interest was quantified using the SYBR Green-based quantification method (FastStart Universal SYBR Green Mastermix, Roche® Life Science #4913914001) and mRNA abundance was calculated using relative quantification methods (Pfaffl Method; Pfaffl et al, 2001).

Transcript levels of mRNAs were normalized to glycerinaldehyde-3-phosphate-dehydrogenase (*Gapdh*) and tatabox binding protein (*Tatabp*) expressions. Primer sequences are provided in Appendix Table S1.

## Histology

Histological analysis was performed as previously described (Sieckmann et al, 2022). Briefly, WAT was fixed and further processed using the automated Epredia Excelsior AS Tissue Processor (Thermo Fisher Scientific). Tissues were dehydrated and cleared in a clearing agent and xylene (AppliChem) before incubating in molten paraffin wax (Labomedic). Tissues were cast into molds together with liquid paraffin and cooled to form a solid paraffin block with embedded tissue (Leica EG1150 H Paraffin Embedding Station and Leica EG1150 C Cold Plate). Paraffin-embedded WAT was sliced into 5-μm sections using a Thermo-Scientific HM 355S Automatic Microtome and mounted on Surgipath X-tra Microscope Slides (Leica Biosystems).

For histological analysis WAT sections were stained with Hematoxylin and Eosin (H&E) or Sirius Red and Elastika van Gieson (EvG) and using the Leica ST5020 Multistainer combined with Leica CV5030 Fully Automated Glass Coverslipper. Deparaffinization of paraffin-embedded WAT slices was performed and three subsequent steps in xylene before incubation in a graded alcohol series (100–70% ethanol) to rehydrate the tissue sections and ending with a final rinsing step in sterile distilled water (dH2O). Next, tissue slices were stained with Mayer's hemalum solution (Sigma-Aldrich). To counterstain with Eosin Y solution (1% in water, Roth), slides were immersed in eosin. For the Sirius Red and EvG staining slides were firstly incubated in Resorcinol-Fuchsin, following staining in haematoxylin and finally Picro-Sirius Red solution. Next, slices were washed in 30% acetic acid and shortly stained in Picric Fuchsine. After each respective staining, the slides were incubated in a graded alcohol series (70–100% ethanol) to dehydrate the tissue. Paraffin embedding, slicing, and staining were conducted by the histology facility at University Hospital Bonn.

## Whole-mount staining

For whole-mount staining, tissues were fixed with 4% PFA for 2 h at 4 °C and subsequently washed with PBS. Tissues were then cut into small pieces (~2 × 2 mm) and placed into a 24-well plate. Blocking solution (1% (w/v) BSA + 1% (v/v) Triton X-100 in PBS) was added, and samples were incubated for 2 h at RT on a rocking platform. Following fixation and blocking, primary antibodies were applied. Antibodies were diluted in blocking solution, and tissues were incubated overnight at 4 °C on a rocking platform. After

washing samples with PBS, secondary antibodies, diluted in DAPI solution, was added to each sample. Tissues were incubated overnight at 4 °C, protected from light. After incubation time, samples were washed with PBS and mounted in Elvanol, covered with a glass cover slip, and stored at 4 °C until imaged.

To prepare a 25% (w/v) Mowiol® 4-88 solution, the powder was dissolved in PBS and stirred continuously at room temperature (RT) overnight. The following day, the solution was heated to 50 °C and stirred for an additional 2–3 h to ensure complete dissolution. Afterward, 50 ml of glycerol was added, and the mixture was stirred at RT overnight. The solution was then centrifuged at 12,000×*g* for 30 min at 4 °C, and the supernatant was carefully transferred to a fresh tube. Next, 30 mg of p-phenylenediamine was gradually introduced at 4 °C while stirring continuously. Following this step, 1 M NaOH was slowly added dropwise (~750 μl) to adjust the pH to 8.0. From this point forward, the solution was kept protected from light. To achieve the desired color transition, β-mercaptoethanol (~1.5–2 ml) was added gradually until the solution shifted from brown to light yellow. Finally, the Elvanol solution was aliquoted and stored at −20 °C for later use.

## Microscopy and image analyses

Confocal z-stacks were recorded with a confocal microscope at the Microscopy Core Facility of the Medical Faculty at the University of Bonn (Leica SP8) or an Olympus Fluoview FV1000 confocal microscope at the Caesar Institute (Bonn). For quantifying fluorescence signals, z-stacks were recorded from at least two random positions per experiment and analyzed using "CiliaQ" (Hansen et al, 2021). CiliaQ was developed to fully automatically quantify the ciliary intensity levels in the different channels. In all plots, the parameter revealing the average intensity of the 10% of cilia pixels with the highest intensity is shown as the ciliary intensity level. From all values, the background intensity level was subtracted.

Fluorescence images of adipogenesis assays were taken at the CellDiscoverer 7 widefield microscope (Zeiss) using automated image acquisition. Two or four images were acquired per well, each in a z-stack (step size 4 μm, 10× magnification). Depicted images are shown as a projection of the sharpest plane, including the plane above and below. A maximum projection around the sharpest plane was generated using the ImageJ plugin ExtractSharpestPlane_JNH (https://doi.org/10.5281/zenodo.5646492) (Hansen et al, 2021). Adipogenesis was quantified using "AdipoQ" (Sieckmann et al, 2022). Briefly, custom preferences for AdipoQ Preparator and AdipoQ were used to identify and analyze adipocytes. Whole-mount imaging was performed using Zeiss laser scanning confocal microscopes (LSM 900), and images were captured using Zen Blue 3.0 or 3.1 software from Zeiss.

## Bulk RNA-sequencing and analysis

Subpopulations were sorted as described above and immediately stored in Trizol. Total RNA was extracted using the miRNeasy Micro kit (Qiagen) according to the manufacturer's protocol. RNA was quantified, and RNA integrity was determined using the HS RNA assay on a Tapestation 4200 system (Agilent). Smart-seq2 was used for the generation of non-strand-specific, full transcript sequencing libraries using standard reagents and procedures as

previously described (Picelli et al, 2014). Briefly, 250 pg of total RNA was transferred to a buffer containing 0.2% Triton X-100, protein-based RNase inhibitor, dNTPs, and oligo-dT oligonucleotides to prime the subsequent RT reaction on polyadenylated mRNA sequences. The SMART RT reaction was performed at 42 °C for 90 min using commercial SuperScript II (Invitrogen) and a TSO. A preamplification PCR of 14 cycles was performed to generate double-stranded DNA from the cDNA template. At least 200 pg of amplified cDNA were used for tagmentation reaction and subsequent PCR amplification using the Nextera XT kit (Illumina) to construct sequencing libraries. Libraries were quantified using the Qubit HS dsDNA assay, and library fragment size distribution was determined using the D1000 assay on a Tapestation 4200 system (Agilent). Samples were pooled and clustered at 1.25 nM on a NovaSeq6000 system (Illumina) to generate ~10 M single-read (75 bp) reads per sample using a NovaSeq6000 XP kit. Raw sequencing data were demultiplexed using bcl2fastq2 v2.20.

The RNA-seq 3′ data were processed with nf-core RNA-seq v1.4.2 (Ewels et al, 2020) pipeline using STAR (Dobin et al, 2013) for alignment and featureCounts for gene quantification (Liao et al, 2014). The library strandedness parameter was set to forward, and the reference was set to GRCm38. Statistical analysis was performed in the R environment (R Core Team, 2019) with the Bioconductor R-package DESeq2 (Love et al, 2014). The Benjamini–Hochberg method was used to calculate multiple testing adjusted p-values. Only genes with at least 10 read counts in at least two samples and at least 20 read counts in total across all samples were considered for analysis. Data visualization, such as volcano plots, were generated upon VST transferred data (Anders and Huber, 2010) using R-packages ggplot2 (v3. 3.3; Wickham, 2016). GO term and pathway enrichment analysis for differently expressed genes (FDR < 0.05, Fisher test) was performed using the Bioconductor packages fgsea, goseq (Subramanian et al, 2005; Young et al, 2010), and clusterProfiler (Wu et al, 2021).

## Single-cell RNA-sequencing—sample preparation

Sequencing primarily relied on the Seq-Well S³ protocol (Hughes et al, 2020), using two arrays per sample. Seq-Well arrays were prepared as described before (Gierahn et al, 2017). Each array was loaded with approximately 110,000 barcoded mRNA capture beads (ChemGenes, Cat: MACOSKO-2011-10) and with 30,000 cells. The procedure was executed as previously detailed (Hughes et al, 2020). After cell loading, cells were lysed, mRNA captured, and cDNA synthesis was performed. For the whole transcriptome amplification, beads from each array were distributed in 18–24 PCR reactions containing ~3000 beads per PCR reaction (95 °C for 3 min, four cycles of 98 °C for 20 s, 65 °C for 45 s, 63 °C for 30 s, 72 °C for 1 min, 16 cycles of 98 °C for 20 s, 67 °C for 45 s, 72 °C for 3 min and final 72 °C for 5 min) using KAPA HiFi Hotstart Readymix PCR Kit (Kapa Biosystems, Cat: KK2602) and SMART PCR Primer (AAGCAGTGGTATCAACGCAGAGT). About 6–8 PCR reactions were pooled and purified with AMPure XP SPRI Reagent (Beckman Coulter), first at 0.6x and then at a 1x volumetric ratio. For the library tagmentation and indexing, 200 pg of DNA from the purified WTA from each pool were tagmented with the Nextera XT DNA Library Preparation Kit for 8 min at 55 °C, followed by a Tn5 transposase neutralization for 5 min at RT. Finally, Illumina indices were attached to the

tagmented products (72 °C for 3 min, 98 °C for 30 s, 16 cycles of 95 °C for 10 s, 55 °C for 30 s, 72 °C for 1 min, and final 72 °C for 5 min). Library products were purified using AMPure XP SPRI Reagent, first at 0.6x and then at a 1× volumetric ratio. Final library quality was assessed using a High Sensitivity D5000 assay on a Tapestation 4200 (Agilent) and quantified using the Qubit high-sensitivity dsDNA assay (Invitrogen). Seq-Well libraries were equimolarly pooled and clustered at 1,25 nM concentration with 10% PhiX on a NovaSeq6000 system using an S2 flow cell with 100 bp v.1.5 chemistry. Sequencing was performed paired-end using a custom Drop-Seq Read 1 primer for 21 cycles, eight cycles for the i7 index, and 61 cycles for Read 2. Single-cell data were demultiplexed using bcl2fastq2 (v2.20). Fastq files from Seq-Well were loaded into a snakemake-based data pre-processing pipeline (version 0.31, available at https://github.com/Hoohm/dropSeqPipe) that relies on the Drop-seq tools provided by the McCarroll lab (Macosko, Basu et al 2015). STAR alignment within the pipeline was performed using the murine GENCODE reference genome and transcriptome mm10 release vM16 (Team 2014).

## Single-cell RNA-sequencing—data analysis

The scRNA-seq data analysis was conducted using the Seurat analysis pipeline (version 4.1.1) unless otherwise specified. To address differences in sequencing depth, the downsampleBatches function from the scuttle package (version 1.0.4) was employed using default parameter settings. Cells expressing fewer than 100 or more than 1000 genes, as well as those with more than 20 percent mitochondrial genes, were removed. The individual sequencing samples were integrated using 2000 variable features. Using the first 30 dimensions, the Louvain algorithm detected 16 cell clusters, which were annotated using known marker gene panels (see Appendix Fig. S4C). Clusters with less than 60 cells (10% percentile) were removed from the dataset.

To account for sample size differences due to varying sequencing depths, the $Bbs8^{-/-}$ dataset was downsampled according to the $Bbs8^{+/+}$ reference by grouping cells into ten bins based on their UMAP 1 coordinates. Finally, 353 cells per bin were randomly selected from the $Bbs8^{-/-}$ dataset. A differential abundance analysis of each APC and FPC subpopulation across genotypes was performed using a Fisher's exact test with p-values corrected using the FDR method. Developmental trajectories within the APC and FPC subpopulations were calculated using the monocle3 package (version 1.0.0). After re-clustering, the graph structure from the $Bbs8^{+/+}$ and downsampled $Bbs8^{-/-}$ datasets was learned using default parameters while disabling graph pruning. Pseudotimes were computed using the APC P1 subpopulation as the root.

Differences in the murine GSEA Hallmark-Hedgehog (Hh) signature across genotypes within the APC and FPC subpopulation were assessed using Seurat's AddModuleScore function. First, the Hallmark genes were filtered for expressed genes before calculating cell-wise Hh-signature scores. Those scores were grouped per genotype and compared using a Wilcoxon test within the down-sampled subset. Further characterization of the FPC population involved an overrepresentation analysis (ORA) employing the C2 MsigDB data repository, using the top 50 up- and down-regulated genes between FPCs and all other cell populations, as described (Kwon et al, 2023). The 50 top and bottom genes were identified by ranking them according to Bonferroni-corrected $p$ values calculated using Seurat's FindMarkers function. The C2 MsigDB dataset was

filtered for entries with less than 300 members to focus on gene-sets of comparable size to our input data. Finally, the pathway overrepresentation was calculated using the clusterprofiler package (version 3.18.1) with a $q$ value cutoff of 5%.

CellChat (v1.6.1) (Jin et al, 2021) was used to analyze possible cell-cell interactions in the data. To find all possible interactions, the "computeCommunProb" command was used with "type = truncatedMean" and "trim = 0" arguments. The analysis was run separately on data from $Bbs8^{+/+}$ and $Bbs8^{-/-}$ mice and compared after merging with the "mergeCellChat" command.

Interaction difference heatmaps were created using the ComplexHeatmap package. Complex heatmaps reveal patterns and correlations in multidimensional genomic data (Gu et al, 2016).

In search for vessel-associated APC cells, the Seurat object was subset to contain only the three APC populations with the Fibroblast cluster. The "FindSubCluster" command was used with the "resolution = 0.3" argument to create two subclusters of the APC P2 cluster. 90.50, 88.46, and 87.34% of the cells of interest ("APC P2_1") still cluster together with 3, 4, or 5 subclusters created, respectively. The marker list (Dataset EV3) was generated using the "FindAllMarkers" command with "logfc.threshold = 0.1" and "only.pos = T" arguments, from the APC populations.

### Quantification, statistical analysis, and reproducibility

Results are presented as mean ± SEM or mean ± SD as indicated. The lean time point for each mouse line was chosen based on the mean body weight with the most similar value and the smallest standard deviation. All body weights were then normalized to this mean, respective for each genotype. Each mouse represents an independent biological sample. Prism 8.2.0 and 10.4.1 software was used to calculate $P$ values using a two-sided Mann–Whitney test, two-sided unpaired $t$-test, one-sample $t$-test, Kruskal–Wallis test with Dunn's posttest, or two-way ANOVA as indicated. Prism 8.2.0 and 10.4.1 software or the R programming language was used to generate graphical representations of quantitative data unless stated otherwise. Exact $P$ values are indicated in the figures. Sample sizes ($n$) are indicated in the figure legends.

The CD9 and FBN intensity in whole-mount stainings were quantified with the Imaris software. For CD9 intensity quantification, high-resolution confocal images in. ims format were opened using the 3D view in the Surpass interface. A 3D surface was created based on the PDGFRα signal using the "Add Surfaces" function. The mean intensity of the CD9 staining within this surface was measured using the "Statistics" tool. For FBN intensity quantification, high-resolution confocal images were similarly visualized in 3D, and the mean intensity was obtained directly from the FBN signal using the "Volume" and "Statistics" functions.

### Graphics

The Synopsis and Fig. 1A were created with BioRender.

## Data availability

Non-commercially available reagents or mouse lines can be made available under a material transfer agreement with the University Hospital Bonn. All data inquiries should be addressed to DW.

The datasets produced in this study are available in the following databases:

Imaging data: Figshare

https://figshare.com/projects/BBSome-dependent_ciliary_Hedgehog_signaling_governs_cell_fate_in_the_white_adipose_tissue/193025

RNA-seq data: Gene Expression Omnibus GSE254259: https://www.ncbi.nlm.nih.gov/geo/query/acc.cgi?acc=GSE254259

This paper does not report original code. All codes used in this paper are available from DW upon request.

The source data of this paper are collected in the following database record: biostudies:S-SCDT-10_1038-S44318-025-00524-y.

## Peer review information

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

## Acknowledgements

We thank Jens-Henning Krause, Maximilian Rothe, Rainer Stahl, and Romina Kaiser for technical support. We thank Heidi Hahn for providing *Ptch1*-flox mice, T. Schulz for providing the *Pdgfrα*-Cre mice, and S. Mukhopadhyay for providing the GPR161 antibody. We also thank the following Core Facilities of the Medical Faculty at the University of Bonn: Dr. Andreas Buness and Dr. Anshupa Sahu from the Core Facility Bioinformatics for help with data analysis;

the Cell Programming Core Facility, University Hospital Bonn for generating iPSC cells lines; the Microscopy Core Facility for providing help, services, and devices funded by the Deutsche Forschungsgemeinschaft (DFG, German Research Foundation)—project numbers 388168919, 388158066, 01EO2107, and 266686698; the Flow Cytometry Core Facility for providing help, services, and devices funded by the Deutsche Forschungsgemeinschaft (DFG, German Research Foundation)—project numbers 216372545 and 471514137; the Histology Platform of the ImmunoSensation[2] Cluster of Excellence, SFB 1454—Project-ID 432325352 (to DW, EM, MB), TRR333/1—Project-ID 450149205 (to DW), under Germany's Excellence Strategy—EXC2151—Project-ID 390873048 (to DW, EM), FOR5547—Project-ID 503306912 (to DW, EM) FOR5547—Project-ID 503306912 (to HLMS), SFB1366 - Project-ID—394046768 (to CRA), WA 3382/8-1—Project-ID 513767027, the Else Kröner Fresenius Foundation (2021.EKFSE.53) as well as intramural funding from the University of Bonn. EM is supported by the European Research Council (ERC) under the European Union's Horizon 2020 research and innovation program (Grant Agreement No. 851257). KS was supported with a PhD fellowship from the Studienstiftung des Deutschen Volkes, and DJSR was supported by a Postdoc fellowship from the Walter-Benjamin Program of the Deutsche Forschungsgemeinschaft (DFG, German Research Foundation). HLMS was funded by the Alexander von Humboldt Foundation (Sofja Kovalevskaja Award). CRA is supported by the European Research Council (ERC) - ERC-consolidator grant (ref. 864875).

## Author contributions

**Katharina Sieckmann**: Conceptualization; Data curation; Formal analysis; Validation; Visualization; Methodology; Writing—review and editing. **Nora Winnerling**: Conceptualization; Data curation; Formal analysis; Validation; Investigation; Methodology. **Dalila Juliana Silva Ribeiro**: Supervision; Validation; Investigation. **Seniz Yüksel**: Data curation; Formal analysis; Validation; Visualization. **Ronja Kardinal**: Data curation; Supervision; Validation. **Lisa Maria Steinheuer**: Formal analysis; Visualization. **Fabian Frechen**: Data curation; Formal analysis; Validation; Visualization. **Luis Henrique Corrêa**: Data curation; Formal analysis; Validation; Visualization. **Geza Schermann**: Data curation; Formal analysis; Visualization; Writing—review and editing. **Christina Klausen**: Data curation; Formal analysis; Supervision; Validation. **Nelli Blank-Stein**: Data curation; Formal analysis. **Jonas Schulte-Schrepping**: Validation; Methodology. **Collins Osei-Sarpong**: Formal analysis. **Matthias Becker**: Formal analysis; Methodology. **Lorenzo Bonaguro**: Formal analysis; Writing—review and editing. **Marc Beyer**: Formal analysis; Methodology. **Helen Louise May-Simera**: Resources; Validation; Methodology; Writing—review and editing. **Jelena Zurkovic**: Data curation; Formal analysis. **Christoph Thiele**: Data curation; Formal analysis; Validation. **Kevin Thurley**: Supervision; Methodology. **Lydia Sorokin**: Conceptualization; Resources; Supervision; Writing—review and editing. **Carmen Ruiz de Almodovar**: Formal analysis; Supervision; Funding acquisition; Validation; Writing—review and editing. **Elvira Mass**: Resources; Supervision; Writing—review and editing. **Dagmar Wachten**: Conceptualization; Data curation; Formal analysis; Supervision; Funding acquisition; Validation; Investigation; Writing—original draft; Project administration; Writing—review and editing.

Source data underlying figure panels in this paper may have individual authorship assigned. Where available, figure panel/source data authorship is listed in the following database record: biostudies:S-SCDT-10_1038-S44318-025-00524-y.

## Funding

## Disclosure and competing interests statement

The authors declare no competing interests.

