## [Peer Review File · The EMBO Journal]

BBS8-dependent ciliary Hedgehog signaling governs cell fate in the white adipose tissue

Katharina Sieckmann, Nora Winnerling, Dalila Silva Ribeiro, Seniz Yüksel, Ronja Kardinal, Lisa Steinheuer, Fabian Frechen, Luis Correa, Geza Schermann, Christina Klausen, Nelli Blank-Stein, Jonas Schulte-Schrepping, Collins Osei-Sarpong, Matthias Becker, Lorenzo Bonaguro, Marc Beyer, Helen May-Simera, Jelena Zurkovic, Christoph Thiele, Kevin Thurley, Lydia Sorokin, Carmen Ruiz de Almodóvar, Elvira Mass, and Dagmar Wachten

Corresponding author: Dagmar Wachten (dwachten@uni-bonn.de)

Review Timeline:

Submission Date:	20th Jun 24
Editorial Decision:	12th Aug 24
Appeal:	14th Aug 24
Editorial Decision:	16th Aug 24
Revision Received:	4th May 25
Editorial Decision:	18th Jun 25
Revision Received:	25th Jun 25
Accepted:	14th Jul 25

Editor: Ieva Gailite

Transaction Report:

Dear Dagmar,

Thank you for submitting your manuscript for consideration by The EMBO Journal. I sincerely apologise for the protracted assessment process due to delays in referee report submission. We have now received a full set of reviewer reports on your manuscript, which are included below for your information. Based on these comments, we unfortunately had to conclude that the study is not a sufficiently strong candidate for publication in The EMBO Journal.

As you can see, the reviewers appreciate the presented link between Bbs8 loss in adipocyte precursor cells and the fibrogenic fate switch. However, they also indicate that the generality of these findings for other BBSome components remains unclear. Furthermore, the finding that the causal relevance of ciliary Hh signalling activation to induction of differentiation defects and obesity would need substantial strengthening and currently is rather indirect. Additionally, reviewer #2 finds that the analysis of Bbs8 KO cell interactions with the ECM needs strengthening, while reviewer #3 raises a concern of potential neuronal impact on the observed obesity phenotypes. Taken together, since these concerns affect the main conclusions of the study and would require extensive additional experimental work with an unclear outcome, I am afraid that we cannot offer to invite a revision of your manuscript at The EMBO Journal.

While we cannot pursue this manuscript further, I would like to suggest a transfer to our not-for-profit open-access sister journal, Life Science Alliance (LSA). I have shared your manuscript and the accompanying reviews with LSA Executive Editor, Eric Sawey, who is interested in these findings, and would like to invite further consideration of this manuscript at LSA pending the following revisions:

- Address Reviewer 1's comments.
- Address Reviewer 2's Major concerns #1, 2 & 9 by toning down the claims and discussing limitations outlined. Major concerns #4, 5, 6, 7 & 8 should be addressed, as well as the Minor concerns.
- Address Reviewer 3's comments #3, 4, 5, 6 & 7. #1 can be addressed by re-organizing the data presentation as suggested.

We understand that such a revision might need to be re-reviewed, in which case, Dr. Sawey will walk the Reviewers through our transfer process.

You do not need to revise the manuscript before transferring it to LSA. Once you transfer, Dr. Sawey will email you an invitation to revise and resubmit, listing the same revision requests as mentioned above. Please feel free to reach out at e.sawey@life-science-alliance.org if you have any questions about the LSA journal, the transfer process or the revisions requested. If you are interested in this option, please use the link below to transfer your manuscript to LSA:

Link Not Available

Thank you in any case for the opportunity to consider this manuscript. I am sorry that I could not offer better news this time, but I nevertheless hope that you will find the transfer offer of interest.

With kind regards,

Ieva

Referee #1:

The relationship between cilia in preadipocyte differentiation into adipocytes is an open question. This article explores the intricate relationship between primary cilia dysfunction and adipose tissue remodeling, with a focus on Bardet-Biedl syndrome (BBS). The study employs a mouse model lacking BBS8 to delve into the role of primary cilia in controlling adipocyte precursor cell (APC) fate and function. The results provide significant insights into subpopulations of APCs affected from lack of BBS8 in

inducing a fibrogenic cell fate change that remodels the white adipose tissue. The manuscript is well written, and the rigor of the experimental results is high. The manuscript will be interest to readers from diverse fields including adipogenesis, cellular differentiation and signalling by primary cilia. I have a few comments on the interpretations of the results which might be addressed by commenting on the limitations of the study or being inclusive of alternative possibilities underlying the interesting fate trajectories described in the paper.

First, the major conjecture in the paper is cilia-dependent constitutive Hedgehog (Hh) signaling in BBC8 knockout (ko) APCs. The APCs/SVF used in the assays in Fig 5 likely have differing sub populations of P1-3 and differing amounts of ciliation. Can the results in Hh pathway be related to the differing amounts of ciliation in these cells? In principle, another model of constitutive Hh signaling could be relevant to address these issues.

Second, the results on fate determination in BBS8 ko can be a direct or indirect effect of the knockout. The age of the mice used in ko in isolating the APCs are not mentioned. I think it's important as the APCs might be different between lean vs obese conditions. The authors replicate data in knockout using Pdgfra-Cre mediated conditional knockout of BBS8, but the analysis timeline of APCs isolated is not clear. Do these fate determinants change with increasing age/fat mass accumulation?

Finally, there could be multiple reasons for obesity upon from BBS8 loss, such as food intake, thermogenesis, and adipocyte differentiation. The authors also use PDGFR α -cre mice, but this gene is widely expressed in other metabolically relevant organs such as the brain. Many ciliary mutations that lead to obesity phenotypes are related to excessive food intake as also in BBS8 ko mice. How can the authors conclude that the effects in differentiation in adipocytes is not from extraneous signals other than adipose tissues? These alternative possibilities should be addressed or discussed.

Minor:

1. Methods: Please mention age of animals during isolation of APCs in ko and conditional ko in all figs and methods.
2. The authors might rescue effects in BBS8 ko MEFs by adding back BBS8?
3. Pg 13 last para: "low grade stimulation of Hh signaling with SAG induced a fibrogenic switch in APCs": not sure if this was actually done?
4. Do the authors see any differences between male and female BBS8 ko/conditional ko mice?

Referee #2:

General summary and opinion about the principal significance of the study, its questions and findings

In manuscript EMBOJ-2024-118260-T, Sieckmann et al studied the role of ciliary signaling in adipose progenitor cells (APC) and adipogenesis. Patient suffering from developmental disorders caused by ciliary dysfunction, including Bardet-Biedl patients, suffer from obesity. The underlying reasons remain incompletely understood. Using a mouse model of Bardet-Biedl syndrome, Sieckmann et al show that loss of a ciliary signaling component (Bbs8) reduces the pool of stem-cell-like P1 APC which reflects a phenotypic switch of the cells into a fibrogenic progenitor state. This phenotypic switch is linked to ectopic hedgehog signaling. The authors show preliminary results suggesting the ectopic Hh activation is the reason of defect in lineage commitment. The manuscript is well written and reinforce the current knowledge on adipogenesis defects in Bardet-Biedl syndrome which new mechanistic insights. However, additional experiments are required to reinforce the general conclusions that are sometimes not enough supported by the current data.

Major concerns essential to be addressed to support the conclusions

1) The authors studied ciliary signaling defects resulting from Bbs8 knockout in the mouse. While the authors findings offer new insights into the mechanisms of adipogenesis defects in BBS patients affected by Bbs8 mutations, it is difficult to conclude whether or not the findings apply to other forms of BBS and even less clear whether or not the findings apply to other forms of ciliopathies as discussed in the paper. The authors should revise their general conclusion or extent their analysis to other Bbs mouse models or mouse models of ciliopathies if they want to broaden their conclusions. One could argue that the reported phenotype upon Bbs8 loss is specific to this model and could be linked to ciliary dysfunction and/or due cilia independent defects linked to cilia-independent function of Bbs8.

2) The authors claim very early in the paper (at the end of Figure 3), that Bbs8 deficiency cause defects in P1 APC and therefore primary cilia dysfunction cause defects in P1 APC. However, at this stage primary ciliogenesis and ciliary signaling upon Bbs8 loss have not been shown. The results presented at this stage therefore do not support the conclusion. Characterization of primary cilia in Bbs8 null mice should be shown earlier if the authors want to claim that ciliary dysfunction cause the phenotype.

3) The obesity phenotype is observed in adult mice (10 weeks) : Why ? does the appearance of the obesity phenotype in the adult correlates with activation of primary ciliogenesis and ciliary signaling in APC only the adult tissue ? The dynamics of ciliogenesis during development of this tissue would help understand the phenotype. Does BBS8 ablation in the adult mice

cause obesity ?

4) In figure 3, additional stainings of collagen fibers in mouse models is necessary to reach the conclusions on ECM remodeling which is mainly based on gene expression studies.

5) In figure 4G, the authors conclude that APC cells are the most active in cellular communications but P2 and P1 cells are very abundant in the analysis in comparison to endothelial cells and pericytes. How does it affect the probability of interactions in this analysis ?

6) In figure 4J, the presented data does not allow to visualize easily the increase in the the collagen-laminin interaction in Bbs8 null cells in comparison to wild-type cells. The interaction strength appears similar in wt vs knockout cells. Another visualization should be used and additional studies should be conducted to conclude on this signaling event (RT-qPCR, western blotting experiments using purified cells) and additional imaging studies which demonstrate localization of the cells in situ and expression of the signaling components could be performed.

7) In figure 5b, stimulation of Hedgehog signaling with SAG should be controlled via expression of Hedgehog component expression and processing (RNA and proteins).

8) In Figure 5i, the analysis of the impact of vismodegib should be conducted on additional Hh genes, since it is not possible to interpret appropriately the impact of Vismodegib with the presented data.

9) The data on SMO inhibition are too preliminary. The authors test whether Hh signaling inhibition via SMO inhibition represses Hh signaling (induced in Bbs8 null cells). The analysis of the impact of vismodegib should be conducted on additional Hh genes. Additionally, the impact of Vismodegib on other ECM genes that are upregulated in P1 APC in Bbs8 null mice should also be analyzed. Finally, if the authors want to test whether SMO inhibition is sufficient to repress Hh signaling in Bbs8 null mice, as claimed, the authors should perform in vivo treatment of Bbs8 null mice and assess the impact on Hh signaling in APC P1 after treatment and whether or not repressing Hh signaling correct the phenotype in mice.

Minor concerns that should be addressed

10) Stainings of basal bodies through would reinforce the observations that Arl13b structures presented in the paper are indeed primary cilia.

11) In differentially expressed gene analysis, the authors indicate that many genes are differentially expressed in the P1 population upon Bbs8 loss. Why does the principal component analysis does not segregate wt from ko cells (in PC2) if it is the case ?

12) The panels in Figure 5 are not appropriately cited.

13) There is a typo in Als(o), page 13.

Referee #3:

This study investigates the activity of adipocyte precursor cells (APCs) and remodeling of adipose tissue in Bbs8 null mice, which exhibit ciliary dysfunction and serve as a model for Bardet-Biedl syndrome. Deletion of Bbs8 globally or with the Pdgfra-Cre driver (which targets APCs) leads to adult onset obesity with corresponding adipose tissue growth and pathological remodeling. APCs from the adipose tissue of the null mice display a fibroblast phenotype shift, including increased expression of Cd9 and many ECM genes. In vitro studies show that APCs isolated from adipose tissue of Bbs8 KO mice are less adipogenic. Additional studies indicate that hedgehog signaling is increased under basal conditions in APCs from Bbs8 deficient mice, and that this contributes to the activation of fibroblast genes. Overall, the studies are well-executed from a technical perspective and there are some interesting and important observations. The effect of FFAR4 agonist on APC differentiation (Fig. 11,J) is striking. However, there are significant concerns, especially regarding the model and interpretations (#1) that should be addressed as outlined below:

1. The central concern relates to whether the observed effects in Bbs8 knockout mice are due to direct actions of ciliary signaling (Bbs8 function) on APCs or secondary to the obesity. Both Bbs8 deficient mouse models utilized develop obesity (and presumably insulin resistance), which is known to strongly modulate APC activity, including activation of Cd9 and fibroblast phenotypes (e.g PMID: 28215843). All the presented data suggest that the effects of Bbs8 KO on APCs are secondary to obesity/insulin resistance, which has not been adequately considered and diminishes novelty and impact of the study.

The authors could do a side-by-side comparison of APCs from HFD-induced obese vs. Bbs8 KO mice to see if there are any

differences. It may also be possible to re-organize the paper to emphasize the effects of obesity on APCs in this model of obesity, rather than emphasizing the direct actions of cilia signaling on APCs. The data in this regard nicely advance previous work in the field.

2. Related to above point, all of the presented in vitro data assess APCs isolated from lean (control) vs. obese Bbs8 deleted mice. Does acute Bbs8 deletion in APCs from lean mice alter adipogenic vs. fibroblast profile? Do the Bbs-KO APCs display a phenotype when isolated from young mice before body weight divergence?

3. The presented data strongly suggests that Pdgfra-Cre deletes Bbs8 in the brain/hypothalamus, leading to alterations in food intake and energy expenditure that underlie obesity development. The authors should examine Bbs8 deletion in brain regions of the Pdgfra-Cre/Bbs8 KO mice.

4. There does not seem to be defining markers of "FPCs" (Fig. 4)- is it more the lack of adipogenic/progenitor genes? Is Cd9 expression enriched in this population?

5. Fig. 5. The effect of basal HH signaling is interesting. Additional fibroblast genes including Cd9 should be assessed in panels j,k to determine if this treatment broadly alters cell phenotype. Also, the results in panels j,k should be analyzed together in the same graph so the reader can properly evaluate effects of Vismodegib treatment on the cells.

6. Fig. 4j. It seems that the main effect of Bbs8 deficiency is an increase in collagen in all APC types, rather than a specific effect on P2-1 as was concluded.

7. (minor) Fig. 1d. It is unclear what this data represents. The results section states that P1 was the most abundant APC subpopulation based on these data. However, it is unclear if P1 express higher levels of SCA1, which would alter the interpretation. The scRNAseq profiles from several labs do not suggest such a dramatic difference in the abundance of P1 vs other APC subtypes.

** As a service to authors, EMBO Press provides authors with the possibility to transfer a manuscript that one journal cannot offer to publish to another EMBO publication or the open access journal Life Science Alliance launched in partnership between EMBO Press, Rockefeller University Press and Cold Spring Harbor Laboratory Press. The full manuscript and if applicable, reviewers' reports, are automatically sent to the receiving journal to allow for fast handling and a prompt decision on your manuscript. For more details of this service, and to transfer your manuscript please click on Link Not Available. **

Referee #1:

The relationship between cilia in preadipocyte differentiation into adipocytes is an open question. This article explores the intricate relationship between primary cilia dysfunction and adipose tissue remodeling, with a focus on Bardet-Biedl syndrome (BBS). The study employs a mouse model lacking BBS8 to delve into the role of primary cilia in controlling adipocyte precursor cell (APC) fate and function. The results provide significant insights into subpopulations of APCs affected from lack of BBS8 in inducing a fibrogenic cell fate change that remodels the white adipose tissue. The manuscript is well written, and the rigor of the experimental results is high. The manuscript will be interest to readers from diverse fields including adipogenesis, cellular differentiation and signalling by primary cilia. I have a few comments on the interpretations of the results which might be addressed by commenting on the limitations of the study or being inclusive of alternative possibilities underlying the interesting fate trajectories described in the paper.

- 1. First, the major conjecture in the paper is cilia-dependent constitutive Hedgehog (Hh) signaling in BBC8 knockout (ko) APCs. The APCs/SVF used in the assays in Fig 5 likely have differing sub populations of P1-3 and differing amounts of ciliation. Can the results in Hh pathway be related to the differing amounts of ciliation in these cells? In principle, another model of constitutive Hh signaling could be relevant to address these issues.*

The experiments shown in Fig. 5 have been performed with cells that are enriched in P1 that only contain a few P2 cells, identified by FABP4 staining. Preliminary data from sorted P1 cells indicates that the effect is similar in this pure cell population. These experiments will be performed with higher n numbers for a revision.

The reviewer is correct that another model of constitutive Hh signaling could be relevant to address these issues. Indeed, these experiments have been in muscle tissue by the Kopinke lab, demonstrating that in FAPs (fibro/adipogenic progenitors), which seem to resemble a similar fibrogenic cell population that we have characterized in the adipose tissue, ectopic Hh activation in mice lacking PATCH1 in FAPs pushes them into a fibrotic fate (Norris et al., Nat. Com. 2023). Based on the availability of Patch1-flox mice, we could also recapitulate those experiments in the adipose tissue.

- 2. Second, the results on fate determination in BBS8 ko can be a direct or indirect effect of the knockout. The age of the mice used in ko in isolating the APCs are not mentioned. I think it's important as the APCs might be different between lean vs obese conditions. The authors replicate data in knockout using Pdgfra-Cre mediated conditional knockout of BBS8, but the analysis timeline of APCs isolated is not clear. Do these fate determinants change with increasing age/fat mass accumulation?*

We are sorry that we were not clear enough in describing our experiments. In fact, all of the data presented is from lean BBS8-KO mice (7 weeks) and not from obese mice. We only present the obese phenotype to demonstrate that it resembles BBS, but then analyze cilia-dependent cell fate changes at the lean stage (7 weeks) without any confounding effect of obesity. It has been shown by others that obesity changes the fate of the progenitor cells in the adipose tissue and indeed, similar changes can be recapitulated in obese BBS8-KO mice (data can be shown in a revised manuscript). Our aim was to reveal any changes that occur in a cilia-dependent manner in the adipose tissue before the onset of obesity, which is what we present in the paper.

- 3. Finally, there could be multiple reasons for obesity upon from BBS8 loss, such as food*

intake, thermogenesis, and adipocyte differentiation. The authors also use PDGFR α -cre mice, but this gene is widely expressed in other metabolically relevant organs such as the brain. Many ciliary mutations that lead to obesity phenotypes are related to excessive food intake as also in BBS8 ko mice. How can the authors conclude that the effects in differentiation in adipocytes is not from extraneous signals other than adipose tissues? These alternative possibilities should be addressed or discussed.

The reviewer is correct that obesity development in BBS8-KO mice can have multiple reasons. As pointed out by the reviewer, it has been demonstrated already that hyperphagia is one the major contributors, but that also other factors contribute as paired-feeding experiments also resulted in obesity (Rahmouni et al., JCI 2008). To avoid the confounding factors of obesity, we analyzed the mice in the lean state to identify the changes that occur in the adipose tissue in a cilia-dependent manner, independent of obesity.

Minor:

1. Methods: Please mention age of animals during isolation of APCs in ko and conditional ko in all figs and methods.

We are sorry about the confusion and will add the information accordingly.

2. The authors might rescue effects in BBS8 ko MEFs by adding back BBS8?

This is an experiment that we will do in the revision.

3. Pg 13 last para: "low grade stimulation of Hh signaling with SAG induced a fibrogenic switch in APCs": not sure if this was actually done?

We indeed analyzed the fibrogenic gene expression pattern upon low grade Hh stimulation in Hh.

4. Do the authors see any differences between male and female BBS8 ko/conditional ko mice?

We did not observe sex-specific differences in our ex vivo or in vitro assays.

Referee #2:

General summary and opinion about the principal significance of the study, its questions and findings

In manuscript EMBOJ-2024-118260-T, Sieckmann et al studied the role of ciliary signaling in adipose progenitor cells (APC) and adipogenesis. Patient suffering from developmental disorders caused by ciliary dysfunction, including Bardet-Biedl patients, suffer from obesity. The underlying reasons remain incompletely understood. Using a mouse model of Bardet-Biedl syndrome, Sieckmann et al show that loss of a ciliary signaling component (*Bbs8*) reduces the pool of stem-cell-like P1 APC which reflects a phenotypic switch of the cells into a fibrogenic progenitor state. This phenotypic switch is linked to ectopic hedgehog signaling. The authors show preliminary results suggesting the ectopic Hh activation is the reason of defect in lineage commitment. The manuscript is well written and reinforce the current knowledge on adipogenesis defects in Bardet-Biedl syndrome which new mechanistic insights. However, additional experiments are required to reinforce the general conclusions that are sometimes not enough supported by the current data.

Major concerns essential to be addressed to support the conclusions

*1) The authors studied ciliary signaling defects resulting from *Bbs8* knockout in the mouse. While the authors findings offer new insights into the mechanisms of adipogenesis defects in BBS patients affected by *Bbs8* mutations, it is difficult to conclude whether or not the findings apply to other forms of BBS and even less clear whether or not the findings apply to other forms of ciliopathies as discussed in the paper. The authors should revise their general conclusion or extent their analysis to other *Bbs* mouse models or mouse models of ciliopathies if they want to broaden their conclusions. One could argue that the reported phenotype upon *Bbs8* loss is specific to this model and could be linked to ciliary dysfunction and/or due cilia independent defects linked to cilia-independent function of *Bbs8*.*

We agree with the reviewer that only presenting data for *Bbs8* limits drawing general conclusions. However, we have already performed experiments similar experiments in mice that completely lack cilia in APCs, which also demonstrate a switch to a fibrogenic gene expression pattern. This will be included in the revised manuscript, and we will explicitly state this in our conclusion to avoid giving the wrong impression.

*2) The authors claim very early in the paper (at the end of Figure 3), that *Bbs8* deficiency cause defects in P1 APC and therefore primary cilia dysfunction cause defects in P1 APC. However, at this stage primary ciliogenesis and ciliary signaling upon *Bbs8* loss have not been shown. The results presented at this stage therefore do not support the conclusion. Characterization of primary cilia in *Bbs8* null mice should be shown earlier if the authors want to claim that ciliary dysfunction cause the phenotype.*

We thank the reviewer for this comment and will change the order of the data in the revised version of the manuscript.

3) The obesity phenotype is observed in adult mice (10 weeks): Why? does the appearance of the obesity phenotype in the adult correlates with activation of primary ciliogenesis and ciliary signaling in APC only the adult tissue? The dynamics of ciliogenesis during development of this tissue would help understand the phenotype. Does BBS8 ablation in the adult mice cause obesity?

The onset of obesity can be explained by the hyperphagia, which requires a couple of weeks to result in a significant increase in body weight. The contribution of BBS8 in the adipose tissue

in adipocyte precursors cells could be analyzed by an inducible PDGF α -CreERT2 model, which allows to induce the deletion in the adult state.

4) In figure 3, additional stainings of collagen fibers in mouse models is necessary to reach the conclusions on ECM remodeling which is mainly based on gene expression studies.

We were of this point and have already started performing whole-mount labeling for ECM markers in the adipose tissue from lean wild-type and Bbs8-KO mice. This data will be included in the revised manuscript.

5) In figure 4G, the authors conclude that APC cells are the most active in cellular communications but P2 and P1 cells are very abundant in the analysis in comparison to endothelial cells and pericytes. How does it affect the probability of interactions in this analysis?

We will reanalyze our data accordingly and include this information in the revised manuscript.

6) In figure 4J, the presented data does not allow to visualize easily the increase in the collagen-laminin interaction in Bbs8 null cells in comparison to wild-type cells. The interaction strength appears similar in wt vs knockout cells. Another visualization should be used and additional studies should be conducted to conclude on this signaling event (RT-qPCR, western blotting experiments using purified cells) and additional imaging studies which demonstrate localization of the cells in situ and expression of the signaling components could be performed.

We will discuss different ways of data presentation and include this in the revised manuscript. Furthermore, as stated in the answer to the previous comment, we are already performing in situ labeling to further analyze the signaling. This will also be included in the revised manuscript.

7) In figure 5b, stimulation of Hedgehog signaling with SAG should be controlled via expression of Hedgehog component expression and processing (RNA and proteins).

We have already analyzed Gli processing using immunoblots, which we will include in the revised version of the manuscript. Additional genes can also be included in our mRNA expression analysis.

8) In Figure 5i, the analysis of the impact of vismodegib should be conducted on additional Hh genes, since it is not possible to interpret appropriately the impact of Vismodegib with the presented data.

We will include this data in the revised manuscript.

9) The data on SMO inhibition are too preliminary. The authors test whether Hh signaling inhibition via SMO inhibition represses Hh signaling (induced in Bbs8 null cells). The analysis of the impact of vismodegib should be conducted on additional Hh genes. Additionally, the impact of Vismodegib on other ECM genes that are upregulated in P1 APC in Bbs8 null mice should also be analyzed. Finally, if the authors want to test whether SMO inhibition is sufficient to repress Hh signaling in Bbs8 null mice, as claimed, the authors should perform in vivo treatment of Bbs8 null mice and assess the impact on Hh signaling in APC P1 after treatment and whether or not repressing Hh signaling correct the phenotype in mice.

We will include additional genes in our mRNA expression analysis. We would love to perform in vivo treatment with Vismodegib in mice, but getting the animal license approved can easily take another 6 months, which is most likely beyond the scope of the current manuscript.

Minor concerns that should be addressed

10) Stainings of basal bodies would reinforce the observations that Arl13b structures presented in the paper are indeed primary cilia.

We will include a basal body marker in the revised manuscript.

11) In differentially expressed gene analysis, the authors indicate that many genes are differentially expressed in the P1 population upon Bbs8 loss. Why does the principal component analysis does not segregate wt from ko cells (in PC2) if it is the case?

We will reanalyze the data to address the question of the reviewer and include the information in the revised version of the manuscript.

12) The panels in Figure 5 are not appropriately cited.

We will change this accordingly.

13) There is a typo in Als(o), page 13.

We will change this accordingly.

Referee #3:

This study investigates the activity of adipocyte precursor cells (APCs) and remodeling of adipose tissue in *Bbs8* null mice, which exhibit ciliary dysfunction and serve as a model for Bardet-Biedl syndrome. Deletion of *Bbs8* globally or with the *Pdgfra*-Cre driver (which targets APCs) leads to adult onset obesity with corresponding adipose tissue growth and pathological remodeling. APCs from the adipose tissue of the null mice display a fibroblast phenotype shift, including increased expression of *Cd9* and many ECM genes. In vitro studies show that APCs isolated from adipose tissue of *Bbs8* KO mice are less adipogenic. Additional studies indicate that hedgehog signaling is increased under basal conditions in APCs from *Bbs8* deficient mice, and that this contributes to the activation of fibroblast genes. Overall, the studies are well-executed from a technical perspective and there are some interesting and important observations. The effect of FFAR4 agonist on APC differentiation (Fig. 11,J) is striking. However, there are significant concerns, especially regarding the model and interpretations (#1) that should be addressed as outlined below:

1. The central concern relates to whether the observed effects in Bbs8 knockout mice are due to direct actions of ciliary signaling (Bbs8 function) on APCs or secondary to the obesity. Both Bbs8 deficient mouse models utilized develop obesity (and presumably insulin resistance), which is known to strongly modulate APC activity, including activation of Cd9 and fibroblast phenotypes (e.g PMID: 28215843). All the presented data suggest that the effects of Bbs8 KO on APCs are secondary to obesity/insulin resistance, which has not been adequately considered and diminishes novelty and impact of the study.

The authors could do a side-by-side comparison of APCs from HFD-induced obese vs. Bbs8 KO mice to see if there are any differences. It may also be possible to re-organize the paper to emphasize the effects of obesity on APCs in this model of obesity, rather than emphasizing the direct actions of cilia signaling on APCs. The data in this regard nicely advance previous work in the field.

The reviewer is correct that analyzing APCs in the obese state would not allow to make proper conclusions due to confounding factors that occur during obesity. This is exactly why we focused on the lean state to only focus on the cilia-dependent effects in the adipose tissue independent of obesity. All the data presented has been performed in lean BBS8-KO mice (7 weeks). We are sorry if we did not make this clear enough in the current version of the manuscript and will change this accordingly.

2. Related to above point, all of the presented in vitro data assess APCs isolated from lean (control) vs. obese Bbs8 deleted mice. Does acute Bbs8 deletion in APCs from lean mice alter adipogenic vs. fibroblast profile? Do the Bbs-KO APCs display a phenotype when isolated from young mice before body weight divergence?

Indeed, APCs change their fate in lean BBS8-KO as we show in the manuscript, which has solely been performed on lean mice. Again, we are sorry that this was not made clear enough in our description and we will change this accordingly.

3. The presented data strongly suggests that Pdgfra-Cre deletes Bbs8 in the brain/hypothalamus, leading to alterations in food intake and energy expenditure that underlie obesity development. The authors should examine Bbs8 deletion in brain regions of the Pdgfra-Cre/Bbs8 KO mice.

We agree with the reviewer that this is an important point to consider when analyzing the mice at the obese state. However, as we only assessed the lean phenotype at 7 weeks and focused on the phenotype in the adipose tissue independent of obesity.

4. *There does not seem to be defining markers of "FPCs" (Fig. 4)- is it more the lack of adipogenic/progenitor genes? Is Cd9 expression enriched in this population?*

We already have analyzed this and will address this in a revised version.

5. *Fig. 5. The effect of basal HH signaling is interesting. Additional fibroblast genes including Cd9 should be assessed in panels j,k to determine if this treatment broadly alters cell phenotype. Also, the results in panels j,k should be analyzed together in the same graph so the reader can properly evaluate effects of Vismodegib treatment on the cells.*

We will perform experiments in this direction and change the presentation accordingly.

6. *Fig. 4j. It seems that the main effect of Bbs8 deficiency is an increase in collagen in all APC types, rather than a specific effect on P2-1 as was concluded.*

This will be addressed in a revised version as we have more expression data that we did not include yet.

7. *(minor) Fig. 1d. It is unclear what this data represents. The results section states that P1 was the most abundant APC subpopulation based on these data. However, it is unclear if P1 express higher levels of SCA1, which would alter the interpretation. The scRNAseq profiles from several labs do not suggest such a dramatic difference in the abundance of P1 vs other APC subtypes.*

The reviewer is correct and we will account for the difference in expression in P1 in the revised manuscript.

Dear Dagmar,

Thank you for contacting me with a preliminary revision plan for your manuscript. I have now gone through it, and I find your responses reasonable. I think that the clarification that the experiments were performed in a lean state will be very helpful. I am also glad to see that you already have set in motion experiments that will likely address most of the remaining concerns.

Therefore, I would like to invite you to submit a revised manuscript in response to the reviewers' comments. However, please note that at this point I am not able to predict whether the reviewers will be satisfied with the revision. We will ultimately require strong support from the reviewers for publication here.

We generally allow three months as standard revision time, which can be extended to six months in the case of major revisions. Should you foresee a problem in meeting this deadline, please let us know in advance to discuss an extension.

As a matter of policy, competing manuscripts published during this period will not negatively impact on our assessment of the conceptual advance presented by your study. However, please contact me as soon as possible upon publication of any related work to discuss the appropriate course of action.

When preparing your letter of response to the referees' comments, please bear in mind that this will form part of the Review Process File and will therefore be available online to the community. For more details on our Transparent Editorial Process, please visit our website: <https://www.embopress.org/page/journal/14602075/authorguide#transparentprocess>. Please also see the attached instructions for further guidelines on preparation of the revised manuscript.

Please feel free to contact me if you have any further questions regarding the revision. Thank you for the opportunity to consider your work for publication. I look forward to receiving the revised manuscript.

With best wishes,

Ieva

Ieva Gailite, PhD
Senior Scientific Editor
The EMBO Journal
Meyershofstrasse 1
D-69117 Heidelberg
Tel: +4962218891309
i.gailite@embojournal.org

- a Reagents and Tools Table as part of the Methods section, which can be downloaded from our author guidelines

(<https://www.embopress.org/page/journal/14602075/authorguide#structuredmethods>)

We realize that it is difficult to revise to a specific deadline. In the interest of protecting the conceptual advance provided by the work, we recommend a revision within 3 months (14th Nov 2024). Please discuss the revision progress ahead of this time with the editor if you require more time to complete the revisions.

Referee #1:

The relationship between cilia in preadipocyte differentiation into adipocytes is an open question. This article explores the intricate relationship between primary cilia dysfunction and adipose tissue remodeling, with a focus on Bardet-Biedl syndrome (BBS). The study employs a mouse model lacking BBS8 to delve into the role of primary cilia in controlling adipocyte precursor cell (APC) fate and function. The results provide significant insights into subpopulations of APCs affected from lack of BBS8 in inducing a fibrogenic cell fate change that remodels the white adipose tissue. The manuscript is well written, and the rigor of the experimental results is high. The manuscript will be interest to readers from diverse fields including adipogenesis, cellular differentiation and signalling by primary cilia. I have a few comments on the interpretations of the results which might be addressed by commenting on the limitations of the study or being inclusive of alternative possibilities underlying the interesting fate trajectories described in the paper.

- 1. First, the major conjecture in the paper is cilia-dependent constitutive Hedgehog (Hh) signaling in BBC8 knockout (ko) APCs. The APCs/SVF used in the assays in Fig 5 likely have differing sub populations of P1-3 and differing amounts of ciliation. Can the results in Hh pathway be related to the differing amounts of ciliation in these cells? In principle, another model of constitutive Hh signaling could be relevant to address these issues.*

The reviewer is correct that in Fig. 5, we analyzed the total SVF (Lin -), which includes all three subpopulations. We have established a magnetic cell separation strategy to isolate P1 cells and verified this by marker staining (i.e., 1-3% impurity of P2 cells in the isolated P1 cells according to marker staining and image analysis). We have performed additional experiments using the isolated P1 cells and the results are similar: basal ciliary SMO levels and mRNA expression of *Gli1* are increased (see below Fig. a and b). In turn, upon stimulation with SAG, the response is blunted (see below Fig. c). We have included the data from enriched P1 cells in Fig. S5f-i in the revised manuscript.

We agree with the reviewer that another model of constitutive Hh signaling would be relevant in this context. Indeed, the Kopinke lab has analyzed Hh signaling in muscle tissue in FAPs (fibro/adipogenic progenitors) and used mice lacking PTCH1 in FAPs. FAPs seem to resemble a similar fibrogenic cell population as the one that we have characterized in the adipose tissue. They could demonstrate that ectopic Hh activation in mice lacking PTCH1 in FAPs pushes the cells into a fibrogenic fate (Norris et al., Nat. Com. 2023). To analyze whether this is also relevant in APCs, we have generated *Pdgfra*^{+creERT2} *Ptch1*^{fl/fl} mice to delete *Ptch1* in PDGFR α -expressing cells upon tamoxifen treatment. To this end, we isolated primary APCs and treated them *in vitro* with 10 nM tamoxifen. Our results demonstrated that deletion of *Ptch1* also led to ectopic activation of the Hh pathway in APCs, as evidenced by increased basal ciliary SMO levels, leading to an increased mRNA expression of *Gli1* and a blunted response to Hh stimulation (see below Fig. e-g). Finally, we assessed another model that lacks primary cilia in APCs (*Pdgfra*^{Cre/+} *Ift20*^{fllox/fllox}) to assess whether this also induces a fate change. Indeed, primary cilia removal led to an upregulation of Hh target genes, which was accompanied by a switch to a fibrogenic gene expression pattern.

We have included the data from the *Pdgfra*^{+creERT2} *Ptch1*^{fl/fl} cells and *Pdgfra*^{Cre/+} *Ift20*^{fllox/fllox} mice in Fig. S6c-k in the revised manuscript.

Constitutive Hedgehog (Hh) response in *Bbs8*^{-/-} APC P1 cells and *Pdgfra*^{+/*CreERT2*} *Ptch1*^{fl/fl} cells. a. Fluorescence confocal images of APC P1 cells isolated by magnetic separation from *Bbs8*^{+/*+*} and *Bbs8*^{-/-} mice at 5-9 weeks of age (lean state). Cells were stained for ARL13B (red, cilia), γ -Tubulin (magenta, basal body), and Smoothed (SMO, green, 5 px shift indicated by the green arrow), and with DAPI (blue). Cells were treated with 1 μ M SAG or vehicle control (H₂O) for 24 h. **b.** Quantification of ciliary SMO localization in APC P1 cells shown in (a). Each data point represents one animal (n = 4; >4 cilia per n), p-values have been determined using one-way ANOVA. **c-d.** Relative mRNA expression of *Gli1* and *Ptch1* in APC P1 from *Bbs8*^{+/*+*} and *Bbs8*^{-/-} mice at 5-9 weeks of age (lean state) under (c) basal conditions or (d) after treatment with SAG (1 μ M) for 24 h, assessed by qRT-PCR. Expression values were normalized to *Bbs8*^{+/*+*} levels. Each data point represents one animal (n = 6-8), p-values have been determined using a one-sample t-test. **e.** Fluorescence confocal images of APCs isolated from *Pdgfra*^{+/*+*} *Ptch1*^{fl/fl} and *Pdgfra*^{+/*CreERT2*} *Ptch1*^{fl/fl} mice at 5-7 weeks of age and treated with 10 nM tamoxifen for 48 h. Cells were stained for ARL13B (red, cilia), γ -Tubulin (magenta, basal body), and SMO (green, 5 px shift indicated by the green arrow), and with DAPI (blue). **f.** Quantification of ciliary SMO localization shown in (e). Each data point represents one animal (n = 4; >35 cilia per n), p-values have been determined using an unpaired Student's t-test. **g.** Relative mRNA expression of *Gli1* in *Pdgfra*^{+/*+*} *Ptch1*^{fl/fl} and *Pdgfra*^{+/*CreERT2*} *Ptch1*^{fl/fl} cells after 48 h of 10 nM tamoxifen treatment, assessed by qRT-PCR. The expression values were normalized to *Pdgfra*^{+/*+*} *Ptch1*^{fl/fl} levels. Each data point represents one animal (n = 5), p-values have been determined using a one-sample t-test. Scale bars are indicated. All data are shown as mean \pm SD.

2. Second, the results on fate determination in *BBS8* ko can be a direct or indirect effect of the knockout. The age of the mice used in ko in isolating the APCs are not mentioned. I think it's important as the APCs might be different between lean vs obese conditions. The authors replicate data in knockout using *Pdgfra*-Cre mediated conditional knockout of *BBS8*, but the analysis timeline of APCs isolated is not clear. Do these fate determinants change with increasing age/fat mass accumulation?

We are sorry that we were not clear enough in describing our experiments. In fact, all the data presented are generated from lean *Bbs8*^{-/-} and lean *Pdgfra*^{Cre/+} *Bbs8*^{fl/fl} mice (5-9 weeks), before the onset of obesity. We only present the obesity phenotype in Fig. 2 to demonstrate that the mouse model resembles BBS. The cilia-dependent cell fate changes were uncovered at the lean stage (5-9 weeks) without any confounding effect of obesity. It has been shown by others that obesity reduces the stem-cell like P1 progenitor cells in the adipose tissue. We have

verified those results in a diet-induced obesity (DIO) model using wild-type mice ($Bbs8^{+/+}$ DIO), demonstrating that the P1 cells are reduced in obese mice compared to age-matched (18-20 wks) lean $Bbs8^{+/+}$ mice (see figure below). Comparing the results from obese $Bbs8^{+/+}$ DIO to experiments using obese $Bbs8^{-/-}$ mice revealed that both, $Bbs8^{+/+}$ DIO and obese $Bbs8^{-/-}$ mice show the reduction in P1 compared to age-matched (18-20 wks) lean $Bbs8^{+/+}$ mice, but only $Bbs8^{-/-}$ mice show an increase in P3 (see figure below). Thus, compared to the lean state, the loss in P1 and the increase in P3 are maintained in obese $Bbs8^{-/-}$ mice.

Changes in APC subpopulations. a. Frequency distribution of P1-P3 from the total APC pool from 18-20 weeks old $Bbs8^{+/+}$ mice on chow diet or on high-fat diet ($Bbs8^{+/+}$ DIO), and obese $Bbs8^{-/-}$ mice. P-values were calculated by Šidák-multiple comparison test, individual dots represent individual mice.

3. Finally, there could be multiple reasons for obesity upon from BBS8 loss, such as food intake, thermogenesis, and adipocyte differentiation. The authors also use $PDGFR\alpha$ -cre mice, but this gene is widely expressed in other metabolically relevant organs such as the brain. Many ciliary mutations that lead to obesity phenotypes are related to excessive food intake as also in BBS8 ko mice. How can the authors conclude that the effects in differentiation in adipocytes is not from extraneous signals other than adipose tissues? These alternative possibilities should be addressed or discussed.

The reviewer is correct that obesity development in $Bbs8^{-/-}$ and in $Pdgfra^{Cre/+} Bbs8^{fl/fl}$ mice can have multiple reasons. As pointed out by the reviewer, it has been demonstrated already that hyperphagia is one of the major but not the only contributor to obesity: paired-feeding experiments, in which Bbs knock-out mice received the same amount of food, also resulted in obesity (Rahmouni et al., JCI 2008). Our results using $Pdgfra^{Cre/+} Bbs8^{fl/fl}$ mice indicate that the lack of $Bbs8$ in $PDGFR\alpha$ -expressing cells is sufficient to induce adipose tissue expansion. To avoid the confounding factors of hyperphagia and obesity, we analyzed the mice in the lean state to identify the changes that occur in the adipose tissue in a cilia-dependent manner independent of obesity. We have included the age of the animals and that they are lean in the text and the figure legend to avoid any confusion. We also demonstrate that stimulation of Hh signaling in APCs is sufficient to promote fibrogenic gene expression and that blocking Smo in $Bbs8^{-/-}$ APCs is sufficient to rescue the phenotype. Thus, although we cannot rule out the contribution of other cells than APCs to the fibrogenic switch observed in the lean state, they seem to be the major contributor.

Minor:

1. Methods: Please mention age of animals during isolation of APCs in ko and conditional ko in all figs and methods.

We are sorry about the confusion and have added the information accordingly.

2. The authors might rescue effects in *BBS8* ko MEFs by adding back *BBS8*?

We thank the reviewer for this suggestion and have performed the experiments accordingly. We aimed to rescue the effects on Hedgehog signaling in *Bbs8*^{-/-} MEFs by overexpressing HA-tagged *Bbs8* using electroporation. MEFs were successfully transfected and *Bbs8*-HA localized to the primary cilium (see figure a, b below). Subsequently, we analyzed the ciliary localization of Smoothened (SMO) under basal conditions and could demonstrate that overexpression of *Bbs8*-HA in *Bbs8*^{-/-} MEFs significantly reduced the ciliary SMO localization (see figure c below). To investigate whether this also affects downstream Hh signaling, we analyzed Hh target gene expression. Indeed, overexpression of *Bbs8*-HA in *Bbs8*^{-/-} MEFs decreased the expression of *Ptch1* (see figure d below). We have included this data set in the Fig. S5m-p in the revised manuscript.

Overexpression of *Bbs8* rescues Hedgehog phenotype in *Bbs8*^{-/-} MEFs. MEFs were transfected with the pBbs8-HA construct via nucleofection. **a.** Transfection was assessed by fluorescent antibody staining against the HA-tag (red, px shift in insets is indicated by the arrow) in non-transfected and transfected *Bbs8*^{-/-} MEFs. **b.** Quantification of ciliary HA-tag localization shown in (a). Each data point represents one cilium ($n = 3$; > 6 cilia per n); p-value was determined by an unpaired Student's t-test. **c.** Quantification of ciliary SMO localization in non-transfected *Bbs8*^{+/+} and *Bbs8*^{-/-} MEFs, and in transfected *Bbs8*^{-/-} MEFs. Each data point represents a biological replicate ($n = 3$; > 5 cilia per n); p-values were determined by one-way ANOVA. **d.** Basal Hh target gene expression in non-transfected *Bbs8*^{-/-} and transfected *Bbs8*^{-/-} MEF cells. Gene expression was normalized to *Bbs8*^{+/+} expression values (shown as the dotted line). Data are shown as individual values (dots) and mean (bars) \pm S.D. Different n are indicated by dots; p-values were calculated by an unpaired Student's t-test. Each data point represents one biological replicate ($n = 3-9$), p-values have been determined using an unpaired Student's t-test. Scale bars are indicated.

3. Pg 13 last para: "low grade stimulation of Hh signaling with SAG induced a fibrogenic switch in APCs": not sure if this was actually done?

We indeed analyzed the fibrogenic gene expression pattern upon low grade Hh stimulation (100 nM SAG) in 3T3-L1 adipocyte progenitor cells, demonstrating that the fibrogenic marker expression is increased (old Fig. 5b for *Col1a1*, now expanded to *Col1a1*, *Col5a1*, *Sparc*, and *TnC* in Fig. 5b of the revised manuscript).

Fibrogenic gene expression after Hedgehog signaling activation in 3T3-L1 cells. Relative mRNA expression of fibrogenic marker genes in 3T3-L1 cells, with and without treatment with 100 nM SAG for 24 h, assessed by qRT-PCR. Expression values were normalized to control treated cells. Data are shown as mean \pm SD; p-values were calculated by a one-sample Student's t-test.

4. Do the authors see any differences between male and female *BBS8* ko/conditional ko mice?

We have color-coded the individual values for the respective mice regarding tissue weights in the obese state and did not observe consistent sex-specific differences (see figure below). Furthermore, our *in vitro* assays were also performed with animals from both sexes, and we did not observe any differences.

Adipose tissue weights. **a.** Weights of gonadal (gWAT) and inguinal white adipose tissue (iWAT) as well as interscapular brown adipose tissue (iBAT) from chow diet-fed *Bbs8*^{+/+} and *Bbs8*^{-/-} mice at 15-18 weeks. **b.** Weights of gonadal (gWAT) and inguinal white adipose tissue (iWAT) from chow diet-fed *Pdgfra*^{+/+}, *Bbs8*^{fl/fl} and *Pdgfra*^{Cre/+}, *Bbs8*^{fl/fl} mice at 18-25 weeks. Data are shown as mean \pm SD; the sex of individual mice is indicated with symbols.

Referee #2:

General summary and opinion about the principal significance of the study, its questions and findings

In manuscript EMBOJ-2024-118260-T, Sieckmann et al studied the role of ciliary signaling in adipose progenitor cells (APC) and adipogenesis. Patient suffering from developmental disorders caused by ciliary dysfunction, including Bardet-Biedl patients, suffer from obesity. The underlying reasons remain incompletely understood. Using a mouse model of Bardet-Biedl syndrome, Sieckmann et al show that loss of a ciliary signaling component (*Bbs8*) reduces the pool of stem-cell-like P1 APC which reflects a phenotypic switch of the cells into a fibrogenic progenitor state. This phenotypic switch is linked to ectopic hedgehog signaling. The authors show preliminary results suggesting the ectopic Hh activation is the reason of defect in lineage commitment. The manuscript is well written and reinforce the current knowledge on adipogenesis defects in Bardet-Biedl syndrome which new mechanistic insights. However, additional experiments are required to reinforce the general conclusions that are sometimes not enough supported by the current data.

Major concerns essential to be addressed to support the conclusions

1) *The authors studied ciliary signaling defects resulting from Bbs8 knockout in the mouse. While the authors findings offer new insights into the mechanisms of adipogenesis defects in BBS patients affected by Bbs8 mutations, it is difficult to conclude whether or not the findings apply to other forms of BBS and even less clear whether or not the findings apply to other forms of ciliopathies as discussed in the paper. The authors should revise their general conclusion or extent their analysis to other Bbs mouse models or mouse models of ciliopathies if they want to broaden their conclusions. One could argue that the reported phenotype upon Bbs8 loss is specific to this model and could be linked to ciliary dysfunction and/or due cilia independent defects linked to cilia-independent function of Bbs8.*

We agree with the reviewer that only presenting data for loss of *Bbs8* limits drawing general conclusions. Therefore, we conducted additional experiments using three other models *in vitro* and *in vivo*.

First, we analyzed human iPSCs carrying a homozygous missense mutation (M390R) in the *BBS1* gene, the most common disease-causing mutation for BBS (Davis et al., *PNAS* 2007). Our aim was to determine whether the loss of other BBSome components also affects SMO localization. Indeed, we observed a phenotype similar to the loss of *BBS8*: SMO localization to the primary cilium was significantly increased at the basal state *Bbs1*^{M390R/M390R} compared to *Bbs1*^{+/+} hiPSCs (see figure below, a-b). These findings support our original results and suggest a general involvement of the BBSome in controlling SMO localization.

Second, we investigated another model of constitutive Hh signaling. Since PTCH1 represses ciliary SMO localization under basal conditions, we deleted *Ptch1* in APCs isolated from *Pdgfra*^{+creERT2} *Ptch1*^{fl/fl} mice by treating these cells with tamoxifen. The deletion of *Ptch1* increased the ciliary SMO localization and, consequently, increased expression of the Hh target gene *Gli1* downstream and blunted the Hh-dependent change in gene expression (see figure below, c-e). The Kopinke lab has analyzed Hh signaling in muscle tissue in FAPs (fibro/adipogenic progenitors) and used mice lacking PTCH1 in FAPs. FAPs appear to resemble the fibrogenic cell population we have characterized in the adipose tissue. They could demonstrate that ectopic Hh activation in mice lacking PTCH1 in FAPs pushes the cells into a fibrotic fate (Norris et al., *Nat. Com.* 2023).

Finally, to assess whether the loss of the primary cilium induces a fate change in APCs, we generated mice that lack primary cilia in PDGFR α -expressing cells (*Pdgfra*^{+cre} *Ift20*^{flox/flox}). Since these mice die within a few weeks after birth, we isolated the gonadal white adipose tissue (gWAT), which fully differentiates postnatally, at P4, when it only contains APCs and not yet mature adipocytes (Han et al., *Development* 2011). PDGFR α ⁺ cells in wild-type gWAT at P4 were ciliated and responded to Hh stimulation (see below f-i). Primary cilia removal led to

an upregulation of Hh target genes (see figure below, j). This was accompanied by a switch to a fibrogenic gene expression pattern in the precursor tissue at P4 (see figure below, k). We have included these data sets in Fig. S6 in the revised manuscript.

Hedgehog (Hh) phenotype in different ciliopathy models. **a**, Fluorescence confocal images of *BBS1*^{+/+} and *BBS1*^{M390R/M390R} iPSCs, labeled against ARL13B (red, cilia) and Smoothed (green, SMO, 5 px shift indicated by the green arrow), and with DAPI (blue). Cells were treated with H₂O (control) or 1 μM SAG for 24 h. **b**, Quantification of the ciliary SMO localization in *Bbs1*^{+/+} and *Bbs1*^{M390R/M390R} iPSCs. Each data point represents a technical replicate from n = 2 biological replicates (>100 cilia per n); p-value has been determined using an unpaired student's t-test. **c**, Fluorescence confocal images of APCs isolated from *Pdgfra*^{+/+} *Ptch1*^{fl/fl} and *Pdgfra*^{+CreERT2} *Ptch1*^{fl/fl} mice at 5-7 weeks of age and treated with 10 nM tamoxifen for 48 h. Cells were stained for ARL13B (red, cilia), γ-Tubulin (magenta, basal body), and SMO (green), and with DAPI (blue). In all images, the green channel (SMO) was shifted by 5 px for better visualizing SMO accumulation. **d**, Quantification of ciliary SMO localization shown in (c). Each data point represents one animal (n = 4; >30 cilia per n), p-values have been determined using an unpaired Student's t-test. **e**, Relative mRNA expression of *Gli1* in *Pdgfra*^{+/+} *Ptch1*^{fl/fl} and *Pdgfra*^{+CreERT2} *Ptch1*^{fl/fl} cells after 48 h of 10 nM tamoxifen treatment, assessed by qRT-PCR. The expression values were normalized to *Pdgfra*^{+/+} *Ptch1*^{fl/fl} levels. Each data point represents one animal (n = 5), p-values have been determined using a one-sample t-test. **f**, Gonadal WAT was harvested from P4 male mice, and whole mount staining against PDGFRα was performed to label APCs. **g**, At P4, APCs are all ciliated

visualized by labeling against ARL13B (red, cilia) and γ -Tubulin (magenta, basal body). **h**, Quantification of the ciliary SMO localization in P4 gWAT APCs after treatment with 1 μ M SAG or vehicle control (H₂O) for 24 h. Each data point represents one cilium from one tissue (>20 cilia per condition), p-values have been determined using an unpaired students t-test. **i**, Relative mRNA expression of *Gli1* and *Ptch1* in P4 gWAT APCs after treatment with 1 μ M SAG or vehicle control (H₂O) for 24 h assessed by qRT-PCR. The expression was normalized to control treated expression values. Each data point represents one tissue (n = 3), p-values have been determined using an unpaired students t-test. **j**, Relative mRNA expression of Hh target genes in P4 precursor tissue from *Pdgfra*^{+/+} *Ift20*^{flox/flox} and *Pdgfra*^{+cre} *Ift20*^{flox/flox} mice. Expression was normalized to expression values of *Pdgfra*^{+/+} *Ift20*^{flox/flox}. Each data point represents a biological replicate (n = 5-6), p-values were calculated by a one-sample t-test. **k**. See **j**. for fibrogenic markers. Scale bars are indicated. All data are shown as mean \pm SD.

2) *The authors claim very early in the paper (at the end of Figure 3), that Bbs8 deficiency cause defects in P1 APC and therefore primary cilia dysfunction cause defects in P1 APC. However, at this stage primary ciliogenesis and ciliary signaling upon Bbs8 loss have not been shown. The results presented at this stage therefore do not support the conclusion. Characterization of primary cilia in Bbs8 null mice should be shown earlier if the authors want to claim that ciliary dysfunction cause the phenotype.*

We thank the reviewer for this comment and agree that the conclusion has been drawn to early. We have modified the text accordingly, only mentioning that loss of *Bbs8*, but not primary cilia dysfunction, causes a certain phenotype.

3) *The obesity phenotype is observed in adult mice (10 weeks): Why? does the appearance of the obesity phenotype in the adult correlates with activation of primary ciliogenesis and ciliary signaling in APC only the adult tissue? The dynamics of ciliogenesis during development of this tissue would help understand the phenotype. Does BBS8 ablation in the adult mice cause obesity?*

It has been demonstrated already that hyperphagia is one of the major but not the only contributor to obesity in *Bbs* mutants (Rahmouni et al., JCI 2008). This has been further verified using mice, in which primary cilia ablation has been induced after weaning, resulting in hyperphagia and obesity development in adult mice (Davenport et al., Curr Biol. 2008). Thus, the uptake of excess energy due to hyperphagia needs some time until it manifests as a significant expansion of the adipose tissue depots, which is why the obesity phenotype gradually develops and manifests in the adult state. Previous studies have shown that specifically deleting cilia in POMC neurons in the hypothalamus during embryonic development also results in hyperphagia (Lee et al., Nat Commun. 2020). Therefore, embryonic neuronal ciliogenesis seems to be critical for regulating adulthood energy balance. The APCs in the adult adipose tissue are always ciliated (see Hilgendorf et al., Cell 2019). To investigate whether this is true from early development onwards, we analyzed the presence of cilia on APCs in gWAT, as this develops postnatally and allows to investigate APCs before they differentiate into adipocytes (see Han et al., Development 2011). As shown above in response to the first comment, at P4 when only PDGFR α -positive APCs and no mature adipocytes are present, the cells are ciliated and Hh responsive. Thus, primary cilia on APCs are present from the beginning of tissue development, and primary cilia dysfunction impacts APCs from early development on (as seen for *Pdgfra*^{+cre}, *Ift20*^{flox/flox} mice see answer to point 2). We have included this data set in Fig. S6 in the revised manuscript. Whether loss of *Bbs8* in adult mice would also lead to obesity, needs to be investigated in further studies but based on data presented here and by others, it is highly likely that those mice develop obesity.

4) *In figure 3, additional stainings of collagen fibers in mouse models is necessary to reach the conclusions on ECM remodeling which is mainly based on gene expression studies.*

We thank the reviewer for this comment and have performed whole-mount labeling of the adipose tissue from lean *Bbs8*^{+/+} and *Bbs8*^{-/-} mice (5 weeks) as well as from *Pdgfra*^{+/+} *Bbs8*^{fl/fl} and *Pdgfra*^{Cre/+} *Bbs8*^{fl/fl} mice (7 weeks). We labeled for CD9 and fibronectin and could demonstrate that both are dramatically increased in lean *Bbs8*^{-/-} and *Pdgfra*^{Cre/+} *Bbs8*^{fl/fl} mice compared to *control* mice (see figures below). We have included the data in Fig. 3i and Fig. 4f and in the Fig. S3i and S4g.

CD9^{high} APCs are present in the gWAT of *Bbs8*^{-/-} and *Pdgfra*^{Cre/+} *Bbs8*^{fl/fl} mice. Whole mount staining of gWAT of (a) *Bbs8*^{+/+} and *Bbs8*^{-/-} mice (5 weeks) and (b) *Pdgfra*^{+/+} *Bbs8*^{fl/fl} and *Pdgfra*^{Cre/+} *Bbs8*^{fl/fl} mice (7 weeks). Tissues were stained with DAPI (blue), and against PDGFRα (white) to label APCs and CD9 (magenta). Scale bar = 50 μm, 20 μm.

Whole mount staining shows fibrogenic phenotype in gWAT of *Bbs8*^{-/-} and *Pdgfra*^{Cre/+} *Bbs8*^{fl/fl} mice. Whole mount staining of gWAT of *Bbs8*^{+/+} and *Bbs8*^{-/-} mice (5 weeks) and *Pdgfra*^{+/+} *Bbs8*^{fl/fl} and *Pdgfra*^{Cre/+} *Bbs8*^{fl/fl} mice (7 weeks). Stained with DAPI (blue), and against Fibronectin (FBN, green) to mark the extracellular matrix, and PDGFR α (white) to label APCs. Scale bar = 50 μ m.

5) In figure 4G, the authors conclude that APC cells are the most active in cellular communications but P2 and P1 cells are very abundant in the analysis in comparison to endothelial cells and pericytes. How does it affect the probability of interactions in this analysis?

We are aware that differences in abundance can impact the finding of the probability of interactions. However, in our analysis, we see effects independent of the size of a cluster (i.e., APC P2_0 vs P2_1; or P3, which is a small cluster but still shows a high interaction strength compared to a bigger cluster), which further strengthens the notion that the APC_P2 is in fact the population that strongly interact with vascular cells. Therefore, we think our analysis is reliable. Still, to address the reviewers concern, we have also tried to down sample the clusters.

However, down sampling reduces the statistical power and, thereby, increases the chance of identifying both false positive and false negative interactions. With fewer cells in down sampled clusters, the variability of interactions would increase, making the identified communication less reliable. As CellChat analyzes average expression levels of ligands and receptors within each cluster, the results are not directly biased by how many cells are in a group. Moreover, the pathways that emerged from this analysis were highly dependent on which cells were randomly included in a cluster. Thus, we refrain from including this analysis in the manuscript.

6) In figure 4J, the presented data does not allow to visualize easily the increase in the collagen-laminin interaction in *Bbs8* null cells in comparison to wild-type cells. The interaction strength appears similar in wt vs knockout cells. Another visualization should be used, and additional studies should be conducted to conclude on this signaling event (RT-qPCR, western blotting experiments using purified cells) and additional imaging studies which demonstrate localization of the cells *in situ* and expression of the signaling components could be performed.

For clarity, we added a bar graph below demonstrating the interaction strength of the APC and FPC populations towards the endothelial cells in *Bbs8*^{+/+} and *Bbs8*^{-/-} mice (see figure below and new Fig. S4h). Since the representation used in the old Fig. 4j (now Fig. 4k in the revised manuscript) allows for more data to be shown in a single graph - namely all sources and receivers including the probability of the interaction (size of the dots) and p-values (color-coded) - we would keep it in the main figure while the bar plot representation would be shown as supplementary figure Fig. S4h.

Furthermore, as stated in the answer to the previous comment, we have performed *in situ* labeling of ECM components to further analyze the signaling (Fig. 3i, Fig. 4f, Fig. S3i, S4h).

Interaction of APCs and endothelial cells via collagen and Laminin pathways is increased in *Bbs8*^{-/-} APCs compared to *Bbs8*^{+/+} APCs. Collagen and Laminin interaction scores in the APC subclusters interacting with endothelial cells.

7) In figure 5b, stimulation of Hedgehog signaling with SAG should be controlled via expression of Hedgehog component expression and processing (RNA and proteins).

We have now analyzed the ciliary SMO localization using immunocytochemistry (see figure a, b below) and investigated the expression of Hh target genes (see figure c below), demonstrating that SAG stimulation of 3T3-L1 cells promotes ciliary SMO localization and Hh target gene expression. Furthermore, treatment of 3T3-L1 cells with SAG induced the expression of GLI1 protein and inhibited processing of GLI3 to its repressor form (GLI3R) (see figure d-g below). The data has also been included in Supplementary Fig. S5 of the revised manuscript.

Hedgehog signaling 3T3-L1 cells. **a.** Fluorescence confocal images of 3T3-L1 cells, labeled against Smoothed (green, SMO, 5 px shift indicated by the green arrow), and ARL13B (red, cilia), γ -Tubulin (magenta, basal body), and with DAPI (blue). Cells were treated with H₂O (control) or 100 nM SAG for 24 h. Scale bars are indicated. **b.** Quantification of the ciliary SMO localization from (a). Each data point represents a biological replicate (n = 3; >20 cilia per n), p-value has been determined using an unpaired Student's t-test. **c.** Relative mRNA expression of *Gli1* and *Ptch1* in 3T3-L1 treated with SAG (100 nM) for 24 h. Expression was normalized to expression values of the vehicle control (H₂O). Each data point represents a biological replicate (n = 6), p-values have been determined using a one-sample t-test. **d-g.** Western Blot analysis of GLI transcription factors GLI3R (**d**) and GLI1 (**f**). β -TUBULIN was used as a loading control. **e.** Quantification of GLI3R by normalizing the band intensity to the loading control (β -TUBULIN). **g.** Quantification of GLI1 by normalizing the band intensity to the loading control (β -TUBULIN). Each data point represents a biological replicate (n = 3-4), p-values have been determined using an unpaired Student's t-test.

8) In Figure 5i, the analysis of the impact of vismodegib should be conducted on additional Hh genes, since it is not possible to interpret appropriately the impact of vismodegib with the presented data.

We were unsure about the reviewer's comment and performed an extensive review of the literature. Here, we did not find additional Hedgehog (Hh) pathway genes that are commonly used in the context of vismodegib and/or cyclopamine studies (Berman et al., Nature 2003 (PMID: 14520411); Yao et al., JBC 2018 (PMID: 30573683); LoRusso et al., Clin Cancer Res 2011 (PMID: 213007629); Wong et al., Clin Cancer Res 2011 (PMID: 21610148); Robarge et al., BMCL 2009 (PMID: 19716296), De Jesus-Acosta et al., Nature 2020 (PMID: 31857726); Chen et al., JPS 2018 (PMID: 30064819). We have considered *Gli1* and *Ptch1* as key genes in the Hedgehog signaling pathway to evaluate the effects of the Hedgehog pathway activators and inhibitors throughout our manuscript but are open to suggestions from the reviewer to consider additional genes.

9) The data on SMO inhibition are too preliminary. The authors test whether Hh signaling inhibition via SMO inhibition represses Hh signaling (induced in *Bbs8* null cells). The analysis of the impact of vismodegib should be conducted on additional Hh genes. Additionally, the impact of vismodegib on other ECM genes that are upregulated in P1 APC in *Bbs8* null mice should also be analyzed. Finally, if the authors want to test whether SMO inhibition is sufficient to repress Hh signaling in *Bbs8* null mice, as claimed, the authors should perform in vivo treatment of *Bbs8* null mice and assess the impact on Hh signaling in APC P1 after treatment and whether or not repressing Hh signaling correct the phenotype in mice.

We agree with the reviewer and have now included more ECM genes in the revised manuscript (see figure below and Fig. 5j revised manuscript). The results show an up-regulation of fibrogenic marker genes in *Bbs8*^{-/-} mice compared to *Bbs8*^{+/+} mice, which is reduced by vismodegib treatment.

Fibrogenic gene expression after *in vitro* vismodegib treatment. Relative mRNA expression of fibrogenic marker genes in lineage-depleted SVF from *Bbs8*^{+/+} and *Bbs8*^{-/-} mice treated with vismodegib (2 μ M) for 24 h. Expression was normalized to control-treated *Bbs8*^{+/+} expression values. Each data point represents one animal (n = 4-5). All data are shown as mean \pm SD, p-values were calculated by two-way ANOVA.

We also followed the reviewer's suggestion and performed *in vivo* treatment of *Bbs8*^{-/-} mice with vismodegib. To this end, we injected *Bbs8*^{-/-} mice in their young, lean state with vismodegib intraperitoneally every other day for a total of 20 days. Following the treatment period, the effect of vismodegib on correcting the Hh phenotype was assessed compared to untreated *Bbs8*^{-/-} P1 cells. Analyzing SMO localization in P1 showed that vismodegib treatment significantly reduced SMO localization to the primary cilium compared to *Bbs8*^{-/-} P1 cells (see figure below, a). Since *Gli1* was the only Hh target gene significantly upregulated in *Bbs8*^{-/-} P1 cells, we specifically investigated whether vismodegib treatment would reduce *Gli1* expression levels. Indeed, *Gli1* expression seems to be decreased following vismodegib treatment compared to untreated *Bbs8*^{-/-} P1 cells (see figure below, b). To test whether suppression of Hh signaling would also revert the fibrogenic fate change in *Bbs8*^{-/-} mice, we analyzed the fibrogenic CD9^{high} population in the gWAT of *Bbs8*^{-/-} mice following vismodegib treatment. We observed a significant reduction in the CD9^{high} population compared to untreated *Bbs8*^{-/-} animals (see figure below, c). These results support our hypothesis that vismodegib can counteract ectopic Hh signaling and may act as an anti-fibrotic agent, as also suggested by others (Philips et al., PLOS ONE 2011; Pratap et al., J. Drug Target 2012). We have included the data in Fig. 5k-m.

***In vivo* vismodegib treatment ameliorates ectopic Hh signaling and fibrogenic fate change in *Bbs8*^{-/-} mice.** **a.** Quantification of the ciliary SMO localization in P1 cells from untreated *Bbs8*^{-/-} animals or following *in vivo* vismodegib injections. The average SMO intensity of each cilium was normalized to control-treated *Bbs8*^{+/+} intensity values. Each data point represents a biological replicate (n = 3; >4 cilia per n), p-value has been determined using an unpaired Student's t-test. **b.** Relative mRNA expression of *Gli1* in *Bbs8*^{-/-} P1 cells isolated after *in vivo* vismodegib injections. Relative expression was calculated in relation to control-treated *Bbs8*^{+/+} expression and then normalized to untreated *Bbs8*^{-/-} P1 cells *Gli1*

expression values. P-value has been determined using a one-sample t-test. Each data point represents one animal (n = 3). **c.** Quantification of CD9^{high} cell surface expression on PDGFR α ⁺ cells from *Bbs8*^{-/-} mice untreated or following *in vivo* vismodegib injections. Data are shown as mean \pm SD, p-values have been determined using an unpaired Student's t-test (n = 3-7).

Minor concerns that should be addressed

10) *Stainings of basal bodies would reinforce the observations that Arl13b structures presented in the paper are indeed primary cilia.*

We have included a basal body marker (gamma-tubulin) in the revised manuscript (see figure below and Fig. 1e revised manuscript).

Ciliated APC subpopulations. Representative confocal images of FACS-sorted P1, P2, and P3 cells showing cilia based on the presence of ARL13B and the basal body marker γ TUBULIN. Scale bar = 20 μ m and 5 μ m.

11) *In differentially expressed gene analysis, the authors indicate that many genes are differentially expressed in the P1 population upon *Bbs8* loss. Why does the principal component analysis does not segregate wt from ko cells (in PC2) if it is the case?*

We indeed reveal that the P1 population shows the highest number of differentially expressed genes compared to P2 and P3. The analysis has been performed to reveal DEGs for each subpopulation between *Bbs8*^{+/+} and *Bbs8*^{-/-} mice. The number of DEGs (as stated in Fig. S3b) does not seem to be the driving force in segregating PC 1 and 2. Rather, the difference between the subpopulations seems to be driving the segregation of the PCs here. We acknowledge that the impact of *Bbs8*^{-/-} seems to be lower in rank for separating PCs. However, the general differences in the P1 between *Bbs8*^{+/+} and *Bbs8*^{-/-} mice remain.

12) *The panels in Figure 5 are not appropriately cited.*

We have changed this accordingly.

13) *There is a typo in Als(o), page 13.*

We have corrected this mistake in the revised version.

Referee #3:

This study investigates the activity of adipocyte precursor cells (APCs) and remodeling of adipose tissue in *Bbs8* null mice, which exhibit ciliary dysfunction and serve as a model for Bardet-Biedl syndrome. Deletion of *Bbs8* globally or with the *Pdgfra*-Cre driver (which targets APCs) leads to adult-onset obesity with corresponding adipose tissue growth and pathological remodeling. APCs from the adipose tissue of the null mice display a fibroblast phenotype shift, including increased expression of *Cd9* and many ECM genes. In vitro studies show that APCs isolated from adipose tissue of *Bbs8* KO mice are less adipogenic. Additional studies indicate that hedgehog signaling is increased under basal conditions in APCs from *Bbs8* deficient mice, and that this contributes to the activation of fibroblast genes. Overall, the studies are well-executed from a technical perspective and there are some interesting and important observations. The effect of FFAR4 agonist on APC differentiation (Fig. 1I, J) is striking. However, there are significant concerns, especially regarding the model and interpretations (#1) that should be addressed as outlined below:

*1. The central concern relates to whether the observed effects in *Bbs8* knockout mice are due to direct actions of ciliary signaling (*Bbs8* function) on APCs or secondary to the obesity. Both *Bbs8* deficient mouse models utilized develop obesity (and presumably insulin resistance), which is known to strongly modulate APC activity, including activation of *Cd9* and fibroblast phenotypes (e.g PMID: 28215843). All the presented data suggest that the effects of *Bbs8* KO on APCs are secondary to obesity/insulin resistance, which has not been adequately considered and diminishes novelty and impact of the study.*

*The authors could do a side-by-side comparison of APCs from HFD-induced obese vs. *Bbs8* KO mice to see if there are any differences. It may also be possible to re-organize the paper to emphasize the effects of obesity on APCs in this model of obesity, rather than emphasizing the direct actions of cilia signaling on APCs. The data in this regard nicely advance previous work in the field.*

The reviewer is correct that analyzing APCs in the obese state would not allow to make proper conclusions due to confounding factors that occur during obesity. In fact, we have verified results by others that in a diet-induced obesity (DIO) model using wild-type mice (*Bbs8*^{+/+} DIO), the P1 cells are reduced compared to age-matched (18-20 wks) lean *Bbs8*^{+/+} mice (see figure below).

This is exactly why we focused on the lean state in *Bbs8*^{-/-} mice, to only focus on the cilia-dependent effects in the adipose tissue independent of obesity. All the data presented (Fig. 3 onwards) has been performed in lean *Bbs8*^{-/-} mice (5-9 weeks). We are sorry if we did not make this clear enough in the current version of the manuscript and have changed this accordingly.

We also followed the reviewer's suggestion and compared the results from DIO-WT to experiments using obese *Bbs8*^{-/-} mice, at least for the frequency analysis of the different APC populations. This revealed that both, DIO-WT and obese *Bbs8*^{-/-} mice show the reduction in P1 but only *Bbs8*^{-/-} mice show an increase in P3 (see figure below). Altogether, APCs in *Bbs8*^{-/-} mice show a cell fate already in the lean state and obesity imposes further changes in the APCs from *Bbs8*^{-/-} mice.

Changes in APC subpopulations. a. Frequency distribution of P1-P3 from the total APC pool from 18-20 weeks old $Bbs8^{+/+}$ mice on chow diet or on high-fat diet ($Bbs8^{+/+}$ DIO), and obese $Bbs8^{-/-}$ mice. P-values were calculated by Šidák-multiple comparison test, individual dots represent individual mice.

2. Related to above point, all of the presented in vitro data assess APCs isolated from lean (control) vs. obese $Bbs8$ deleted mice. Does acute $Bbs8$ deletion in APCs from lean mice alter adipogenic vs. fibroblast profile? Do the Bbs -KO APCs display a phenotype when isolated from young mice before body weight divergence?

We are sorry for the confusion, but all the data presented are generated from lean control and lean $Bbs8^{-/-}$ mice (5-9 weeks), before the onset of obesity. Thus, APCs indeed change their fate in lean $Bbs8^{-/-}$ mice, as we show in the manuscript. We have emphasized this in the revised version of the manuscript whenever possible.

3. The presented data strongly suggests that $Pdgfra$ -Cre deletes $Bbs8$ in the brain/hypothalamus, leading to alterations in food intake and energy expenditure that underlie obesity development. The authors should examine $Bbs8$ deletion in brain regions of the $Pdgfra$ -Cre/ $Bbs8$ KO mice.

We agree with the reviewer that this is an important point to consider when analyzing the $Pdgfra^{Cre/+} Bbs8^{fl/fl}$ mice and a contribution of a brain/hypothalamus phenotype cannot be fully excluded. However, we only assessed the lean phenotype at 7 weeks and focused on the phenotype in the adipose tissue independent of obesity. Thus, alteration in food intake and energy expenditure have not been considered at this state but can be considered in the future when analyzing obese mice. We also demonstrate that stimulation of Hh signaling in APCs is sufficient to promote fibrogenic gene expression and that blocking Smo in $Bbs8^{-/-}$ APCs is sufficient to rescue the phenotype. Thus, although we cannot rule out the contribution of other cells than APCs to the fibrogenic switch observed in the lean state, they seem to be the major contributor.

4. There does not seem to be defining markers of "FPCs" (Fig. 4)- is it more the lack of adipogenic/progenitor genes? Is $Cd9$ expression enriched in this population?

We have annotated the FPCs based on the combination of their Top 10 Marker Genes and the GO Term analysis of their upregulated genes (see Fig. S4f, Tab. S3). Additionally, we have included the signature genes (DEGs) when comparing FPCs to the other APC P1-P3 genes (see Table S4). The reviewer is correct that FPCs share some of their defining markers with

other APC populations, which might be due to a high plasticity in mesenchymal cells and a heterogeneous origin of fibrogenic cells. However, FPCs are lacking in defining progenitor or adipogenic genes (see figure below). Regarding Cd9 expression, we were unable to detect its expression in the scRNAseq dataset across any cluster. This likely reflects the fact that RNA expression does not always correlate with protein expression, as highly expressed RNA does not necessarily correspond to high protein levels and *vice versa* (Gardner et al., *npj Syst Biol Appl.* 2024). However, CD9 is a great marker for flow cytometry, and we also established CD9 labeling in whole mount adipose tissue (see figure below). This approach nicely supports our flow cytometry data, demonstrating an increased expression of CD9 in WAT from lean *Bbs8*^{-/-} mice compared to *Bbs8*^{+/+} mice (see Fig. 3i in the revised manuscript).

Gene signature of FPCs. Heat-map depicting gene expression of APCs P1-P3 for different progenitor and adipogenic genes across all samples.

CD9^{high} APCs are present in the iWAT of *Bbs8*^{-/-} mice. Whole mount staining of iWAT of *Bbs8*^{+/+} and *Bbs8*^{-/-} mice. Stained with DAPI (blue), Pdgfra (white) to label APCs, and CD9 (magenta). Scale bar = 50 μm, 20 μm.

5. Fig. 5. The effect of basal HH signaling is interesting. Additional fibroblast genes including Cd9 should be assessed in panels j,k to determine if this treatment broadly alters cell phenotype. Also, the results in panels j,k should be analyzed together in the same graph so the reader can properly evaluate effects of vismodegib treatment on the cells.

We agree with the reviewer and have changed and amended the figure accordingly. We have combined the data and, as stated above in the response to the previous comment, we have included more genes. This data is shown below and in the revised manuscript (see figure below and Fig. 5j).

Fibrogenic gene expression after vismodegib treatment. Relative mRNA expression of fibrogenic marker genes in lineage-depleted SVF from *Bbs8*^{+/+} and *Bbs8*^{-/-} mice treated with vismodegib (2 μM) for 24 h. P-values were calculated by an unpaired Student's t-test. Expression was normalized to control-treated *Bbs8*^{+/+} expression values (indicated as the dotted line). Each data point represents one animal. All data are shown as mean ± SD, p-values were calculated by two-way ANOVA.

6. Fig. 4j. It seems that the main effect of *Bbs8* deficiency is an increase in collagen in all APC types, rather than a specific effect on P2-1 as was concluded.

Indeed, the reviewer is correct that *Bbs8* deficiency increases collagen expression in all APCs. We apologize if this was not well described. The results shown in Fig. 4j (now Fig. 4k in the revised version) depict the general probability of each cell type for interacting with the other cell types via the collagen or laminin pathway. However, Fig. 4i in the revised version depicts the highest strength for the interaction, which lies within the P2_1 cluster (see yellow color representing stronger interactions). In general, we conclude that *Bbs8* deficiency drives collagen expression on all APCs, and, furthermore, we have identified APC P2_1 to be the cell type that is interacting strongly with endothelial cells via this pathway.

7. (minor) Fig. 1d. It is unclear what this data represents. The results section states that P1 was the most abundant APC subpopulation based on these data. However, it is unclear if P1 express higher levels of SCA1, which would alter the interpretation. The scRNAseq profiles from several labs do not suggest such a dramatic difference in the abundance of P1 vs other APC subtypes.

We are sorry that we were not clear in description of Fig. 1d. Indeed, this plot demonstrates the abundances of the different subpopulations within the total APC pool, which is defined as Lin-/CD29+/Sca1+ (see gating scheme in Fig. S1a). We have changed the title of the y-axis accordingly.

The reviewer mentioned that the high abundance of the P1 population is not in accordance with other literature. Throughout all our experiments, P1 shows an abundance of 60-80% based on flow cytometry analysis. However, this was measured in rather young animals (6-9 weeks), where the P1 stem-cell like precursor population in gWAT is bigger compared to an older age (see figure a below). Moreover, we have analyzed iWAT and gWAT in comparison and there are indeed depot-specific differences in the abundance of the subpopulations (see figure b below).

APC subpopulations at different ages and in different AT depots. **a** Frequency distribution of P1-P3 from the total APC pool from 18-20 weeks old *Bbs8*^{+/+} mice on chow diet or on high-fat diet (*Bbs8*^{+/+} DIO), and obese *Bbs8*^{-/-} mice. **b** Frequency distribution of P1-P3 from the total APC pool from the iWAT or the gWAT of *Bbs8*^{+/+} mice. P-values were calculated by Šidák-multiple comparison test, individual dots represent individual mice.

Dear Dagmar,

Thank you for submitting a revised version of your manuscript. I sincerely apologise for the protracted assessment process for your manuscript due to the delay in reviewer report submission and the holidays here in South Germany.

We have now received input from all original reviewers. While reviewer #1 and #3 are satisfied with the revisions, reviewer #2 finds that some of their original points have not been sufficiently clarified. Please address these points with further discussion and toning down of the statements. For their point 2, please add the requested quantifications if feasible.

Additionally, there are a few editorial points that need addressing before I can extend official acceptance of the manuscript:

1. Please submit up to five keywords.
2. Please make sure the funding information is correct and identical both in the manuscript and our online system. Currently, SFB1366 - Project-ID - 394046768, DFG, German Research Foundation) - project numbers 388168919, 388158066, 01EO2107, and 266686698, 216372545 and 471514137 are missing from the online system.
3. Please submit a complete author checklist, which you can download from our author guidelines (<https://www.embopress.org/pb-assets/embo-site/EMBO%20Press%20Author%20Checklist-1642513524327.xlsx>). Please insert information in the checklist that is also reflected in the manuscript. The completed author checklist will also be part of the Review Process File.
4. All Materials and Methods need to be described in the main text using our 'Structured Methods' format. According to this format, the Methods section includes a Reagents and Tools Table (listing key reagents, experimental models, software and relevant equipment and including their sources and relevant identifiers) followed by a Methods and Protocols section describing the methods, ideally using a step-by-step protocol format. The aim is to facilitate adoption of the methodologies across labs. Please download and fill our Reagents and Tools Table template (.docx), which you can find in our author guidelines: <https://www.embopress.org/page/journal/14602075/authorguide#structuredmethods>
When submitting your revised manuscript, please do not include the Reagents and Tools Table in the Methods section of the manuscript but upload it as a separate file choosing the file type "Reagent Table".
An example of a Method paper with Structured Methods can be found here: <https://www.embopress.org/doi/10.15252/msb.20178071>.
5. Please add the content from the 'Code Availability' section to the 'Data Availability' section. More information about the format of this section can be found here: <https://www.embopress.org/page/journal/14602075/authorguide#dataavailability>.
6. CRediT has replaced the traditional author contributions section because it offers a systematic, machine-readable author contributions format that allows for more effective research assessment. Please remove the Authors Contributions from the manuscript and use the free text boxes beneath each contributing author's name in our online submission system to add specific details on the author's contribution. More information is available in our guide to authors.
7. Please rename 'Competing interest statement' section into 'Disclosure and competing interests statement' (further info: <https://www.embopress.org/page/journal/14602075/authorguide#conflictsofinterest>).
8. Please update references according to The EMBO Journal style - where there are more than 10 authors on a paper, the first 10 should be listed, followed by 'et al.' Please see further information here: <https://www.embopress.org/page/journal/14602075/authorguide#referencesformat>
9. Please move 'References' after 'Disclosure and competing interests statement'
10. Please rename the file with suppl. information into 'Appendix' and add a table of contents with page numbers to the first page. Please correct the nomenclature to Appendix Figure S1 etc. and Appendix Table S1. Tables S6-8 should be renamed Appendix Table S1 - S3. Tables S1 - 5 should be renamed Dataset EV1 - 5, and each dataset should have a legend added to the file in a separate tab/worksheet.
11. Please ensure that all tables and datasets need to be called out in the correct sequential order.
12. Please remove the list of supplemental information from the manuscript text file.
13. Our data editors have flagged the following issues in figure legends that need correcting:
 - Please note that the figure S3 H is mislabeled as figure S3 G in the manuscript. This needs to be corrected.
 - Please provide the exact p values in the legends of figures 2B, F, H; 3A, 5H, J; S5J, N; S6D, H.
 - Please indicate the statistical test used for data analysis in the legends of figures S3C-F
 - Please note that the box plots need to be defined in terms of minima, maxima, centre, bounds of box and whiskers, and percentile in the legends of figures 3C, F; 5A, K, L, M; S1 B.
 - Please provide information on the number and nature of replicates in the legends of figures 3C, D; 4B, S3C, E.
14. Papers published in The EMBO Journal are accompanied online by a 'Synopsis' to enhance discoverability of the manuscript. It consists of A) a short (1-2 sentences) summary of the findings and their significance, B) 3-4 bullet points highlighting key results and C) a synopsis image that is 550x300-600 pixels large (width x height, jpeg or png format). You can either show a model or key data in the synopsis image. Please note that the image size is rather small and that text needs to be readable at the final size.

Please let me know if you have any questions regarding any of these points. You can use the link below to upload the revised

files.

With kind regards,

leva

leva Gailite, PhD
Senior Scientific Editor
The EMBO Journal
Meyerhofstrasse 1
D-69117 Heidelberg
Tel: +4962218891309
i.gailite@embojournal.org

We realize that it is difficult to revise to a specific deadline. In the interest of protecting the conceptual advance provided by the work, we recommend a revision within 3 months (16th Sep 2025). Please discuss the revision progress ahead of this time with the editor if you require more time to complete the revisions.

Referee #1:

The authors have responded to all my comments satisfactorily and have significantly improved the manuscript with the new additions.

Referee #2:

In the revised manuscript No. EMBOJ-2024-118260R1, the authors made significant changes to the paper, including textual changes and the addition of new data. The modifications reinforce some aspects of the paper. However, many of the initial comments were not fully addressed. Additionally, the conclusion reached by the authors for some of the newly presented data are not strongly supported by the results. The manuscript remains well written and the research reinforce the current knowledge on adipogenesis. However, the authors additional experiments should be conducted to support the conclusions.

1) In the first round of revision, it was difficult to conclude (i) whether or not the adipogenesis defects linked to BBS8 loss associated with a fibrogenic switch of APC is a universal phenotype in different forms of BBS, as discussed by the authors, or a specificity of BBS8 syndrome (ii) the extent to which BBS8-dependent ciliary dysfunction contributes to the phenotype.

The authors aimed at addressing this point by studying hedgehog signaling using different cellular models in which BBS1 or cilia/ciliary signaling regulators were inactivated. The presented data are interesting but they do not fully address the initial comment. The authors simply analyzed the impact of Bbs1 KO or Ift20 KO or Ptch1 KO on Hh signaling in different cellular models without assessing systematically the fibrogenic switch resulting from these other perturbations. Additionally, it is noteworthy that in the BBS1 KO experiment, the impact on ciliary SMO is modest in comparison to the result after BBS8 loss initially presented and it seems that ciliogenesis is affected and may blur the interpretation. This is not discussed.

2) With regard to the new data on CD9 and fibronectin immunostaining presented in Fig3 and 4, quantifications should be conducted on tissue samples from different animals, to reach any kind of conclusion.

3) The data presented on the interaction between APC and endothelial cells mediated by collagen-laminin does not fully address the initial comment. Additional work should be conducted to reach the conclusion that this potential interaction is a genuine interaction.

4) The data on SMO inhibition in Bbs8 null cells using vismodegib indicate that only Gli1 expression is reduced but not Ptch1

(Fig 5i). This is not consistent with the following statement : "Bbs8^{-/-} APCs was sufficient to significantly reduce the expression of Hh target genes, i.e., Gli1 mRNA expression (Fig. 5i)." Can the author explain the reason why Ptch1 expression is not affected upon vismodegib treatment?

Additionally, vismodegib treatment impacts only Col1a1 expression but not Col5a1 or Loxl2. How do the authors interpret this puzzling data?

5) The authors performed in vivo treatment of Bbs8^{-/-} mice with vismodegib to determine whether SMO/Hh signaling inhibition corrects the fibrogenic phenotype. Surprisingly, while ciliary SMO appears to be inhibited in P1 cells, the treatment does not impact downstream Hedgehog signaling (no significant difference in Gli1 expression). Vismodegib treatment results however in a decrease in the % of CD9 high PDGFR high cells. How do the authors explain the fact that Gli1 expression do not change when SMO is inhibited and with their general model ?

Referee #3:

The authors have thoughtfully addressed the points raised by reviewers. The paper is improved, and acceptable for publication.

Referee #1:

The authors have responded to all my comments satisfactorily and have significantly improved the manuscript with the new additions.

We thank the author for acknowledging our efforts.

Referee #2:

In the revised manuscript No. EMBOJ-2024-118260R1, the authors made significant changes to the paper, including textual changes and the addition of new data. The modifications reinforce some aspects of the paper. However, many of the initial comments were not fully addressed. Additionally, the conclusion reached by the authors for some of the newly presented data are not strongly supported by the results. The manuscript remains well written and the research reinforce the current knowledge on adipogenesis. However, the authors additional experiments should be conducted to support the conclusions.

We thank the reviewer for acknowledging that we have significantly improved the manuscript. We have addressed the remaining issues below.

1) In the first round of revision, it was difficult to conclude (i) whether or not the adipogenesis defects linked to BBS8 loss associated with a fibrogenic switch of APC is a universal phenotype in different forms of BBS, as discussed by the authors, or a specificity of BBS8 syndrome (ii) the extent to which BBS8-dependent ciliary dysfunction contributes to the phenotype.

*The authors aimed at addressing this point by studying hedgehog signaling using different cellular models in which BBS1 or cilia/ciliary signaling regulators were inactivated. The presented data are interesting but they do not fully address the initial comment. The authors simply analyzed the impact of *Bbs1* KO or *Ift20* KO or *Ptch1* KO on Hh signaling in different cellular models without assessing systematically the fibrogenic switch resulting from these other perturbations. Additionally, it is noteworthy that in the BBS1 KO experiment, the impact on ciliary SMO is modest in comparison to the result after BBS8 loss initially presented and it seems that ciliogenesis is affected and may blur the interpretation. This is not discussed.*

The three additional models that we analyzed all displayed ectopic Hh signaling. As it has been shown before that the *Ptch1*-KO model induces a cell fate change in a Hh-dependent manner, we focused on the *Ift20*-KO model and analyzed the fibrogenic gene expression response in WAT. Indeed, loss of *Ift20* induced a fibrogenic response. The BBS1-iPSC model does show ectopic activation of Hh signaling using the ciliary SMO localization as a read-out and indeed, the response is not as strong as in *Bbs8*^{-/-} APCs but still significant. This might be due to the different Hh responsiveness in the different cell lines. We have included a sentence in the manuscript, describing that the response is more pronounced in *Bbs8*^{-/-} APCs.

2) With regard to the new data on CD9 and fibronectin immunostaining presented in Fig3 and 4, quantifications should be conducted on tissue samples from different animals, to reach any kind of conclusion.

We thank the reviewer for this useful comment and have included the quantification in Fig. 3j, Fig. 4j, and Fig. S4h.

3) The data presented on the interaction between APC and endothelial cells mediated by collagen-laminin does not fully address the initial comment. Additional work should be conducted to reach the conclusion that this potential interaction is a genuine interaction.

We thank the reviewer for highlighting the importance of this cellular interaction. In fact, laminin and collagen interaction, influencing cell behavior, have been shown for different cellular

interactions. They create a network that supports cell attachment and signaling, which is crucial for cell adhesion, migration, and tissue organization. Based on our scRNA-seq data, the bioinformatic analysis, and the in-situ labeling in WAT, we propose that there is an interaction between APCs and endothelial cells that is augmented in *Bbs8*^{-/-} mice. We will study these interactions in further detail in a follow-up project, which is beyond the scope of this manuscript.

4) *The data on SMO inhibition in Bbs8 null cells using vismodegib indicate that only Gli1 expression is reduced but not Ptch1 (Fig 5i). This is not consistent with the following statement : "Bbs8-/- APCs was sufficient to significantly reduce the expression of Hh target genes, i.e., Gli1 mRNA expression (Fig. 5i)." Can the author explain the reason why Ptch1 expression is not affected upon vismodegib treatment?*

Additionally, vismodegib treatment impacts only Col1a1 expression but not Col5a1 or Loxl2. How do the authors interpret this puzzling data?

The differential response of PTCH1 and GLI1 to SAG stimulation in terms of gene expression can be due to their distinct roles within the Hedgehog (Hh) signaling pathway. Thus, SAG treatment not necessarily induces a similar gene expression response for both genes. To address the reviewer's comment, we have changed the sentence accordingly: *Bbs8*^{-/-} APCs was sufficient to significantly reduce *Gli1* mRNA expression (Fig. 5i)." The same seems to hold true for other downstream target genes like the collagen family although *Col5a1* shows a similar trend, but the data was not significant.

5) *The authors performed in vivo treatment of Bbs8-/- mice with vismodegib to determine whether SMO/Hh signaling inhibition corrects the fibrogenic phenotype. Surprisingly, while ciliary SMO appears to be inhibited in P1 cells, the treatment does not impact downstream Hedgehog signaling (no significant difference in Gli1 expression). Vismodegib treatment results however in a decrease in the % of CD9 high PDGFR high cells. How do the authors explain the fact that Gli1 expression do not change when SMO is inhibited and with their general model?*

Unfortunately, due the restrictions by the local authorities, we were only allowed to perform and n = 3 for this procedure. This is why the reduction in *Gli1* mRNA expression is not significant but shows a clear trend. As it would take another 9 months to get the permit to perform more experiments and in the interest of the 3R principles, we believe that the CD9 data and the *Gli1* expression data together clearly show the effect of vismodegib treatment.

Referee #3:

The authors have thoughtfully addressed the points raised by reviewers. The paper is improved, and acceptable for publication.

We thank the reviewers for acknowledging that we have improved our manuscript and that it is now ready for publication.

Dear Dagmar,

Thank you for addressing the final editorial requests. I am now pleased to inform you that your manuscript has been accepted for publication. Congratulations with a nice study!

Before we forward your manuscript to our publishers, I would like to propose some edits in the manuscript title, abstract and synopsis (please see the attached text file). I have also written a short blurb that will accompany the title of your manuscript in our online table of contents. Please let me know if any corrections or adjustments are needed.

If you have any questions, please do not hesitate to contact the Editorial Office. Thank you once more for your contribution to The EMBO Journal!

With best wishes,

Ieva

Ieva Gailite, PhD
Senior Scientific Editor
The EMBO Journal
Meyerohofstrasse 1
D-69117 Heidelberg
Tel: +4962218891309
i.gailite@embojournal.org
